# Social buffering in rats reduces fear by oxytocin triggering sustained changes in central amygdala neuronal activity

Chloe Hegoburu[1,3], Yan Tang [1,3], Ruifang Niu[1], Supriya Ghosh[1],
Rodrigo Triana Del Rio[1], Isabel de Araujo Salgado [1], Marios Abatis[1],
David Alexandre Mota Caseiro[1], Erwin H. van den Burg[1],
Christophe Grundschober [2] & Ron Stoop [1] ✉

The presence of a companion can reduce fear, but the neural mechanisms underlying this social buffering of fear are incompletely known. We studied social buffering of fear in male and female, and its encoding in the amygdala of male, auditory fear-conditioned rats. Pharmacological, opto,- and/or chemo-genetic interventions showed that oxytocin signaling from hypothalamus-to-central amygdala projections underlied fear reduction acutely with a companion and social buffering retention 24 h later without a companion. Single-unit recordings with optetrodes in the central amygdala revealed fear-encoding neurons (showing increased conditioned stimulus-responses after fear conditioning) inhibited by social buffering and blue light-stimulated oxytocinergic hypothalamic projections. Other central amygdala neurons showed baseline activity enhanced by blue light and companion exposure, with increased conditioned stimulus responses that persisted without the companion. Social buffering of fear thus switches the conditioned stimulus from encoding "fear" to "safety" by oxytocin-mediated recruitment of a distinct group of central amygdala "buffer neurons".

In behavioral therapy, social support can play an important role in immediate as well as long-term reduction of stress and anxiety[1,2] and is effective in treating patients suffering from anxiety disorders, such as post-traumatic stress disorder[3,4]. The knowledge of neural systems and how they interact during social support remains, however, still largely incomplete. A recent neuroimaging study of social support in humans by a non-familiar person suggested the involvement of the prefrontal cortex in combination with the amygdala and mediodorsal thalamus in the downregulation of aversive feelings caused by mild electric shocks and fearful screams[5]. Some preclinical rodent studies have successfully modeled the fear-reducing effects of social support, referred to as the "Social Buffering of Fear" (SBF)[6]. In these models, the presence of a conspecific during the recall of the conditioned stimulus reduced freezing behavior, enhanced extinction and blunted peripheral

cortisol responses in the animal demonstrating fear[7–12]. This may depend on neuronal activity within the infralimbic part of the medial prefrontal cortex (mPFC)[13], possibly through activation of projections to the basolateral amygdala. These projections are well-known for their role in fear extinction, a process in which repeated exposure to a conditioned stimulus gradually reduces the initially acquired fear response[14]. However, whether these projections play a role in SBF remains to be shown.

Interestingly, the neuropeptide oxytocin, either intraventricularly injected or targeting the mPFC, has been reported to facilitate contextual fear extinction in the presence of a companion whereas injection of its receptor antagonist inhibited this socially-induced fear extinction[10]. Oxytocin (OT) is a neuropeptide of nine amino acids and produced in the hypothalamic paraventricular and supraoptic nuclei

[1]Center for Psychiatric Neuroscience, CHUV, Prilly-Lausanne, Switzerland. [2]Roche Pharma Research and Early Development, Neuroscience Discovery, Roche Innovation Center Basel, Basel, Switzerland. [3]These authors contributed equally: Chloe Hegoburu, Yan Tang. ✉e-mail: ron.stoop@unil.ch

(PVN and SON). From these regions, OTergic neurons project throughout the brain where they can release OT and affect synaptic transmission[15–18]. Importantly, OT release can be triggered by social interactions as has been demonstrated for bonding- and reproduction-related behavior in prairie voles as well as in other rodents for social buffering of stress[19–22]. In mice and rats, OT release is likewise promoted during the retrieval of memory of previous encounters with juvenile and adult conspecifics[23]. Endogenous oxytocin, released by the presence of a conspecific, seems therefore an important candidate for mediating SBF.

Remarkably, the role that OT may play in SBF has been studied very little in the amygdala itself, the region of the brain whose activation directly underlies both fear learning and the triggering of the expression of fear. This is even more surprising since the central amygdala (CeA) is known to express high levels of oxytocin receptors (OTRs). We have previously shown that OT originating from the PVN[18] activates a population of GABAergic cells in the lateral CeA (CeL) that project to the medial CeA (CeM) to inhibit fear-stimulating neurons that project from the CeA to the brainstem[24,25]. As OT release in the CeA can be triggered by social stimuli, for example in lactating dams by the presence of pups or intruders[19,26], we hypothesized that OT release in the CeA plays a central role in SBF.

In the current study, we addressed this potential function of OT signaling in the CeA for SBF. We employed a behavioral protocol in which we fear conditioned rats to two distinct auditory stimuli (CS1 and CS2), re-exposed them the next day to only CS1 or only CS2 (one group in the presence of the companion, the other without companion), and re-exposed both groups one day later again to both CS1 and CS2 without the companion in an "SBF retention" test. The fear response to the CS that was replayed in the presence of the companion was acutely reduced, and this reduction persisted the next day while leaving the freezing response to the other CS unaffected. Pharmacological, chemogenetic, and optogenetic interventions as well as multiple single-unit electrophysiological recordings revealed a mechanism that depends on OT signaling from the PVN to the CeA, where it induces concomitant changes in neuronal activity of two distinct types of CeA neurons. Our results suggest the presence of a neural encoding mechanism in the CeA that allows for the acute and persistent switch from fear- to safety-related behavioral responses. This mechanism seems to involve a group of "safety encoding" CeA "buffer neurons" that are recruited during SBF and whose subsequent activation by the CS inhibits the freezing response.

## Results

### Freezing is acutely and persistently decreased by social presence

We measured SBF in two groups of rats, both fear conditioned to a CS1 and CS2 (for internal comparisons), after which only one group was re-exposed to the CS1 in the presence of a companion (Fig. 1a). CS1 and CS2 consisted of respectively 5 and 15 kHz tones of 30 s durations, presented 4 times each, randomly interleaved, at varying intervals of 40–120 seconds, except during fear conditioning when 8 exposures were used. Day 1 started with exposures to CS1 and CS2 for "Tone Habituation" followed by "Auditory Fear Conditioning" (CS1 and CS2 co-terminating with a 2 s, 0.5 mA unconditioned stimulus, or US). "Fear Recall" was measured 24 h later (on "Day 2") revealing equally high freezing levels to both tones in both groups (Fig. 1b). Three hours later all rats were re-tested in a two-compartment cage during 10 min of "social habituation" during which time we introduced, in the adjacent compartment, a naïve (not fear conditioned) companion rat of same sex and weight ("Companion group") or a polystyrene ball ("No companion group"). Both groups were then re-exposed to CS1 only (Fig. 1b). The "Companion group" drastically decreased freezing to CS1, the "No companion" group showed freezing as before (Fig. 1b).

This decrease was instantaneous, occurring immediately to full extent starting from the first of the four CS1 exposures (Supplementary Fig. 1). Thus, social interaction can acutely buffer the freezing response, i.e. can exert a fear-reducing effect. SBF effects were highly significant and frequency independent: they occurred regardless of whether we co-exposed CS1 or CS2 with the companion (Fig. 1b versus Fig. 1c).

Next, we assessed whether this acute SBF might translate into a maintained reduction of fear without the presence of the companion: We re-exposed both groups on "Day 3" to CS1 and CS2 alone in a different context (Fig. 1a, "Retention of SBF"). In the "No companion" group, CS1 and CS2 induced equally high freezing but in the "Companion" group, freezing to CS1 remained reduced as if the companion was still present (Fig. 1b). This "retention of SBF" was also independent of the CS frequency used (see Fig. 1b). Furthermore, it was not caused by extinction, because only exposure to the CS 3 h after fear recall (CS1 in Fig. 1b or CS2 in Fig. 1c) in the absence of the companion did not lead to a significant decrease to CS1 with respect to CS2 on day 3 and vice versa (CS1 and CS2 responses on Day 3 were not significantly different, Two-tailed Mann-Whitney U = 13, $p = 0.48$, $n = 6$, Fig. 1b respectively U = 12, $p = 0.39$, $n = 9$, Fig. 1c).

Housing conditions did not influence acute or retained levels of SBF (effects were similar when animals were group-housed between sessions, Supplementary Fig. 2a), were independent of sex (female rats showed similar levels, Supplementary Fig. 2b) and familiarity with the companion (cage mates induced similar levels, see Supplementary Fig. 2c). Furthermore, SBF effects were specific to the CS-evoked freezing behavior and the amount of fear reduction compared across animals and sessions did not significantly correlate with the level of social interaction (Supplementary Fig. 3). Thus, social interaction reduces freezing both acutely and with retention specifically to the tone that was presented in the presence of the companion. In the remainder of the experiments, we used exclusively CS1 for SBF.

### Acute and retained SBF requires oxytocin release in the CeA

To test whether OT signaling in the CeA might mediate SBF, we infused bilaterally a specific OTR antagonist (OTA, at 0.3 microliters, a volume known to stay restricted to the CeA[25]) on Day 2 before re-exposure to CS1 during social interaction. Although without effect on overall activity before "Fear Recall" (Supplementary Fig. 3a) or, as previously shown, on fear conditioned freezing in non-socially buffered rats[18], OTA completely prevented the SBF-induced acute reduction in freezing to CS1 in accompanied rats (Fig. 2a) and, as a result, the time of social interaction during exposure to the CS (Supplementary Fig. 3b–d). Furthermore, OTA also prevented the retention of SBF on Day 3 (Fig. 2a, compare Fig. 2a with 1b). As we can assume that OTA has washed out within these 24 h[27], these findings indicate that OT signaling in the CeA on Day 2 is required both for the acute and retention effects of SBF.

We also assessed whether maintained increases in local OT signaling on Day 3 could underlie the "retention of SBF". Therefore, we injected OTA just before testing the Retention of SBF. OTA completely blocked the Retention of SBF, causing a full-fledged freezing response to CS1 similar to the freezing response to CS2 (Fig. 2b). It had no additional effects on CS2 responses, these remained high without increasing further on Day 3 (Fig. 2b, $p = 1$). This indicates that endogenous OT signaling in the CeA is persistently required for the maintained decrease in the fear response on Day 3.

We next examined whether OT signaling in the CeA by itself could recapitulate SBF. Instead of exposure to the companion, we pharmacologically activated OTRs by infusing 0.3 microliters of the specific agonist TGOT bilaterally in the CeA before re-exposure to the CS1. Consistent with previous findings[25], TGOT acutely reduced freezing to CS1 on Day 2, but not on Day 3 in the retention test (Fig. 2c). Taken together, OT signaling in the CeA seems necessary and sufficient for

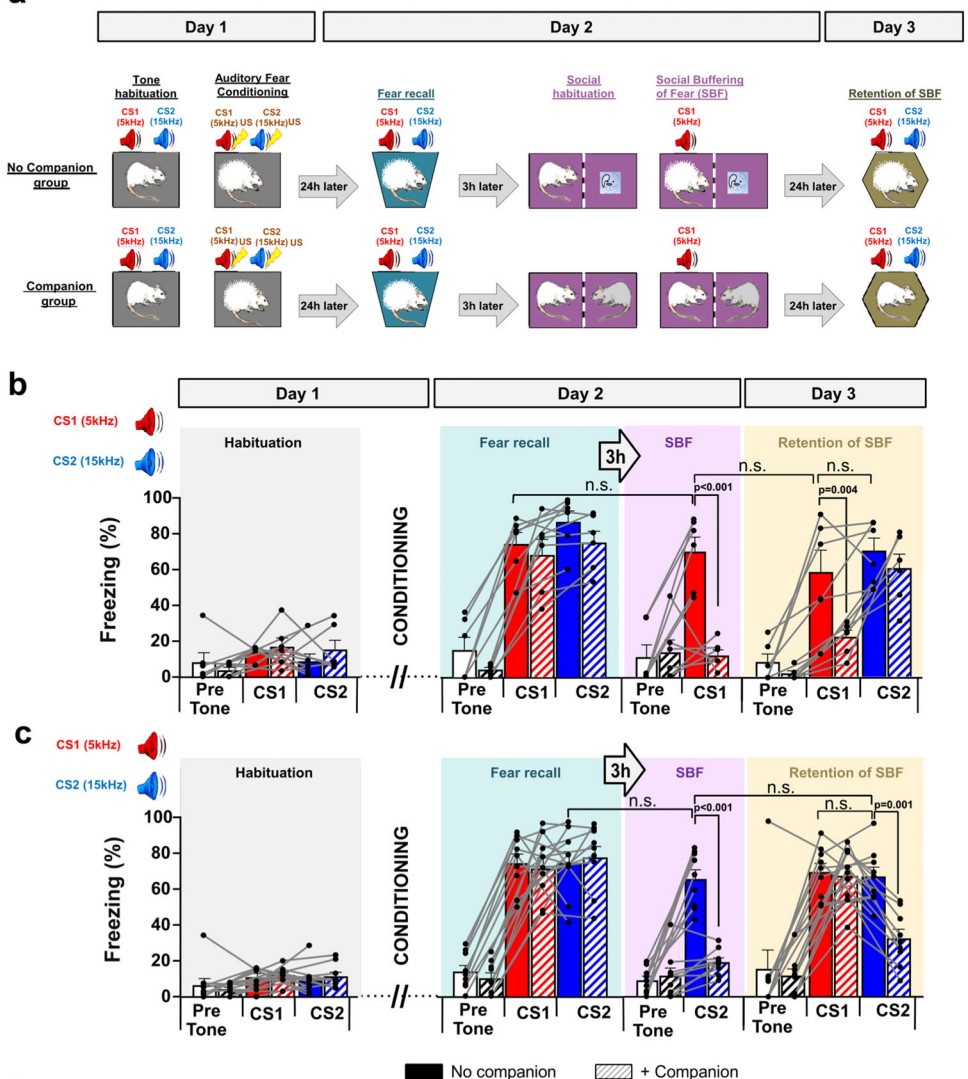

**Fig. 1 | Social buffering of fear: Experimental paradigm and quantification of freezing levels to conditioned stimuli "CS" during different sessions.**
**a** Behavioral protocol. During habituation on Day 1, rats were exposed four times to two different auditory conditioning stimuli (CS1: 5 kHz, red; CS2: 15 kHz, blue) and subsequently fear conditioned by pairing each CS with an electric foot-shock of 0.5 mA (unconditioned stimulus; US). On Day 2 memory of fear was assessed by re-exposure to each CS (fear recall) and, 3 h later, fear memory was again assessed for 10 min during re-exposure to CS1 (as in (**b**) and further experiments) or CS2 (as in (**c**)) after introduction of a polyester ball (No Companion, upper panel) or a companion rat in the adjacent compartment (Companion, lower panel) to test the social buffering of the fear response to the CS (SBF). On Day 3, both CSs were presented again in a new cage without the companion (Retention of SBF).
**b** Freezing levels are equally high upon recall in the "No companion" and "Companion" groups, but freezing to CS1 (5 kHz) is reduced by the presence of the companion (Day 2; "SBF"; F(3,20) = 19.47; $p < 0.001$), and one day later in the

absence of the companion (Day 3; "Retention of SBF", F(5,30) = 18.16; $p < 0.001$; ns, no effects without companion for CS1 and CS2). Freezing levels of rats across the experimental protocol as shown in (**a**) in the absence ("No companion", filled bars, $n = 6$) or presence of a companion ("+Companion", striped bars, $n = 6$). **c** Freezing to CS2 (15 kHz) was high upon recall in both groups, similar to (**b**), and reduced by the presence of the companion (Day 2) and again one day later (Day 3) revealing that both the acute (F(3,32) = 51.48, $p < 0.001$, ns, fear reduction without companion) and memory effects of SBF (F(5,48) = 19.97, $p < 0.001$; ns, no effects without companion for CS2 and CS1) are independent of CS frequency. ("No companion", filled bars, $n = 9$; "+companion", striped bars, $n = 9$). Pre-tone (white bars), basal freezing levels before testing. Two-way ANOVA (pre-tone, CS1, CS2) and group (No Companion, Companion), for each session ("Habituation", "Fear recall", "SBF", "Retention of SBF"). All Bonferroni-corrected $p$-values are indicated in the figures. Individual and mean values ± SEM are shown. See also Supplementary Figs. 2–4.

acute SBF effects, and necessary but not sufficient for persistent freezing reduction.

**PVN OT neuron activity is sufficient for the acute but not retention effects of SBF**
To assess the origin of the OT effects in the CeA we injected, three weeks before the behavioral experiments, an AAV in the PVN expressing an inhibitory designer receptor exclusively activated by designer drug (DREADD) under the OT promoter, thus specifically targeting the (parvo,- and magnocellular) OT neurons. In previous

work, we had shown that this viral construct selectively expresses hM4Di in PVN OT neurons and allows their efficient inhibition by Clozapine-N-oxide (CNO) application[28], which we reconfirmed in the present study (Supplementary Fig. 5a, see also Fig. 4d for expression in the PVN). CNO, intraperitoneally injected 30 min before SBF, impaired both the acute and retained reduction of freezing to CS1, while CNO had no effect in rats that had not been injected with a virus (Fig. 3a) or with a virus expressing ChR2 (Fig. 3a, green bar and supplementary Fig. 4b). Since CNO is known to completely wash out within 24 h[29], these findings indicate that activity of OT neurons in the

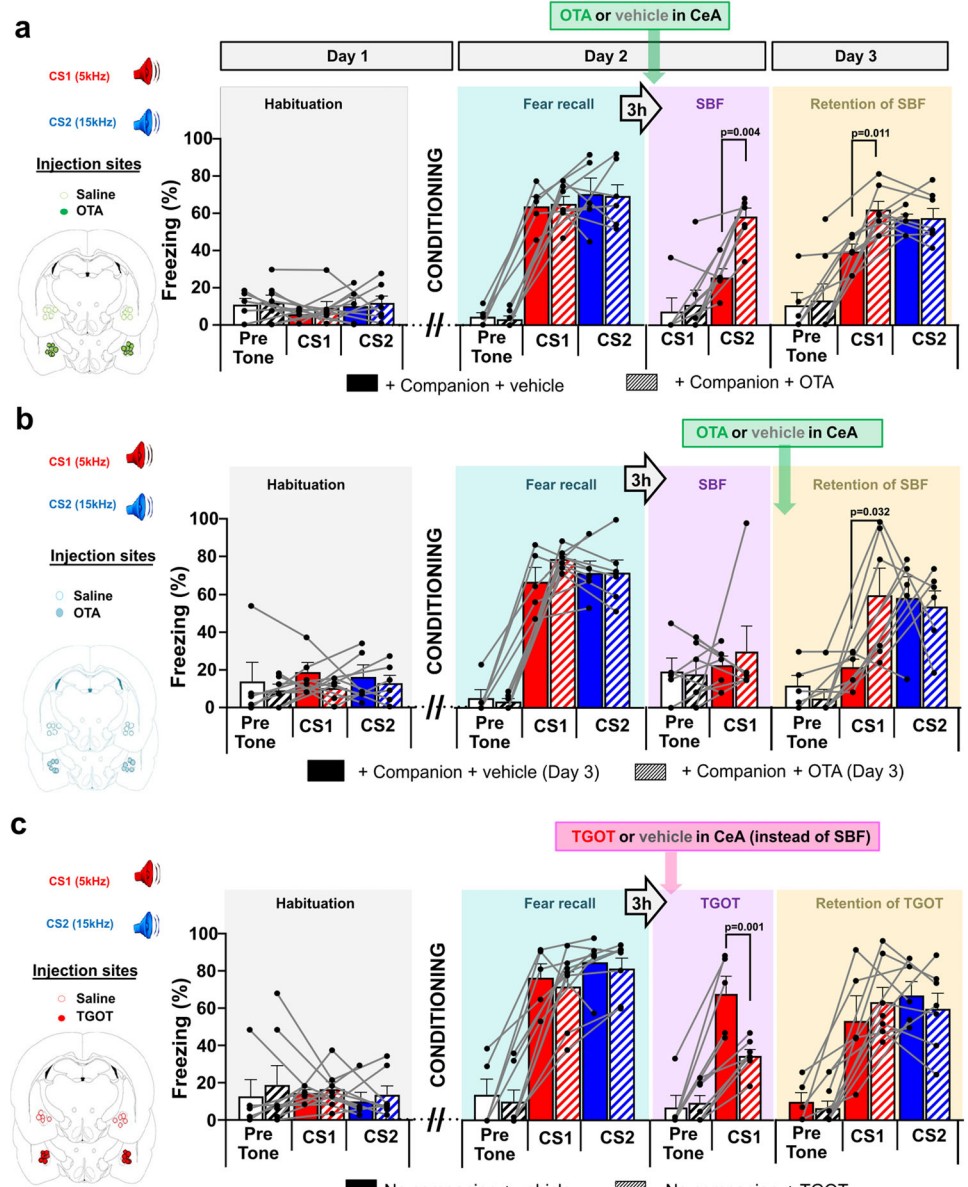

**Fig. 2 | Oxytocin receptor activation in the central amygdala (CeA) is required for the immediate and maintained fear-reducing effects of social buffering, and is sufficient for inducing immediate but not maintained reduction of fear. a**, **b** Effects of bilateral injection of vehicle or oxytocin receptor antagonist OTA (21 ng/side/0.3 μl) in the CeA (central amygdala) 10–15 min (**a**) before exposure to "conditioned stimulus" CS1 in the presence of companion rat—on the acute (Day 2, SBF + OTA "Oxytocin antagonist", $n = 7$ or vehicle, $n = 5$, $F(3,20) = 14.12$; $p < 0.001$) and retention of SBF (Retention of SBF + OTA or vehicle, $F(5,30) = 14.79$, $p < 0.001$), and (**b**) before re-exposure to the CS1 and CS2 on Day 3—on the retention of SBF

(Retention of SBF + OTA, $n = 6$, or vehicle, $n = 5$, $F(5,27) = 7.732$; $p < 0.001$). **c** Effects of injection in CeA of vehicle ($n = 5$ animals) or oxytocin receptor agonist TGOT ($n = 7$ animals; 10–15 min before SBF, 7 ng/side/0.3 μl before exposure to CS1 on the immediate ($F(3,20) = 22.6$, $p = 0.0011$) and maintained reduction of freezing without companion (Retention, $F(5,30) = 0$; $p = 1$, n.s.). Two-way ANOVA (pre-tone, CS1, CS2) and group (["Companion + vehicle" or "+ OTA"]; ["No companion + vehicle" or "+ TGOT"]), for each session (Habituation, Fear recall, SBF, Retention of SBF), Bonferroni-corrected p values in the figures. Insets on the left indicate injection sites. Individual and mean values ± SEM are shown.

PVN is required on Day 2 both for the acute and maintained effects of SBF.

The PVN projects to many regions besides the CeA (Supplementary Tables 1 and ref. 18). To obtain a first impression of whether OT release across the brain could contribute to the acute and long-term effects of SBF, we infected PVN neurons with OTp-ChR2-mCherry virus, expressing channelrhodopsin-2 (ChR2) under the OT promoter in OT neurons only, and replaced SBF by blue light activation in the PVN. In previous work[18], we had shown that this viral construct selectively expresses ChR2 in PVN-OT neurons and their projections to, amongst others, the CeA and mPFC. Also, we had previously found that ChR2 is efficiently activated by blue light (BL) up to frequencies of

30 Hz, hence stimulating neurotransmission in PVN-CeA projections[18], which we further confirmed independently in the present study (Supplementary Fig. 5b). Three weeks after virus injection, in place of SBF, OTergic cells were stimulated with a blue laser light (BL, 473 nm) during the recall of CS1 (Fig. 3b).

BL stimulation acutely and significantly decreased freezing on Day 2 (Fig. 3b), whereas it had no effect in control rats infected with a hM4Di expressing AAV (Fig. 3b, orange bar & supplementary Fig. 4c). Thus, optogenetic activation of OTergic neurons in the PVN can acutely reduce freezing similar to SBF and TGOT injection in the CeA. However, unlike SBF, BL in the PVN on Day 2 did not decrease freezing on Day 3 (compare Fig. 3b with Fig. 1b). This is in line with the absence

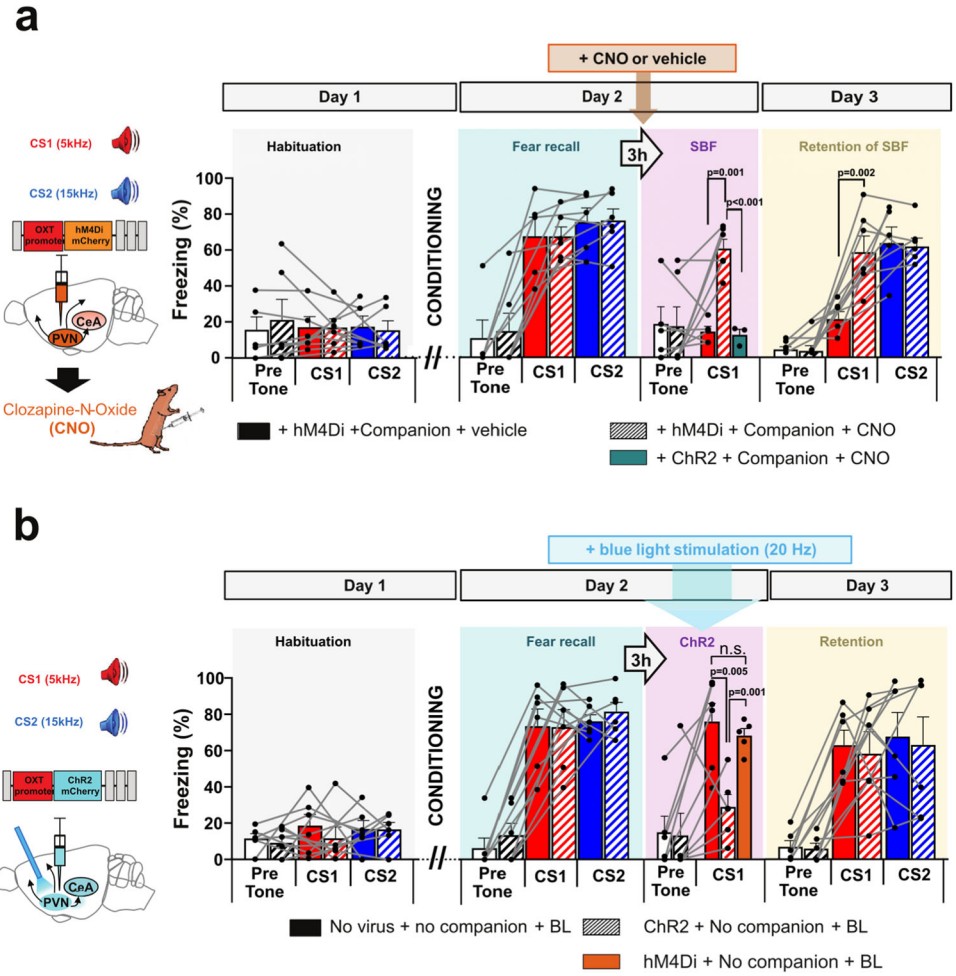

**Fig. 3 | Endogenous OT release from the PVN is required for the immediate and maintained-reducing effects of social buffering, and is sufficient for inducing immediate but not maintained reduction of fear. a** Disruption of acute SBF (SBF + CNO, F(3,18) = 8.107, p = 0.001, p = 0.001 Bonferroni-corrected) and retention of SBF (F(5,27) = 23.71, p < 0.0001, p = 0.002 Bonferroni-corrected) after inhibition of endogenous OT release from PVN by local inhibitory DREADD (hM4Di) expression under the OT promoter followed by ip injection of CNO, 30 min before SBF. CNO with unrelated construct (ChR2) showed normal SBF (two sides Student's t-test, t = 6.137, df = 7, p = 0.002) (Companion + hM4Di+ vehicle, n = 5; Companion + hM4DI + CNO, n = 6, green box: Companion + ChR2 under OT promoter + CNO, n = 3). **b** Acute (F(3,20) = 9.245, p < 0.001, p = 0.001 Bonferroni-corrected), but no

retention (F(5,30) = 7.925, p = 1) of reduction of freezing after optogenetic activation with blue light (BL) of OT PVN neurons infected with an AAV expressing ChR2 under the OT promoter without companion (No Companion + No virus + BL, n = 6; No companion + ChR2 + BL, n = 6, hM4Di + No companion + BL, n = 5). BL in animals with hM4Di did not show fear reduction (two -sided Student's t-test, t = 4.698, df = 9 vs ChR2+ no companion+BL, p = 0.001 and n.s., one sided Mann Whitney test, U = 10, p = 0.21). Technical details are given in the methods section. Insets on the left: Chemogenetic and optogenetic protocols, including the injected viral constructs. Individual and mean values ± SEM are shown. See also Supplementary Fig. 5. Insets on left show injection and implantation sites (CeA = central amygdala, PVN = paraventricular nucleus of hypothalamus).

of effects of TGOT infusion in the CeA on SBF retention (Fig. 2c) and confirms that OT alone in the CeA is not sufficient to recapitulate the fear-reducing effect of the presence of the companion in the SBF retention test. Putative OT release in other brain regions may likewise not be sufficient, although it should be stated that our BL protocol has, as mentioned, been validated specifically for PVN-CeA connections[18].

## SBF evokes changes of neuronal activity in PVN and CeA

To identify which neuronal changes in the CeA could underlie the SBF-induced acute effects on Day 2 and maintained effects on Day 3, we recorded single-unit spiking activity in vivo in the PVN and the CeA across the different behavioral sessions. The PVN was infected with the OTp-ChR2-mCherry virus (as in Fig. 3b) three weeks before implanting home-made microdrive-mounted tetrodes with optical fibers ("optrodes", Fig. 4a). We identified OTergic PVN neurons by their responsiveness to blue light ("BL", Fig. 4b), with waveforms similar to spontaneous spikes and short onset responses (<10 ms) with little jitter (2–3 ms, Fig. 4c). Most of the neurons could be reliably recorded

throughout the different behavioral sessions (Supplementary Fig. 6). Post-mortem immunohistochemical analyses showed specific expression of mCherry restricted to OT neurons (as before[18,28]), confirming the specificity of the OT promoter used in our viral constructs (Fig. 4d, Supplementary Fig. 5). Freezing levels throughout the protocol were comparable to previous recordings demonstrating no interference from these electrophysiological measurements (compare Fig. 4e with e.g., Fig. 1b).

In OTergic PVN neurons, baseline activity increased during Fear Recall (FR), and increased further to peak values during SBF (Fig. 5a, d). These findings are consistent with a role for OT release from PVN neurons during stress (induced by the FR) and during SBF, in line with known fear-reducing and prosocial effects of OT[30]. In the CeA we found neurons which, similar to the OTergic PVN neurons, showed increased baseline activity during FR that, upon introduction of the companion, increased further and peaked during SBF (Fig. 5b, d middle trace). The strong similarity between these responses in OTergic PVN and this group of CeA neurons suggested a direct connection between the two

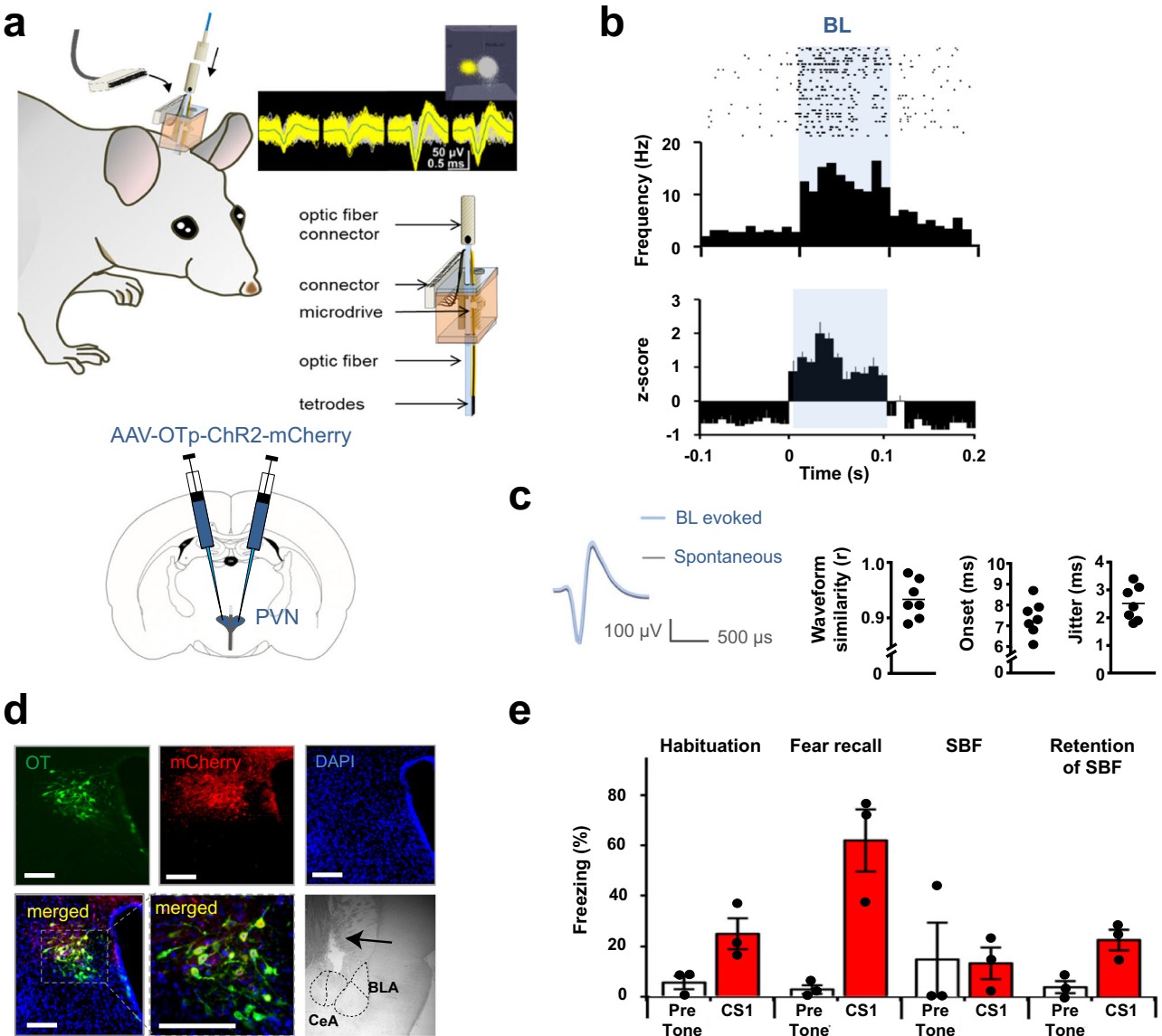

**Fig. 4 | In-vivo electrophysiological characterization of oxytocin (OT) neurons in the paraventricular nucleus (PVN). a** Top: Schematic of in-house made optetrodes on microdrives and their surgical implantation with representative example of a recorded raw neuronal waveform signal. Inset: single units plotted in 3-D principal component space, with one manually-defined cluster corresponding to an identified neuron (yellow) distinct from noise* (grey). Single units of an identified neuron have a clear refractory period of > 1.2 ms. In-house made optetrodes consisted of an optic fiber mounted on the fixed part of an in-house designed microdrive and tetrodes that were mounted on its movable part 200 μm away from the optic fiber. A copper screen was fitted around the implanted microdrive as a partial Faraday cage to reduce noise during the recordings. Bottom: Scheme for bilateral virus injections targeting PVN. **b** Top: Representative raster plot of spiking frequency of a PVN OTergic neuron responding to blue light (BL, 35 × stimulus duration, 100 ms). Bottom: Average z-score of all identified OTergic cells ($n = 7$). **c** Identification of OTergic neurons by 1) waveform similarity (r is Pearson correlation coefficient) between spontaneous and light-evoked activity, 2) time of "onset" and 3) "jitter" of responses to BL ($n = 7$, "blue light", "BL evoked"). **d** Example showing antibody stainings for OT (Oxytocin, green), mCherry (red), DAPI (blue) and merged (yellow, showing that all cells fluorescent for mCherry were also positive for OT) and in PVN (scale bars = 400 μm) and of location of tetrode implantation in CeA (central amygdala, arrow, BLA= basolateral amygdala) for one of the 3 rats, others not shown. **e** Freezing levels in response to CS1 across experimental sessions, as shown in Fig. 1a, in the presence of a companion. Individual and mean values ± SEM are shown. ($n = 3$ implanted rats). See also Supplementary Fig. 6.

as we had previously found in vitro[18]. Indeed, when we virally expressed ChR2 and inhibitory DREADD hM4Di specifically in OT-ergic PVN neurons that project to the CeA and implanted optetrodes one week after virus injection (Supplementary Fig. 7a1), we found that blue light (BL) stimulation directly excited a subpopulation of CeA neurons, and that this response could be inhibited by OTA and completely blocked by CNO (Supplementary Fig. 7a2, b).

Interestingly, in these in vivo recordings we also found CeA neurons that showed a response that, during SBF, was opposite to the OTergic PVN neurons. Thus, while baseline activity also increased following fear learning, it instead reversed to habituation baseline levels upon the introduction of a companion in the cage (Fig. 5c, d lower trace). Along the same line we found, when we stimulated OTergic PVN projections with BL during our in vivo recordings, CeA neurons that decreased firing rates as during SBF (Supplementary Fig. 7a3, c). These opposite responses of the two CeA neuron subpopulations are reminiscent of those described for the so-called CeL-Off and CeLOn neurons that are inhibited and excited, respectively, by

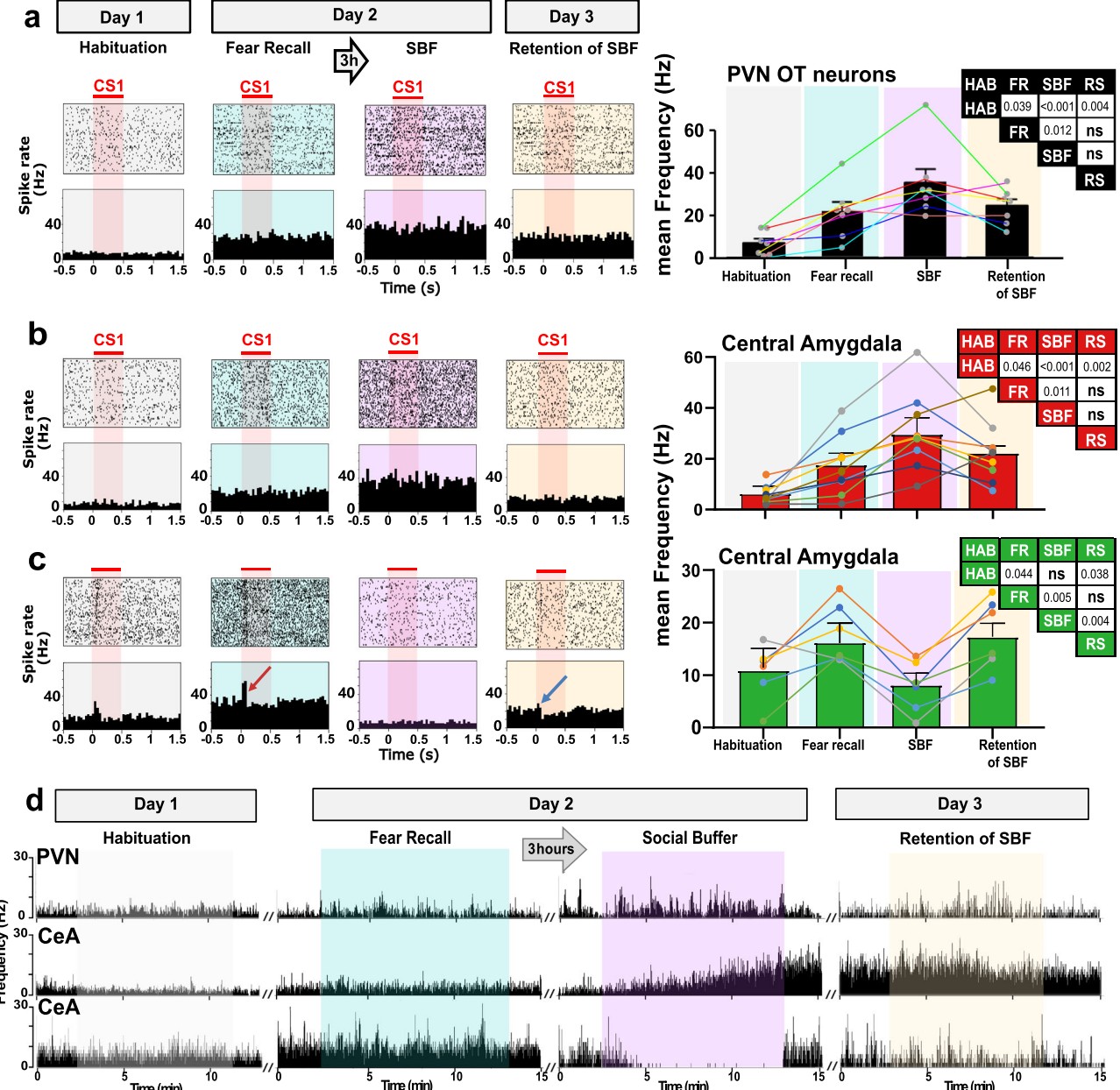

**Fig. 5 | Different development of neuronal spiking patterns in PVN and CeA across subsequent sessions of the SBF paradigm. a** Left: Raster (top) and frequency (bottom) plots of firing patterns of a representative PVN cell during different sessions of the behavioral protocol (indicated above the panels). One CS presentation of 30 s was discontinuous, consisting of 16 series of 500 ms blips presented at randomized intervals (1–1.9 s). The raster plot represents spiking from 0.5 s before until 1.5 s after each blip (aligning 64 blips from 4 CS × 16 blips/CS). Right: Averaged spike rates of 7 PVN cells across sessions (from 2 rats, each cell indicated by one dot and connected by colored lines), low rates during "Habituation" increased during "Fear recall", peaked during "SBF", and dropped during "Retention of SBF". **b** Left: As (**a**), but in CeA displayed similar spiking patterns as

PVN cells. **c** Left: As (**a**), cells that were potentiated by the CS during fear recall (red arrow), had highest baseline activity during fear recall and SBF retention, and no response to CS1 (blue arrow), with low baseline activity during SBF. Right: Averaged spike rates of CeA cells across sessions. **b** $n = 8$, **c** $n = 6$ cells/3 rats respectively). Individual cells are shown by dots and connected by colored lines. Tables indicate significance between sessions, ANOVA followed by Bonferroni correction, $p$-values as indicated in the insets, ns, not significant. HAB = Habituation; FR = Fear Recall; SBF = Social buffering of fear; RS = SBF Retention. Individual and mean values ± SEM are shown. See also supplementary Fig. 7. **d** Development of baseline changes of a PVN neuron, a CeA SBF neuron and a CeA fear neuron during the different sessions in the SBF protocol. Images represent one cell for each group, as depicted in a-c.

the presentation of a CS following fear learning, and that inhibit each other[31,32]. CeLOff neurons express OT receptors (OTR)[32], so it is possible that their excitation by PVN OT neurons could underlie the encoding of SBF in the CeA.

However, the baseline activity changes we observed do not provide causal proof for this concept. Indeed, baseline activity seems unlikely to encode for the SBF of freezing behavior that we have observed, since the reduction of freezing during and after SBF is

specific to the CS that is buffered (Figs. 1–3). Because SBF on Day 2 and SBF retention on Day 3 both appeared to depend on OT signaling (Fig. 2a, b), we hypothesized that retention of SBF reflects long-term plasticity changes in response to the CS that require both the instantaneous (on Day 2) and continued (on Day 3) activity of OTergic PVN projections[18,24]. We therefore assessed neuronal responses in the CeA to CS1 and CS2, by viral opto- and chemotagging specifically of PVN OT neurons to the CeA, as validated above (Supplementary Fig. S7).

Using optogenetic tagging, we could identify 137 neurons in the CeA, 44 of which were stimulated and 31 were inhibited by BL stimulation of fibers from OTergic PVN neurons. Within these recordings we found CeA neurons that increased their acute excitatory responses to the CS1 and CS2 after fear learning, as is expected from CeLOn neurons (Fig. 6d, e and supplementary Fig. 8b). Furthermore, in these neurons, pairing of the CS1 with SBF significantly decreased their acute responses on Day 2 and this reduction remained on Day 3 (Fig. 6d, g lower panel). The excitatory responses to CS2 (not paired with SBF) remained unchanged (Fig. 6e, g lower panel). The decreased response to CS1 required the presence of the companion, because in the absence of SBF on Day 2 (but in the presence of the extra CS1 exposure) it did not appear on Day 3: Extra exposure to the CS without concomitant SBF on Day 2 did not reduce freezing levels or single unit responses on Day 3 (Fig. 1b, c, supplementary Fig. 9). In fact, the single unit responses to CS1 and CS2 were highly correlated with the changes in freezing responses to CS1 and CS2 (Fig. 6g–i and supplementary Fig. 8b,c) throughout these sessions as also reported for CeLOn neurons during classical fear extinction[33]. Furthermore, most of these putative CeLOn neurons, when exposed to blue light, showed inhibitory responses, consistent with their inhibition through a local GABAergic circuit that is activated by BL-stimulation of OTergic PVN projections in the CeA (Fig. 6f). We further refer to these neurons as "fear neurons", as they seem to encode for the freezing responses when exposed to CS1.

We also identified a second group of CeA that showed no increased acute spiking responses to CS1 nor CS2 during fear recall on Day 2. Interestingly, these neurons started to exhibit only on Day 2 acute spiking responses to CS1, during the exposure to SBF (Fig. 6b, c and supplementary Fig. 8a). These acute CS1 responses could still be found on Day 3 ("retention" Fig. 6b, g upper panel and supplementary Fig. 8a). Their activation by BL indicates that these neurons are under excitatory control of PVN OT neurons, which is in line with the reported OTR expression in the majority of fear-reducing CeLOff cells[31,32]. As these neurons may thus not only encode cessation of freezing, which is at the base of the definition of CeLOff neurons, but also seem to actively encode the buffering of fear, we refer to these neurons as "buffer neurons".

As we had previously found that "Retention of SBF" on Day 3 remained dependent on OT signaling in the CeA (Fig. 2b), we tested whether these persistent acute responses to CS1 required activation of OTergic PVN projections. Indeed, chemogenetic inhibition of the OTergic PVN projections through administration of CNO significantly blocked acute spiking responses to CS1 in both CeA cell types (Day 3 vs CNO, $p < 0.01$, Fig. 6g, supplementary Fig. 8). Thus, in the fear neurons, the response to CS1 was restored, whereas that in the buffer neurons was suppressed. Taken together, these findings are consistent with a continued need of oxytocin signaling in the CeA for the retention of SBF.

Besides these two groups of neurons, we also identified other CeA neurons that did not show any response to BL, nor to CS1 or CS2. We did not examine these neurons further in this study.

## Discussion

In the present study we found that fear can be buffered robustly by the presence of a companion. This social buffering of fear (SBF) is immediate, in contrast to fear extinction, and is retained at least the next day, indicating SBF learning and memory. SBF is encoded by two groups of CeA neurons that are under excitatory and inhibitory control of the neuropeptide oxytocin from the hypothalamic PVN. The group of neurons that is excited by PVN OT neurons resembles the earlier identified CeLOff neurons[31,32], based on their reduced or absence of response to a conditioned stimulus. However, we found that these neurons are activated by the CS after SBF, and can thus encode stimuli with positive valence. Their responses to environmental stimuli seem

thus to be more dynamic than previously thought, and we therefore refer to them as "buffer neurons", to reflect their enhanced CS responses in our SBF paradigm. Similarly, the cell population that is inhibited by oxytocin (released by BL stimulation), the CeLOn neurons, may indeed encode fear, and the responses to the CS are dampened by SBF. We refer to these neurons as "fear neurons". SBF may thus induce a switch from fear to safety encoding by evoking opposite responses to a CS in fear and buffer neurons under the influence of oxytocin from the PVN.

We found that our SBF protocol produced a very robust reduction of freezing behavior. Thus, in rats matched in age and weight, SBF occurred equally well in male-male and female-female dyads and in familiar and unfamiliar rats. Similar SBF in female dyads was not necessarily expected, given the lower levels of social interaction in female rats[34] and gender divergence of anxiety disorders in humans[35]. Levels of SBF were not affected by the single housing protocol as compared to group-housing (compare Fig. 1b vs. supplementary Fig. 2a), possibly because single-house stress[36] was relatively short and in an enriched environment (see M&M). Levels of fear conditioning and SBF were comparable for 5 and 15 kHz which deviates from reported differences in auditory frequency responses in mice[37], possibly as a result of the broader frequency hearing range in rats[38]. The precise sensory modality that underlies SBF remains to be determined, but from our experimental setup we can conclude that it is not through direct physical contact, because the animals were separated by a perforated plexiglass barrier. Also, it does not seem to depend on active attention to the conspecific as we did not find any significant correlation between social interaction and reduction of fear (Supplementary Fig. 3d, black dots). On the other hand, after blocking with OTA we found a correlation between freezing and social behavior (Supplementary Fig. 3d, green dots), but we think this is indirectly caused by the increase of freezing caused by OTA (forcibly decreasing social interaction), as OTA applied during the habituation period (Supplementary Fig. 3a) did not directly affect social interaction, nor social motivation.

Although we did find robust SBF in all our experiments (Figs. 1–3, supplementary Figs. 1–4) independent of CS frequency, sex, grouped housing between experiments or familiarity with between rats), absolute freezing levels during "Fear Recall" or "SBF" could vary between animals. This might be related to individual differences in endogenous OT signaling: Thus, we found similar increased baseline activity of OTergic PVN neurons and a population of CeA neurons already during "Fear Recall", suggesting that endogenous OT can internally buffer fear through a homeostatic mechanism (Fig. 5a, b). Introduction of the companion gradually increased baseline activity further (Fig. 5d), precluding a homeostatic increase. Interestingly, the same individual PVN and CeA (buffer) neurons that increased baseline activity during fear recall also increased their activity further during SBF (Fig. 5a, b), indicating an encoding of endogenous and social buffering of fear by the same cells. Variability between individuals might thus arise through differences in endogenous OT signaling. Such basic differences may lead to different sensitivities to SBF and thus, could constitute an animal model equivalent of how in humans individual differences in attachment styles could affect sensitivity to social support[39].

### Oxytocinergic connections from PVN to CeA

The combined use of CAV2 in the CeA and AAV-based constructs under the OT promoter in the PVN has enabled us to assess single unit electrophysiological responses throughout the sessions of the SBF protocol that are mediated by the PVN-CeA subcortical pathway. The observed direct connection between PVN OT neurons and CeA buffer neurons was further indicated by similar baseline activity changes in both cell types, and the excitatory responses of buffer neurons to blue light stimulation of axon terminals from OT-ergic PVN neurons. Our

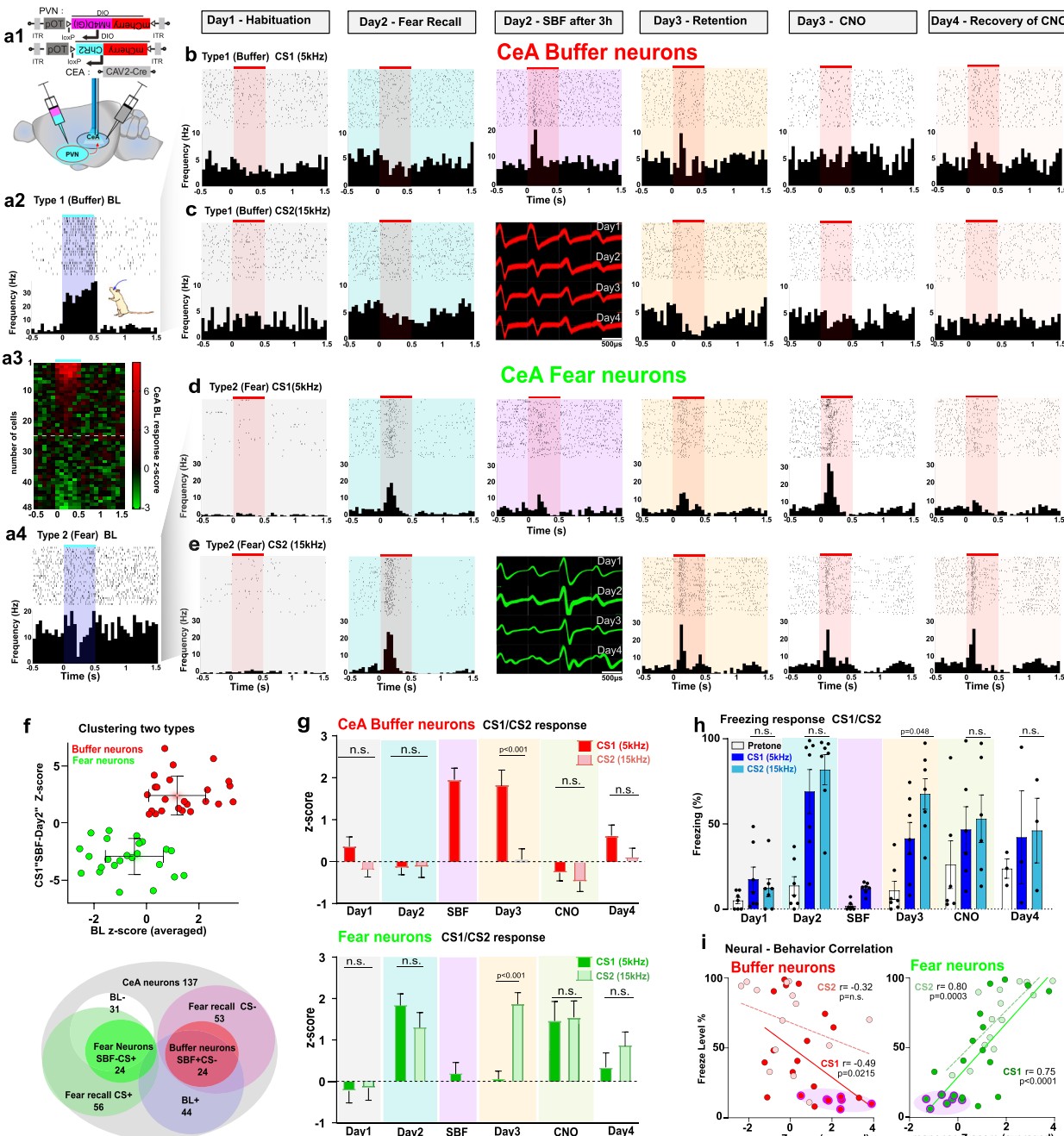

**Fig. 6 | Neuronal representation of fear and social buffering in the CeA, defined by responses to CS. a1** AAV expressing double-floxed (DIO) mCherry-ChR2 under the OT promoter (pOT) injection in the PVN (paraventricular nucleus of the hypothalamus), and CAV2 expressing CRE in the CeA (central amygdala) to express ChR2 specifically in oxytocinergic PVN neurons projecting to the CeA where optetrodes were implanted to record neuronal responses to CS1, CS2 and blue light (BL). **a2**–**a4** Firing responses to BL (0.5 s, 30 Hz, blue shade) differ between "Buffer" and "Fear" neurons, as shown in Peri-stimuli raster (top) and frequency (bottom) plots, and heat map ($n = 48$; color bar; z-score of optogenetic response in 0.1 s time bins). **b**–**e** Raster and frequency plot examples of neuronal firing to CS1 and CS2 (0.5 s, 5 Hz or 15 kHz, red shade) across sessions with single unit spike waveforms (shown in Day 2/SBF column) recorded across 4 days. **b**, **c** Persistent excitability increase to CS after SBF in a "Buffer" neuron. **d**, **e** Increased excitatory responses to CS1 and CS2 following fear learning in a "Fear" neuron. SBF reduces excitability which remains during retention. **f** Top: Neuron ($n = 48$) separation according to changes in CS1 ("SBF-Day2") and BL responses into CeA "Buffer " (red) and "Fear"

(green) neurons (means±S.D). Bottom: Venn diagram showing total number of recorded neurons recruited after FC ("Fear recall CS + " $n = 56$), inhibited by BL ("BL-", $n = 31$) and subsequently inhibited by SBF ("Fear Neurons SBF-CS + ", $n = 24$) or activated by BL ("BL + ", $n = 44$) and recruited after fear conditioning by SBF ("Buffer neurons SBF + CS-", $n = 24$). **g** Increased (Buffer neurons, $n = 24$) or reduced (Fear neurons, $n = 24$) responses specifically to CS1 during SBF and SBF retention (Day 3) when compared to Fear recall (Day 2; $p < 0.001$ for all; see Supplementary Fig. 8a1, 8b1 for details), and to CS1 vs CS2 during SBF retention (two-tailed paired Student's t-tests; buffer: $t = 4.87$; df = 23; fear: $t = 6.080$, df = 24, both Bonferroni corrected $p$-values in the figure, means ± sem. **h** Freezing to CS1 is significantly reduced when compared to CS2 following SBF (Day 3, One-way ANOVA $F(2,18) = 12.19$, $p < 0.001$, Bonferroni corrected value in the figure, n = 7 animals, means ± sem). **i** Negative correlation between freezing and CS responses except for CS2 in buffer neurons (Pearson correlation). Shaded areas indicate SBF on Day 2; Day 1 (habituation) not included.

previous work with a virus expressing mCherry and ChR2 under the OT promoter had revealed OTergic projections from magnocellular PVN and SON neurons to the CeA and we had found that their optogenetic activation in the CeA increased activity of CeA neurons in vitro in an OT-dependent manner and inhibited fear-conditioned freezing in vivo[18]. Our present single unit recordings allowed us to also identify differential CeA neuronal responses in vivo that are likely mediated by local OT release, although we cannot exclude some co-release of glutamate (Supplementary Fig. 7) in agreement with our previous in vitro findings[18].

### Opposite responses of fear and buffer neurons in the CeA network

Through their changes in baseline activity during FR and SBF, their responses to BL and their distinct acute responses to CS1 and CS2, we could distinguish three categories of neurons in the CeA: One series of neurons which showed no significant responses to our protocols and which we did not study further. A second group which showed enhanced acute CS1 responses during FR, and with a decreased baseline activity and decreased CS1 responses during SBF as compared to FR. These were often inhibited by BL. These cells seem to correspond to the so-called "CeLOn" neurons that have been identified by the Luthi and Anderson groups in 2010[31,32] as they directly trigger freezing responses through their projections to the brainstem[31,40]. We referred to these neurons as "fear neurons".

A third population of neurons showed during the SBF protocol opposite responses to the ones displayed by the fear neurons. Thus, they increased baseline activity during SBF, were directly excited by BL and showed enhanced CS1 responses during and following SBF. They may correspond to the previously reported protein kinase C (PKC) δ-positive "CeLOff" neurons[31,32]. We named these cells "buffer neurons" to do justice to their apparent coding of safety as revealed in our SBF protocol. PCR analyses had shown that CeLOff (now buffer) neurons often express mRNA for the OT receptor[32]. In vitro, we have previously reported that OT-sensitive neurons in the CeA make local inhibitory GABAergic connections and, by their effects on projections to the brain stem, can inhibit freezing[24,25]. The acute effects of SBF could thus be explained by a neuronal mechanism in which the presence of the companion triggers OT release in the CeA and excites buffer neurons, whose local inhibitory connections with fear neurons would inhibit the freezing response.

What is unexpected in our acute response findings, is that these buffer (CeLOff) neurons can, after pairing CS1 with the SBF, become directly activated by the CS1. This goes together with a decreased response of fear (CeLOn) neurons to CS1. It is currently unclear what causes the CS1 to shift after SBF to activate buffer instead of fear neurons. It may occur through a Hebbian mechanism in which an SBF-triggered OT release postsynaptically activates buffer neurons concomitant with a presynaptic activation of a (hitherto silent) input from the CS1. This pairing would increase the synaptic strength of the CS1 input onto these buffer neurons and lead to an "encoding of safety" by the CS1 (similar as reported in mice, see ref. 41). We have previously described such a potentiation of synaptic input onto OT-sensitive synapses originating from the BLA in rats that had learned to suppress their freezing response in the face of imminent danger (a CS-signaled electric footshock)[42]. The proposed changes in synaptic plasticity would require the endogenous release of oxytocin by social contact, as has been observed in the auditory cortex[43].

Although synaptic plasticity may underlie this shift, a maintained decrease in CS1-freezing response continues to require OT signaling after SBF (Fig. 6), even if it is not accompanied by a maintained high elevation of buffer neuron basal activity (Fig. 5). This suggests that endogenous OT on Day 3 is still needed to trigger sufficient action potentials in buffer neurons for inhibition of fear neurons. The recently published OT biosensor that we validated for measuring endogenous

OT release in the PVN may proof a useful tool to monitor the corresponding dynamics of OT release in real-time in the CeA[44].

### OT signaling is necessary but not sufficient for sustained effects of SBF

Contrary to a recent study by Gorkiewicz et al.[45], we found a sustained effect of SBF on freezing upon next-day re-exposure to the CS in the absence of the companion. This is possibly related to our lower foot shock intensity (0.5 mA) as compared the one they used (0.7 mA)[45]. Indeed, Mikami et al.[12,46] used 0.55 mA and also found a sustained SBF effect. It is possible that higher intensity fear-inducing stimuli surpass the endogenous buffering by OT, as well as the calming effects of OT released during SBF.

Whereas our behavioural and electrophysiological observations clearly demonstrate the necessity of OT during SBF, pairing of CS1 exposure with locally-infused exogenous OT (applied as the specific OTR agonist TGOT), or optogenetically-released endogenous OT in the CeA, failed to recapitulate the sustained SBF effects. Clearly, the presence of a companion on Day 2 is also needed to induce a retention of SBF the next day without the partner. This suggests the need for an observational learning component and associated signaling in the brain. Indeed, the subcortical PVN-to-CeA pathway that we identified may not operate alone to induce a retention of SBF. Recently, Yu et al.[47] reported that the CeL can control plasticity (upstream) in the BLA during fear conditioning and Fuzzo et al.[48]. reported decreased CS responsiveness in the BLA during social buffering. Possibly such CeA feedback to the BLA[47], where it might converge with mPFC projections, is required for retention of SBF. To induce retention of SBF, the presence of the partner may lead to an "internalized representation" of its social buffering effects.

Recent publications have indeed shown that the mPFC can store memory of conspecific actions[49] and similarly, that the medial amygdala and lateral septum (LS) can strengthen social memory regardless of its valence[23,50–52]. Even the OT neurons in the PVN itself have recently been shown to store a memory of a CS/US association[30]. Furthermore, OT released from the PVN in the CeA seems necessary for emotion discrimination in mice[53]. OT signaling from the PVN onto the mPFC, LS, CeA, and possibly other regions such as the paraventricular thalamic nucleus that project to the CeA to control fear-related behavior[54] could be involved. These wide-spread OT projections may constitute a highly-coordinated peptidergic signaling network that affects the representation of the emotional state of oneself and others to modulate the expression of fear-related behavior, both acutely and chronically. Enhanced oxytocin signaling to all these areas may be implicated in the memory association of the tone with the social encounter and emotional state of the conspecific and for the memory effects of SBF.

### SBF effects are different from fear extinction

Previous studies have implicated the mPFC in the inhibition of freezing during fear extinction[10,12,55,56] and have suggested a role of OT in the mPFC in facilitating extinction in the presence of a conspecific[10]. We nevertheless believe the maintained reduction of fear that we found is different from classical fear extinction. First, contrary to the gradual (mPFC, OT-modulated) reported extinction[10], SBF-induced reduction of freezing is immediate and total during SBF (Supplementary Fig. 1). Second, our SBF to CS1 was insensitive to change in context throughout all experiments (Figs. 1–3 & 6, Supplementary Figs. 1,2 & 4). This is contrary to the SBF-enhanced extinction as reported by Mikami[12], that undergoes "renewal" upon change of context. The latter is typical of classical fear extinction that is caused by active inhibition. Furthermore, Whittle et al. recently reported that the increased responses to a CS by CeLOn neurons after fear learning requires 16 additional consecutive CS presentations for extinction to occur[33]. It is therefore unlikely that the four extra CS1 presentations during SBF

have led to extinction (see also Fig. 1b, c and Supplementary Fig. 9). It thus remains to be examined to what extent extinction mechanisms overlap with the neuronal encoding mechanism in the CeA that underlie the next-day maintained effects in the absence of the companion.

In conclusion, SBF in rats recruits an OT-ergic PVN-to-CeA subcortical pathway that controls fear-related behavior. However, for permanent fear-reducing effects of SBF, this pathway likely operates in concert with other brain regions that are activated during the presence of the companion. The buffer neurons (probably corresponding to the formerly identified CeLOff neurons) in the lateral part of the CeA (CeL)[31-33] have emerged as central elements for successful SBF: Their enhanced activity during SBF, their excitatory response to endogenous OT and their reported inhibition of the fear neurons puts them at the heart of reduction of fear by social support. Also, the continued dependence of SBF on OT may explain why some people are more sensitive to social support than others, depending on their attachment style that is defined by endogenous OT levels[39]. This clinical observation, combined with the immediate reduction of fear that contrasts with classical fear extinction, makes SBF a promising paradigm for the development of future fear-reducing therapies.

## Methods

### Ethics statement

All studies were approved by the Swiss Veterinary Office of the Canton of Vaud. Across the different sessions and 3-5 days before starting the behavioral protocol, the animals were individually housed to use the same housing conditions for implanted rats (as these need to be single-housed to avoid injuries by cage mates). Individually-housed animals were kept in an enriched environment that included wood shelter, straw for nesting and hollow wood toy cubes to play and nibble following precise legal requirements of the veterinary office.

### Animals and husbandry

Male and female Sprague Dawley rats were obtained from Janvier Labs, Genest-Saint-Isle, France or from our animal facility and maintained at $20 \pm 2\,°C$ and 40–50% humidity under a 12/12 dark–light cycle (7 am/7 pm) with ad libitum access to food and water and tested at 8–10 weeks old (300–400 g).

### Fear conditioning and Social buffering protocol

Prior to the start of the behavioral protocol rats were handled for one week each day during 20 min by the same experimenter who would subsequently conduct the complete behavioral procedure. All sessions took place in different contexts as schematically shown by the different shaped and colored boxes in Fig. 1b with dimensions as specified below. Boxes were cleaned in between sessions with a 2% cleaning soap (Deconex, Instrument Plus, Borer Chemie AG, Zuchwil, Switzerland) and at the end of the day with 70% ethanol. Ethanol was avoided in between sessions in order to avoid irritation to the eyes of the rats. In between sessions, animals were returned to their home cage, in which they were single housed, but environmentally enriched as stated above.

Conditioned stimuli consisted of 5 kHz and 15 kHz tones (CS1 and CS2) that both lasted 30 sec continuously, except for the electrophysiological recordings where they consisted of a succession of 16 blips of 500 ms given at randomized intervals (1–1.9 seconds). For all sessions, CS exposures were given at intervals varying between 40 and 120 seconds, and presented four times each, except for fear conditioning when they were presented eight times. CS1 and CS2 were presented alternatingly on day 1 (habituation and fear conditioning) and continuous for the testing of freezing responses on subsequent days (4x CS1 then 4xCS2, or inverse)

Fear conditioning was conducted in a chamber from Med Associates, Inc, Fairfax, VT, USA with dimension $30 \times 25 \times 25$ cm in which the rat was placed Day 1, during 10–15 min before fear conditioning, during which, for the electrophysiological experiments, the quality of the recording signal was tested and subsequently the habituation responses to the CS1 and CS2 rats were acquired, after which rats were exposed to the tones which were co-terminated with a mild electric foot-shock (0.5 mA, 2 s duration). On Day 2 (24 h later) fear recall occurred in a new context (EU 3H rat cage, top $42.5 \times 26.5$ cm, bottom $37.5 \times 21.5$ cm, height 18 cm). Three hours later, social buffering was initiated in a two-chamber PVC cage (dimensions $53 \times 40 \times 33$), separated by a plexiglass wall with $7 \times 7$ holes of 0.5 cm diameter. The experimental rat spent alone 10–15 min in the compartment (time, for to test 5–8 min quality of recording for electrophysiological experiments). Then the companion rat (or a polystyrene ball as a control for novelty) was introduced and the CS1 presented over 10–15 min (maximally, to keep an interest for social interaction). Companions were always from different cages, of the same sex, age/weight, unfamiliar and never exposed twice, never exposed to the US or the CS prior to the SBF. Finally, on Day 3, the experimental animals were exposed to CS1 and CS2 in a new context (hexagonal cage of carton boards 29 cm height and 40 cm diagonal). Exposure to CNO and recovery (Day 4) were performed in home cages. All boxes allowed for simultaneous behavioral assessments, optogenetic stimulation and electrophysiological recordings. For practical reasons, maximally 8 rats were tested per week for the pure behavioral experiments and 2 for the electrophysiological experiments.

To assess the efficiency of SBF by a familiar or unfamiliar conspecific, animals were fear conditioned and tested in the same context the next day in the presence of a brother or a stranger. The fear conditioning and the testing protocols have been described earlier[25].

### Behavioral analyses

Animal behavior was recorded by a video camera (Microsoft lifecam HD 3000, Redmond, WA, USA) placed above the cage, operated with Showbiz acquisition software (Showbiz™, ArcSoft, Inc., CA, USA). Animal's behavior was encoded offline by experimenter blind to the protocol using an ethologic keyboard connected to homemade MatLab software (MathWorks™, Natick, MA, USA) for "exploration", "social interaction" (nose to nose or nose to body through the plexiglass wall), "social motivation" (close to the plexiglass wall), "grooming" (licking of entire body), and "freezing", which was defined as crouching posture and absence of any visible movements except breathing[57]. Freezing behavior was measured from the beginning of the CS until the next CS or (for the pre-tone interval) during 5 min before tone delivery and converted to a percentage of time (of each trial).

### Stereotaxic surgeries

For all procedures, anesthesia was induced in an induction chamber with $O_2$/isoflurane (95%/5%) and maintained at $O_2$/isoflurane (97.5%/ 2.5%) in a stereotaxic frame (Kopf Instruments, Tujunga, CA, USA) through a facemask. Body temperature was maintained with a heating pad (Solis S8-S, Zurich, Switzerland).

**Cannulae implantations.** Rats were implanted with two stainless steel guide cannulae ($23\,G \times 12$ mm, Phymep, France) bilaterally with tips ending 1.5 mm above the CeA (antero-posterior, −2.5 mm relative to Bregma; lateral, ± 4.5 mm from midline; ventral, −6.4 mm from dura). The cannulae were fixed to the skull with dental acrylic cement and anchored with a surgical screw placed in the skull. Stylets were inserted into the guide cannulae to prevent clogging. The animals were allowed two weeks of post-surgical recovery, during which they were regularly handled.

**Virus injections.** Virus injections (bilateral, 0.3 microliters each side) were stereotaxically targeted to PVN (antero-posterior, −1.8 mm

relative to Bregma; lateral, ±0.4 mm from midline; ventral, 7.4 mm from dura) and CeA (antero-posterior, −2.5 relative to Bregma; lateral, ±4.5 mm from midline; ventral, −7.5 mm from dura. two weeks prior to the behavioral experiments[58].

**Chemogenetics.** AAV expressing hM4Di and mCherry under the oxytocin promoter ([OTp-hM4D(Gi)-mCherry], were generously provided by Valery Grinevich, Mannheim, Germany). Two weeks after infusion, Clozapine-N-oxide (CNO) was injected intraperitoneally (IP) 30 min before SBF on Day 2 of the experimental paradigm[59].

**Optogenetics.** An AAV1 encoding ChR2 and mCherry, both under the oxytocin promoter ([OTp-ChR2-mCherry], was generously provided by Valery Grinevich)[18].

### Optic fiber and tetrode implantations
Optic fibers (200 μm, 0.39 NA; Thorlabs GmbH, Munich, Germany) or optetrodes were implanted bilaterally in the PVN at an angle of 23° to prevent steric hindrance (antero-posterior, −1.8 mm relative to Bregma; lateral, ± 3.4 mm from midline; ventral, −8.6 mm from dura), and in the CeA (antero-posterior, −2.5 mm relative to Bregma; lateral, ±4.5 mm from midline; ventral, −7.5 mm from dura) and fixed to the skull with dental acrylic cement[58].

After surgery, Iodine was used for disinfection and analgesic cream (Bepanthen) before and during the 3 days as well as Dafalgan (500 mg in 500 ml) in the drinking water. During recovery (7 days) implanted rats were single-housed (with double wood meshes) with daily observation.

The optic fiber was connected through a coupler (custom made with plastic, Fig. 4a, coupling efficiency of 87−95%) to a DPSS blue light laser (MBL-473/50 cmW; Laserglow, Canada) with final output of 6-8 mW per side to deliver train pulses of 2 ms at 20 Hz, 10 s during the CS.

After the experiments, cannulae, optical fibers and optetrodes were removed and animals were perfused for histological analysis of the implantation sites (see below).

### Drugs for pharmacological and chemogenetic modulation
**Pharmacology.** The specific OTR agonist (Thr4, Gly7)-Oxytocin (TGOT, 7 ng/side) and OTR antagonist (d(CH$_2$)$_5$,$^1$,Tyr(Me)$^2$,Thr4,Orn8,des-Gly-NH$_2$9)-Vasotocin (OTA, 21 ng/side) (Bachem, Switzerland) in 0.3 μl saline were infused bilaterally over 1 min through canulae targeting the CeA 10−15 min before SBF[25]. For in vitro electrophysiology, OTA was used at 1 microM and 6-cyano-7-nitroquinoxaline-2,3-dione (CNQX, Tocris) at 1 microM.

**Chemogenetics.** Clozapine-N-Oxyde (CNO, Tocris, UK) was intraperitoneally injected at 3 mg/kg in 1 ml saline 30 min before placing the rat in the box for behavioral assessments.

### Ex vivo electrophysiological recordings in brain slices
Four-to-eight weeks after combined viral injections into the PVN and CeA, brains were removed, cut into 400 mm horizontal slices, and kept in artificial cerebrospinal fluid in the dark to avoid ChR2 activation. Whole-cell patch-clamp recordings were visually guided by infrared videomicroscopy (DM-LFS; Leica), using 4−9 MOhm borosilicate pipettes filled with 140 mM KCl, 10 mM HEPES, 2 mM MgCl$_2$, 0.1 mM CaCl$_2$, 0.1 mM BAPTA, 2 mM ATP Na salt, 0.3 mM GTP Na salt (pH 7.3), 300 mOsm, and amplified with an Axopatch 200B (Axon Instruments, Molec. Devices, USA). ChR2-mCherry expression was identified by fluorescent microscopy and post hoc immunohistochemistry. For in vitro blue light stimulation experiments, optical stimulation was done via a mercury lamp (Short Arc 103 W/2, Osram; around 5 mW/mm$^2$) in combination with a shutter (VS25S22M1R1, Uniblitz) or a

TTL-pulsed LED source (LXHL-LB3C, Roithner; around 10 mW/mm$^2$), both yielding similar results. For further details, see ref. 18.

### In vivo electrophysiological recordings in freely moving rats
Multi-wired electrodes and optetrodes were employed to record single-unit neuronal activity (see Fig. 4). Tetrodes consisted of 8 × 4 twisted nickel-chrome wires (0.25 μm; California Fine Wire, Grover Beach, CA, USA) each plated with gold solution (Gold Non-Cyanide 32 gr/L, SIFCO, France) down to an impedance below 0.2−0.4 MΩ and attached to the nut of a copper screw with a 270 μm step per full turn. Four fixation points surrounding the microdrive were created each harboring a small bone screw for fixation, two of which were attached to a ground wire. Dental cement was used to secure the screws to the skull. Electrodes were positioned just 0.5 mm above the PVN or CeA, and gradually lowered to obtain high-quality recordings. The space between the electrodes and the skull was filled with softened paraffin. An additional layer of dental cement firmly attached the microdrive to the skull.

Optetrodes were each day lowered 50−70 μm until reaching target and delivering a good spike signal (see also opto-tagging and CS-response below). Optetrodes were connected to a head-stage containing 32 unity-gain operational pre-amplifiers (1000x gain, bandpass-filter 400−7 kHz, Plexon, TX, USA). Neuronal activity was digitized at 40 kHz, bandpass filtered from 250 to 8 kHz, and sampled for 1400 microseconds to capture whole spike events in the PVN (OT neurons have wider waveforms) and 800 microseconds in the CeA (for interneurons), each included a 200 microseconds pre-spike waveform (See supplementary Fig. 6c, upper panel). During each different session of behavioral testing, the spike waveforms and their associated time stamps were saved in data files using Plexon system format (around 100−200 MB per 20 min section).

### Spike sorting and quality control
To keep data analysis robust, we first merged multiple raw data (PLX files) from single sections into one mega file per rat (by PLexUtil, Plexon USA). During offline spike sorting (Offline sorter, 4.0 Plexon, USA) we first automatically removed outliers (by calculating the Mahalanobis distance between the points and the centroid, using a distance greater than the Outlier Threshold 2.0) and shorter ISI (refractory period <1200 microseconds within 0.1% error tolerance) waveforms. Well-sorted clusters (see supplementary Fig. 6) were then stored as Neuroexplorer format (Nex5) and Plexon format (PLX) for further analysis. Waveform tracking, autocorrelation, and PCA features were used to properly transmit the manual inspection of each units. Waveforms should be constant over all recordings sections, auto-correlograms should have refractory periods exceeding 1−2 ms, and PCA features for each unit should keep the same cluster in feature projections across all sections. In addition, the correspondence between two same units was checked day by day using the Mahalanobis distance (MatLab, MathWorks™, USA).

For the overall 147 units sorted in 8 rats, on average 12−25 units per rat were identified with the 32 channel electrodes. If units changed waveform shape or disappeared during later sections, we considered it a lost unit (Supplementary Fig. 6f3, 6h). We also plotted a scatter graph to separate the interneurons from excitatory neurons according to their Full Width at Half Maximum (FWHM) and mean firing rate (Supplementary Fig 6g). Only putative interneurons (as CeA's major neural type) were adopted in later analysis.

### Optogenetically guided neural type identification (Opto-tagging)
To electrophysiologically identify ChR2 expressing neurons under the OT promoter in the PVN (Fig. 4), local blue light (BL) was given with durations of 100 ms, 35 times repeated at 2 s intervals[58]. Delay was

measured between the onset of the blue light stimulation and the peak of the first evoked action potential, and jitter as the inter-trial standard deviation between the measured delays, waveform similarity was assessed in MatLab using the correlation function, giving the Pearson correlation coefficient (PCC) as plotted in Fig. 4c (waveforms were considered similar for PCC > 0.85 and p-value < 0.05).

Moreover, we used waveform similarity analysis to compare waveforms from spontaneous activity and spikes triggered by light induced for optogenetical identification of OTergic neurons. Blue light stimulation in the PVN (Figs. 3b and 4a–c) and CeA (Figs 6a2, a4) was given at a frequency of 20 Hz (2 ms ON and 48 ms OFF) for 10 s during the CS starting 5 s before the CS to locally activate cell bodies, respectively OTergic projections and induce OT release.

## CS presentations and responses
For each sorted unit, responses to the CS were plotted in peri-stimulus time histograms (PTSH) starting 0.5 s before until 1.5 s after the CS blip with 64 trials per graph (Figs. 5, 6 and Supplementary Fig. 7). Firing rate frequency (FRF) was binned at 50 ms and for each bin the average baseline FRF (preceding tone presentation by 0.1–0.5 s) was subtracted and the resulting value divided by the standard deviation (SD) of the baseline FRF, yielding a corresponding z-score bin. If the baseline firing rate SD was equal to 0, a corresponding value was calculated over the entire recording instead.

## Post-mortem brain analyses
At the end of the electrophysiological experiments, recording sites were marked with electrolytic lesions. Rats were deeply anesthetized with a lethal dose of sodium pentobarbital (450 mg/kg intraperitoneal) and perfused transcardially with 250 ml of 0.9% saline followed by 250 ml of 4% paraformaldehyde (PFA) in 0.1 M phosphate buffer (pH 7.35, PB). Brains were transferred to 30% sucrose in 0.1 M PB at 4 °C for 48 h, then frozen and sliced (40 μm) coronally. Verification of cannula, optical fiber and optetrode implantation tracks in the PVN and CeA was performed by visual inspection and documented with a light microscope.

## Immunohistochemistry
Unspecific binding sites were blocked in phosphate buffered saline (PBS, pH 7.4) containing 10% bovine serum albumin (BSA) and 0.3% Triton X-100. Next, a mix of an mCherry-DsRed rabbit polyclonal antibody (Living Colors-Clontech Antibody 632496, 1:1000) and a mouse monoclonal antibody against oxytocin (P38, 1:400, generous gift of Dr.H. Gainer) was applied in the same solution, but now containing only 3% BSA, overnight at 4 °C. Slices were then washed three times for 15 min in PBS (room temperature), and then incubated in a goat anti-rabbit 1:1000 Alexa 568-conjugated secondary antibody (Life Technologies) for 2 h at room temperature. After three 15 min washes in PBS at room temperature, a goat anti-mouse Alexa-488 conjugated antibody was applied (1:1000) for 2 h at room temperature (Life Technologies). Finally, the sections were washed twice with PBS and once in PB, and mounted with DAPI containing anti-fading mounting medium (Vectashield, Vector Laboratories). Images were obtained with a confocal laser-scanning microscope equipped with a Fluoview 300 system (Olympus), a 488-nm argon laser an 537 nm helium-neon laser, or with a Leica SP5 confocal microscope using additional 350-nm laser with a 203/0.7 NA oil immersion or 403/0.8 NA water immersion objectives. Colocalization was determined by overlap of the ROI obtained from the two independent fluorescence signals. Analysis was performed in at least three sections per animal.

## RNAscope
In order to assess OTR expression in the CeA, we employed the RNAscope™ in situ hybridization technique using the Probe- Rn-Oxtr (Advanced Cell Diagnosis, ACD, Catalog Number: 483671) designed specifically for rat OTR mRNA. The target region spanned from base pairs 124–1155. Following tissue fixation and permeabilization, the RNAscope probe was applied to the tissue sections. The hybridized probes were then amplified and visualized using a 2.5 HD Assay signal amplification system. Signal localization was assessed, allowing for the specific detection and localization of individual mRNA molecules within the tissue context.

## Statistical analysis
For the behavioral analyses, we made multiple comparisons by formulating our hypotheses in a set of contrasts that were tested in a two-way ANOVA, and we adjusted p-values based on Bonferroni correction. In our testing, we assessed whether CS1 and/or CS2 induced significant freezing when compared to pre-tone freezing levels, and simultaneously whether SBF, or pharmacological, optical, or chemogenetical interventions influenced freezing behavior. Two-way ANOVA was used for each session (Habituation, FR, SBF, Retention of SBF), with pre-tone, CS1, CS2 as dependent factors and groups as independent factor (for example: companion / no companion or Vehicle / OTA, etc.). Power analysis (calculated with G*power 3.1.9.4., Franz Faul, Univ. Kiel, Germany)—was based on the effect size (calculated as Cohen's d effect sizes) of our first observations (in Fig. 1b, c), and with this we designed subsequent experiments with large enough sample sizes to acquire the power above 0.8 to test our hypotheses of interest. Average spike rate was analyzed with a one-way ANOVA, and corrected with a Bonferroni test. Direct comparisons between neuronal responses to CS1 and CS2 were assessed by the paired Student's t-test, and corrected for multiple comparisons by Bonferroni testing. For statistical analysis we used the program R 4.2.2 (Vienna, Austria)[60] and Graphpad Prism 9.1.0 (GraphPad Software, San Diego, California USA, www.graphpad.com), the latter also for graph design. z-score: We first calculated the mean baseline value M (mean of the baseline value from −0.5 to 0 s), the value S was calculated from this as the standard deviation. Peak z-score was then calculated as (HistogramMaximum−M)/S and "trough z-score as (HistogramMinimum -M)/S. The time-locked response to CS was calculated as peak+trough z-score (from 0 to 0.5 s).

## Reporting summary
Further information on research design is available in the Nature Portfolio Reporting Summary linked to this article.

## Data availability
Source data are provided with this paper on the Zenodo database under accession code https://doi.org/10.5281/zenodo.10492711[61].

## Code availability
R code used in this study has been deposited in the Zenodo database under accession code https://doi.org/10.5281/zenodo.10492711[61].

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

## Acknowledgements

We deeply thank A. von Gunten (Head of old-age psychiatric service (SUPAA), Dept. Psychiatry, CHUV, Prilly-Lausanne) for sharing the valuable clinical observations that led to the initial idea for conducting this study, V. Grinevich (Central Institute of Mental Health, Heidelberg University, Mannheim, Germany) for his generous donation of virus, F. Magara and B. Boury Jamot (behavioral core facility, Lausanne, Switzerland) for their precious advices and technical assistance, Y Blake for help with the English writing and N. Verda, R. Toma and S. Chen for verifying of the proofs. Special thanks to Drs. Setareh Ranjbar and Ivana Arsic for their valuable advice on statistical analysis. This work was supported by the Roche Postdoc Fellowship (Project RPF-ID:285) and the Marie Heim Vögtlin subsidy from Swiss National Science Foundation (PMPDP3 164468), both attributed to C. Hegoburu, and by grants from the Swiss National Science Foundation No's 31003A_138526, IZLSZ3_148803, 310030_192463, IZLIZ3_200297, IZLCZ0_206045 and Synapsis Foundation No. 2020-PI02, attributed to R. Stoop.

## Author contributions

C.H., C.G., Y.T., and R.S. conceived and designed the study. C.H., R.T.D.R., I.S., D.M.C. and Y.T. performed the behavioral experiments. C.H., S.G., R.N., and Y.T. performed the electrophysiological recordings. M.A. wrote the MatLab scripts for the electrophysiological recordings and analyses. C.H., R.N. and Y.T. performed the spike sorting and analyses of spiking patterns. C.H. and R.T.D.R. performed the immunohistochemical analysis. E.H. B. performed the statistical analysis, E.H.B., C.G., and R.S. supervised the project. C.H., Y.T., E.H.B. and R.S. wrote the manuscript.

## Competing interests

The authors declare no competing interests.
