## [Peer Review File · Nature Communications]

Social buffering in rats reduces fear by oxytocin triggering sustained changes in central amygdala neuronal activityREVIEWER COMMENTS

Reviewer #1 (Remarks to the Author):

This is an interesting study exploring the role of oxytocin-producing neurons in the hypothalamic paraventricular nucleus (PVN) and their projections to the central nucleus of the amygdala (CeA) in the mechanisms of social buffering of fear. SBF is an understudied phenomenon that is manifesting as a decrease in the conditioned fear response when fear memory is tested in the presence of an unconditioned conspecific. Conceptually, this work, focusing on the role of oxytocin in SBF, is very novel. However, there are certain issues with this study which should be resolved before the manuscript could be recommended for publication.

1. As acknowledged in the manuscript, the PVN sends projections to many brain regions in addition to the CeA. Therefore, the most straightforward way to test the role of OTergic PVN-CeA projection in the mechanisms of SBF would be to use optogenetic stimulation of OTergic terminals in the CeA, not OT neurons in the PVN as it was done here. The results of chemogenetic experiments are also inconclusive in respect to the projection-specificity of the endogenous OT effects on SBF because CNO was administered systemically.
2. Ideally, the manuscript should contain at least some evidence (e.g., in slice recordings) that chemo- or optogenetic interventions affected neurotransmission in PVN-CeA projection in the expected way. That's routinely done by other investigators.
3. Whereas behavioral studies are well designed and produced interpretable findings, the results from single unit recordings in the CeA are less conclusive. It remains unclear whether the observed changes in spiking patterns in the CeA necessarily resulted from increased activity in direct projections from spiking OT PVN neurons to the CeA.
4. Another potential issue with an interpretation of the single unit results is that PVN OT neurons are likely co-releasing glutamate with oxytocin. The authors may think about additional experiments linking the effects of activation of OT neurons in the PVN on neuronal spiking in the CeA to activation of OT receptors in the CeA. It could also provide direct evidence that activation of OT-positive PVN projections in the CeA does, in fact, result in oxytocin release in the CeA.
5. It might be helpful if the manuscript contained at least some ex vivo recordings demonstrating the properties of synaptic transmission in the OT PVN projections to CeA neurons. For example, the authors could directly demonstrate in slice recordings that OT released from OTergic PVN terminals triggers spiking of CeA neurons even with glutamatergic neurotransmission being blocked.
6. I personally would like to see examples of ChR2 expression in OT PVN neurons as well as ChR2-expressing OTergic PVN projections in the CeA. The expression of inhibitory DREADD in the PVN should also be illustrated.

Reviewer #2 (Remarks to the Author):

Hegoburu et al. investigate the mechanism of a unique form of fear extinction that is achieved through social buffering (SBF). Using chemogenetic, optogenetic, and pharmacological tools, they provide evidence that OT released from the PVN axons in the CeA is required for SBF and fear extinction. The authors detect changes in firing of OT neurons in PVN and in three types of neurons in the CeA along the course of the SBF paradigm and conclude that the observed long-lasting changes mediate the SBF-driven fear extinction.

The finding of the role of OT in CeA in fear extinction and SBF are unique and add important mechanistic information in the field of social modulation of fear. They are strongly supported by experiments with highly specific chemogenetic, optogenetic, and pharmacological manipulations of OT signaling in the CeA. On the other hand, there are technical issues and questions about interpretation of the in vivo recording data. Without proper controls, the causal link between neuronal firing and fear extinction remains unproven (detailed in 3-6 below).

1. The CNO control in DREADD experiments was performed on mice without viral transduction. Potential non-specific effects of "viral transduction" x "CNO" interaction must be controlled by using AAV that expresses something else instead of DREADD molecules. Such control must be included in the main Figure 3.
2. In experiments with Chr2, the blue light was at high power. There is no control for non-specific effects of "light+viral transduction". Stimulation in rats expressing GFP instead of Chr2 could provide such control.
3. The in vivo recording of neuronal activity in the CeA and in PVN along the entire behavioral paradigm lacks an essential control, the animals that were not exposed to the companion. Without this control, it is impossible to attribute any changes in neuronal activity to social buffering.
4. It is necessary to know whether the companion presence alters OT neuronal activity in PVN of the animals that are not fear conditioned. Do fear conditioning and companion presence during fear recall have cumulative, additive, or other effects on OT neuronal firing during CS presentation?
5. The activity of OT neurons and CeA neurons (except the Type 2 cells) on Fig. 5 appears independent on CS and invariant throughout each recording session. Can the activity differences among sessions represent homeostatic changes triggered by fear conditioning per se unrelated to fear recall and SBF? Is the increase on Day2 at SBF really caused by the companion animal? Or it happens gradually during the 3 h interval between "fear recall" and "SBF" and does not require the companion animal?
6. What is the dynamic of neuronal activity following fear conditioning training without fear recall or SBF along the time course of the behavioral paradigm?

7. Do the changes in CeA neuronal activity require OT?
8. Is there evidence that OT neurons were indeed silenced and less OT was released during DREADD suppression?
9. In Figure 2, for the SBF “Companion” group, the freezing level is twice higher than for the SBF “Companion” group in Figure 1. They are supposed to be comparable.

Minor:

1. Context-independence of the SBF-mediated fear extinction is a unique and important property. It is mentioned in Discussion but the supporting data is not shown in the main figures.
2. The analysis SBF retention with CS2 must be included in the main Fig. Given that fear conditioning depends on CS frequency (<https://www.ncbi.nlm.nih.gov/pmc/articles/PMC7190350/>), it is essential to show the extinction does not depend on CS frequency.
3. The AAV titer information is missing.

Reviewer #3 (Remarks to the Author):

In this manuscript, the authors delineate the function of the neuropeptide oxytocin (OT) in social buffering of fear in rats. The authors hypothesize that OT release from the PVN into the central amygdala—a fear processing center with dense OT innervation and receptor expression—likely underlies this social support behavior. Using a series of pharmacogenetic manipulations and in vivo electrophysiological recordings, the authors conclude that OT transmission in the central amygdala is necessary for the anxiolytic effects of social buffering and that the circuit, extending from paraventricular hypothalamus to central amygdala, is fundamentally different from previously characterized fear extinction pathways involving the medial prefrontal cortex and surrounding amygdalar nuclei. Multiple aspects of study design, analysis, and interpretation of the data diminish the enthusiasm for the strength of current conclusions.

Here, animals must be individually housed for all behavioral experiments, to match housing conditions for animals with implanted electrodes. There is extensive literature on the stress associated with single housing of animals and associated neuropeptidergic mechanisms (Zelikowsky et al., 2018, Lee et al., 2021, Matthews GA, et al., 2016). This might preferentially increase the fear-reduction aspects of social buffering of fear (SBF) and its retention, characterized in Figure 1, as well as present potential confounds for oxytocin modulation. For the conclusions of this study it is important to demonstrate that

single housing of animals does not alter SBF. Generally, stressor levels are not well matched in conditions across the study, including combinations of isolation housing, novel conspecific introduction, etc.

In the main body of results, the authors do not further clarify or explain the finding that TGOT application/ChR2 stimulation produces only an acute reduction in fear responses and does not play a role in retention of SBF. The authors should further expand on this and discuss that the more long-term effects of SBF likely rely on a form of observational learning component, whereas a single injection of TGOT (or OT stimulation) likely produces only a transient anxiolytic effect that does not actually contribute to the extinguishing of fear.

Does increased social interaction (%time nose/nose and %time nose/wall) positively correlate with reduction of fear response? If so, active attention to conspecific may predict SBF expression on Day 3. This measure may also account for some of the individual variability present in the data.

The authors used two different auditory conditioned stimuli CS1(5kHz) and CS2 (15kHz), subsequently fear conditioned by pairing each CS with an electric foot-shock of 0.5 mA on Day1. On day 2 recall of fear memory was assessed on to each CS, but the authors only measured SBF during CS1 re-exposure. Why was CS2 was not assessed in SBF, while for the retention of SBF assessment both CS1 and CS2 were used?

Since the modulatory effects of Oxt can reconfigure circuits and alter behavior on the orders of days, the authors might consider expanding the time course of the social buffering retention. The authors perform the retention task just 48 hours after the initial training, which is not necessarily 'long-lasting'. This data may also further draw helpful comparisons between fear extinction—where reappearance of fear responses can occur via renewal, spontaneous recovery, or reinstatement shortly after conditioning—and social buffering of fear that may operate in circuits separate from extinction, as the authors posit, and therefore be a more persistent type of fear inhibition.

Similarly, another potential internal control for the validation of the paradigm would be to have a cohort of animals re-exposed to both auditory cues during SBF induction, but only a companion is present during one auditory cue (while the other cue is replayed without companion). The authors could then compare the fear reduction caused by extinction (repeated exposure to stimulus without foot shock) and that caused by presence of an untrained conspecific to provide further mechanistic clarity into the two forms of fear reduction. This control would also help with the interpretation of fig s3 and fig 6, which lack a 'no companion' internal control.

Line 121, the authors mention that Day 3 CS1 + no companion group was "similar to CS2" in figure 1B. Please provide statistical evidence for this comparison.

Overall, the authors have devised a paradigm that might offer the opportunity to compare SBF-induced fear reduction and fear extinction but do not perform these types of analyses. Also, considering that TGOT/OT stimulation in CeA produces only a transient effect in fear reduction, it is unlikely that SBF is mediated through a pathway "distinct from mPFC-BLA pathway" as mentioned in the introduction and discussion. Cortical activity and BLA are likely needed for the prolonged fear reduction and therefore the

authors cannot confirm that PVN-CeA pathway is sufficient for SBF, as mentioned in discussion (line 302), so all statements mentioning the ‘independence’ of PVN-CeA pathway for SBF or how SBF is ‘completely subcortical’ should be moderated. The study provides no piece of evidence that BLA is not involved in the used SBF behavioral paradigm.

In fig 5 or supplemental fig 6, authors should provide more data related to waveform/spike sorting and PCA analyses that led to derivation of only three neuron types (e.g. % variance plots, k-means clustering). Additionally, the authors mention in line 455 of methods that neurons were selected for recording based on responses to CS1/2, however, the only type 2 neurons appear to show a distinct, time-locked response to presentation of auditory cue. Authors should clarify this experimental criterion further and explain how they ‘manually’ defined clusters as opposed to alternative automated methods.

Considering that the authors are able to track neural activity of units reliably over the course of days, it would be helpful that bar plots to the right in Fig 5, showed paired data in order to depict neural activity changes over the course of the behavioral paradigm for each neuron type (as opposed to grouping all the data together). In addition, tracking same units over time should be shown for more than a selected example. How consistent is this? Of note – it is unusual to be able to stably track units for many days, so failure to do so is acceptable. However, we just do not have enough QC info on the current dataset.

In line 199, authors state “Type 1 neuronal changes correlated closely with PVN neuronal activity”, but do not provide any concrete correlational analyses.

In line 201-203, authors state that Type 1 neuron activity “derives directly from OT-sensitive CeA neurons”. This finding is not currently supported by any data provided. There is no way to exclude numerous indirect circuit effects in these experiments.

It also suggests that PVN-CeA circuit is primarily glutamatergic (i.e. that PVN input is directly eliciting activity in CeA neurons) in which there is conflicting evidence in the field that OT neurons can release glutamate as opposed to a primary neuromodulatory role in the brain.

While it is compelling that the authors found a neuron group that increased activity during the retention of SBF, it would be more significant if the increase in activity was not so generalizable and time-locked to the presentation of the CS. Is this a feature to be expected for maintained inhibition of CeLON neurons (as mentioned in lines 275-76) or should the activity (and therefore posited inhibition of lateral amygdalar neurons) be more time-locked to the conditioned stimulus?

The behavioral analysis depicted in fig 4e also shows low subject number, but it appears in fig 5 that the authors have more animals/data to contribute to this dataset. Is there an explicit reason why the authors have a low n for fig 4e?

Molecular identification for functionally defined amygdala neurons modulated by OT would be very useful for the resulting overall model of OT effects on SBF.

Minor points:

Figure 2b: Day 2 (right) plot should have 'SBF' (plot under light purple color) instead of "TGOT" similar to Fig 2a.

Generally, confirmed expression patterns from DREADD and Chr2 experiments should be shown.

Figure 4 d: The images are arranged in a confusing manner. Areas marking BLA and CeA are a bit off than normal. A full section image would be ideal.

The data points on all histograms and plots are too small

In each figure CS1 (5kHz) and CS2(15 kHz) are designated as Experimental and Control respectively. It is mentioned that CS1 and CS2 are for internal comparisons, however, designating them as Experimental and Control is confusing.

Include more human lit in the introduction, such as Bratec et al 2020

Figure 4c: the y axes of all the plots are not starting from zero

Figure 4e: needs more data points (error bar of "Pre-Tone" in SBF is extreme). "Hab" should be replaced with "Habituation" to maintain the uniformity across the paper. Same is in Figure 5 a,b,c (histograms)

Figure 6 seems out of order and should likely be combined with figure 2, as both figures are related to how oxytocin may regulate the acute and/or long-term effects of SBF.

REVIEWER COMMENTS

Reviewer #1 (Remarks to the Author):

This is an interesting study exploring the role of oxytocin-producing neurons in the hypothalamic paraventricular nucleus (PVN) and their projections to the central nucleus of the amygdala (CeA) in the mechanisms of social buffering of fear. SBF is an understudied phenomenon that is manifesting as a decrease in the conditioned fear response when fear memory is tested in the presence of an unconditioned conspecific. Conceptually, this work, focusing on the role of oxytocin in SBF, is very novel. However, there are certain issues with this study which should be resolved before the manuscript could be recommended for publication.

1. As acknowledged in the manuscript, the PVN sends projections to many brain regions in addition to the CeA. Therefore, the most straightforward way to test the role of OTergic PVN-CeA projection in the mechanisms of SBF would be to use optogenetic stimulation of OTergic terminals in the CeA, not OT neurons in the PVN as it was done here. The results of chemogenetic experiments are also inconclusive in respect to the projection-specificity of the endogenous OT effects on SBF because CNO was administered systemically.

We thank the reviewer for this suggestion. To test more in depth the role of OTergic PVN-CeA projections in the mechanisms of SBF, we have added a new series of experiments in which we combined *in vivo* electrophysiological with optogenetic and chemogenetic manipulations to selectively tag and modulate CeA-projecting OT neurons from the PVN. For this, we injected in the CeA a retrograde CAV2 virus expressing Cre, and in the PVN an AAV virus expressing a double floxed DIO mCherry-ChR2 (respectively DIO mCherry hM4Di) under the OT promoter (pOT, see **new supplementary Fig. 5a&b**). This combination allowed us to express ChR2 (respectively hM4Di) in those oxytocinergic neurons in the PVN that project to the CeA.

To further study the mechanisms of SBF, we then implanted for the optogenetic experiments optrodes in the CeA to stimulate local OTergic fiber endings from the PVN with blue light and simultaneously record neuronal activities in the CeA, and we applied the social buffering of fear protocol to observe changes in behavior and in spiking patterns. For the chemogenetic experiments, we injected CNO to observe effects on neuronal spiking activity *in vivo* in the CeA. The precise results of these experiments can now be seen in **new Fig. 6 and new supplementary Figures 5, 6, 7 and 8** and will be described in further detail together with our answers to the question of the reviewer about the electrophysiological recordings (see below points 3-6).

These new electrophysiological *in vivo* experiments expand on previous behavioral observations in which we had already found that local optogenetic stimulation in the CeA of ChR2-expressing OTergic fibers in the CeA could acutely reduce fear responses by their endogenous release of OT (Knobloch et al., Neuron 2012, Fig. 5), similar to the effects that we had found after local injection of the specific oxytocin receptor agonist TGOT in the CeA (Viviani et al., Science 2011).

We would like to mention that the initial rationale to optogenetically stimulate the OTergic neurons in the PVN instead of their projections in the CeA was driven by our finding that injection of TGOT directly in the CeA (previous "Fig. 2b", now "Fig. 2c") showed acute, reversible reduction of freezing, but was unable to reproduce the retention effects of SBF. We had reasoned that the induction of local release by optogenetic stimulation in the CeA might only induce acute fear reduction (as found

in Knobloch et al., 2012) without retention of SBF: That the ability of the presence of the companion to induce retention of SBF was due to its stimulation of the PVN OTergic projections to other brain regions besides the CeA. We therefore tested whether the optogenetic stimulation in the PVN (and with it, all OTergic projections from the PVN), might be sufficient for inducing retention of SBF. However, as can be seen in Fig. 3b, although this stimulation of OT neurons in the PVN was again able to induce an acute SBF (similar to the local application of TGOT in the CeA) it was not sufficient to also induce a retention of SBF.

It thus appears that, besides the additional OTergic projections from the PVN, there is another factor involved for retention of SBF. As we hope the reviewer may appreciate (see answers below as well as **new Fig. 6**) our new *in vivo* electrophysiology experiments provide deeper insights in this mechanism of "retention of SBF".

2. Ideally, the manuscript should contain at least some evidence (e.g., in slice recordings) that chemo- or optogenetic interventions affected neurotransmission in PVN-CeA projection in the expected way. That's routinely done by other investigators.

We thank the reviewer for raising this point. To confirm this, we did several additional experiments:

1. For our optogenetic intervention we now include *in vitro* slice recordings to prove that activation with blue light of OTergic projections from the PVN in the CeA leads to an increased neurotransmission from these projections. This is evidenced by an increased frequency of inhibitory postsynaptic currents in the medial part of the central amygdala (**see new supplementary Fig. 5b**). We furthermore show that this increase is significantly reduced by perfusion with the oxytocin receptor antagonist OTA (d(CH₂)₅¹,Tyr(Me)²,Thr⁴,Orn⁸,des-Gly-NH₂⁹)-Vasotocin.

These results confirm our previous results with the same AAV construct (Knobloch et al., Neuron 2012) that we now combined with CAV2 expressing CRE for retrograde labeling. These had shown that after AAV expression of Chr2 under the OT promoter in the PVN:

a) *In vitro* blue light (BL) exposure of the PVN induces inward currents in OT PVN neurons that are sufficiently large to trigger action potentials (Knobloch et al., Neuron 2012, Fig.4C).

b) *In vitro* BL exposure in the CeA increases spontaneous spiking in the GABAergic neurons in the CeL whose projections to the CeM caused increased inhibitory postsynaptic currents. These latter could be significantly blocked by OTA and completely by a cocktail of OTA and NBQX, pointing to a co-release of OT and glutamate from the OTergic PVN projections (Knobloch et al., Neuron 2012, Fig.4D&E).

c) *In vivo* in the CeA: We had found that BL exposure reversibly inhibited freezing after fear conditioning. This BL effect could be blocked by prior injection of oxytocin antagonist OTA into the CeA (Knobloch et al., Neuron 2012, Fig.5).

2. For the Chemogenetic inhibition, we now provide *in vitro* evidence in slices that the application of CNO (after selective expression of hM4Di in OT PVN neurons that project to the CeA) decreases the number of action potentials induced by current injections (see **new supplementary Fig. 5a**). Again, the outcome of this experiment is comparable to previous results with the same viral construct on which the current virus approach is based (Eliava et al Neuron, 2016). There we had expressed hM4Di DREADD under the same oxytocin promoter in PVN neurons and showed in *in vitro* slice preparations that exposure to CNO led to an inhibition of the action potentials triggered by depolarizing currents (Eliava et al Neuron 2016, Figure S7).

3. In addition, we have also tested *in vivo* how neuronal spiking activity in the CeA is affected by optogenetic and chemogenetic modulation of OTergic PVN projections. These findings reveal that, after expression of Chr2 in the OTergic projections from the PVN to the CeA, BL applied locally in the CeA increases spiking activity of one group of CeA neurons and decreases activity in another CeA group. Both the excitatory and inhibitory BL effects can be significantly blocked by the blood brain barrier permeable OT receptor antagonist L-368 899 (Tocris, Bristol, UK, as above) and completely blocked by CNO. These findings, shown in **new supplementary Fig. 7**, demonstrate also *in vivo* how BL in the CeA affects neurotransmission by releasing OT from the PVN.

Taken together, both these *in vitro* and *in vivo* experiments demonstrate how chemo,- and optogenetic interventions ***affected neurotransmission in oxytocinergic PVN-CeA projection in the expected way***. These new findings are shown in **new supplementary figures 5 and 7** and referred to in the main text (***lines 153-157 & 219-222***) and by referring to our previous publications.

3. Whereas behavioral studies are well designed and produced interpretable findings, the results from single unit recordings in the CeA are less conclusive. It remains unclear whether the observed changes in spiking patterns in the CeA necessarily resulted from increased activity in direct projections from spiking OT PVN neurons to the CeA.

We thank the reviewer for this interesting question.

As shown in figure 5, we found in the CeA changes in baseline activity across the consecutive behavioral exposures (on Day 2 after fear conditioning, during SBF, and on Day 3 after SBF) and three distinguishable patterns in these changes which allowed us to identify type 1, type 2 and type 3 neurons. Of particular interest, the pattern in type 1 neurons fully correlated with the pattern that we found in OTergic PVN neurons and as now further confirmed by cluster analysis (see **new Fig. 5c**). Considering the presence of OT receptors on (CeLoff) neurons in the CeA (Haubensak et al., Nature 2010), this correlation suggests that the changes in type 1 neurons are caused by OT released in the CeA from OTergic PVN projections.

To obtain more conclusive evidence for a causal relation, we performed a new series of *in vivo* electrophysiology experiments in which we expressed Chr2 and hM4Di in projections from the PVN to the CeA. In 7 rats we were able to identify a total of 146 single units among which we found 48 units with excitatory or inhibitory responses to blue light. These showed changes in acute responses to CS1 and CS2 throughout the behavioral protocol that allowed us to distinguish two types of neurons. We have focussed the question of the reviewer further on these changes in acute responses to the CS induced by SBF. To show a causal relation we performed the following series of experiments:

1. Optogenetic stimulation and SBF exposure in CeA neurons:

The introduction of local BL exposure in the CeA allowed us to identify two types of spiking responses in CeA neurons (**see new Fig. 6 and new supplementary Fig. 7a**): Type 1 neurons that increased their spiking frequency to BL (and increased spiking frequencies to CS1 during SBF) and Type 2 neurons that mostly decreased spiking frequencies to BL (and decreased CS1 responses during SBF) .

2. CNO is effectively blocking both types of changes

By co-expressing hM4Di in the OTergic neurons projecting from the PVN to the CeA, we could show that the BL-evoked excitatory and inhibitory responses could be blocked by intraperitoneal

injections of CNO. This shows that CNO can effectively block activity of OTergic neurons with projections in the CeA (***new supplementary Fig. 7b3 and c3***).

3. SBF after fear conditioning induces similar changes as BL excitation

We found that the distinction in Type 1 and Type 2 neurons based on their additional excitatory and inhibitory responses to BL (as defined under point 1) corresponded with similar changes in baseline firing patterns during SBF in Type 1 and Type 2 neurons (***new supplementary Figure 5c***) as well as to their changes in acute responses to CS1 (&CS2) after fear conditioning that, after SBF, for CS1 decreased in Type 2 neurons and increased in Type 1 neurons (***new Figure 6***). Type 1 and Type 2 neurons thus appear to correspond to respectively the CeLoff and CeLon neurons of Ciochi et al., and Haubensak et al., (Nature 2010a,b)

4. SBF-induced changes in basic spiking activity can also be blocked by CNO.

We then administrated, after exposure to SBF, CNO to observe effects on acute spiking responses and on behavioral responses to CS1 (as compared to CS2). As can be seen in ***new Fig. 6g & 6h*** after SBF (Day 3), CNO blocked both the changes in acute responses to CS1 (increases in type 1 neurons and decreases in type 2 neurons) and the buffering of fear responses to CS1 (as compared to CS2).

Together, we think these findings show more conclusively, that "*the observed changes in spiking patterns in the CeA (in acute responses of type 1 and type 2 neurons after SBF) necessarily resulted from increased activity in direct projections from spiking OT PVN neurons to the CeA.*"

At the same time, we realize that our new experimental results also raise further questions. Thus, one can wonder whether a sustained activation of OTergic PVN neurons after SBF is necessary to maintain the changes in the CeA or whether their transient activation (during SBF) is sufficient for a sustained induction of changes. Indeed, it is possible that a one-time increase in release of OT in the CeA (triggered by SBF) can induce long-term changes in activity in combination with the exposure to the CS through Hebbian plasticity mechanisms. A larger, in depth, study would be necessary to find the precise requirements for when and under which conditions activity in projections from the OTergic PVN neurons is able to induce immediate and maintained changes in spiking patterns in the CeA.

In the revised version of the manuscript we have added ***new figures 6 and new supplementary figures 5-8***. In the revised discussion we have inserted text to clarify how we think, based on these results, that the "*observed changes in spiking patterns in the CeA resulted from increased activity in direct projections from spiking OT PVN neurons to the CeA*" (***lines 324-333***).

4. Another potential issue with an interpretation of the single unit results is that PVN OT neurons are likely co-releasing glutamate with oxytocin. The authors may think about additional experiments linking the effects of activation of OT neurons in the PVN on neuronal spiking in the CeA to activation of OT receptors in the CeA. It could also provide direct evidence that activation of OT-positive PVN projections in the CeA does, in fact, result in oxytocin release in the CeA.

We thank the reviewer for raising this outstanding issue. In previous electrophysiological recordings in *in vitro* slices of the central amygdala (as mentioned above), we had found that effects of BL activation of OT terminals in the CeA were significantly, but not completely blocked by the oxytocin antagonist OTA. The remaining activation in the CeA could be completely blocked by additional slice perfusion of NBQX, indicating a co-release of glutamate (Knobloch et al., Neuron, 2012).

In further answer to the question of the reviewer we have co-expressed, in a new series of *in vivo* electrophysiology experiments, ChR2 and hM4Di in projections from the PVN to the CeA (using

the viral approach detailed above, **new supplementary Fig. 7a1**). We identified, *in vivo*, a series of neurons in the CeA that were acutely excited by blue light application (as can be seen in **new supplementary Fig. 7b1**). This response was significantly reduced by the oral administration of BBB-permeable oxytocin receptor antagonist L-368 899 (Tocris, Bristol, UK, **new supplementary Fig. 7a2&b2**) leaving a smaller remaining excitation that was subsequently fully blocked by IP administration of CNO (**new supplementary Fig. 7a3&b3**). In another series of neurons that were inhibited by BL we found similar progressive blocking effects by OTA and CNO (**new supplementary Fig. 7a3, c1-c3**). These findings suggest that the BL activation in the CeA of fibers projecting from the PVN causes indeed release of another neurotransmitter besides OT. Based on our previous *in vitro* recordings, this seems most likely glutamate. To test this further *in vivo*, however, we would need to apply an antagonist of glutamatergic transmission onto AMPA receptors similar to above-described experiments *in vitro*. Unfortunately, *in vivo* application of an AMPA receptor blockers such as NBQX would cause too many side effects on other neuronal pathways to furnish any further interpretable results. Indeed, our present *in vivo* preparation is not ideal for in depth pharmacological studies of the co-release of transmitters.

In the revised version of the manuscript we have included these latest *in vivo* findings in a **new supplementary Fig. 7**, and we have addressed this point in the discussion of the revised manuscript (**see lines 219-222 in result section & lines 301-303 in discussion**).

5. It might be helpful if the manuscript contained at least some *ex vivo* recordings demonstrating the properties of synaptic transmission in the OT PVN projections to CeA neurons. For example, the authors could directly demonstrate in slice recordings that OT released from OTergic PVN terminals triggers spiking of CeA neurons even with glutamatergic neurotransmission being blocked.

In answer to this question, we would first of all like to refer to our previous *ex vivo* recordings in which we extensively characterized the synaptic transmission in these OT PVN projections onto CeA neurons in terms of their reliance on OTergic and glutamatergic synaptic transmission for both the spiking of these GABAergic neurons (in the lateral CeA, CeL) and the postsynaptic effects on their GABAergic release in the medial CeA (CeM, Knobloch et al., Neuron 2012).

For the present manuscript we have, in addition, conducted the following experiments:

1. *Ex vivo*: We have used a double viral approach (as mentioned above) to specifically infect OTergic neurons in the PVN that project to the CeA in order to induce expression of Chr2. We then recorded by whole cell patch clamp from CeM neurons in which we observed increases in inhibitory postsynaptic currents upon blue light exposure. Following incubation with the OT receptor antagonist OTA, these currents were significantly reduced, thus demonstrating the excitatory effects of blue light on OT release (see **new supplementary Fig. 5b**). These results are consistent with our previous findings in which we characterized the same type of viral infection of PVN neurons with Chr2 under the oxytocin promoter and in which we showed after BL application: a) Increases in spiking of OTergic neurons in the PVN. b) Increases in spiking of GABAergic neurons in the CeL. c) Increases in postsynaptic inhibitory current in neurons of the CeM. The latter two types of increases could be significantly blocked by incubation with OT receptor antagonist and fully blocked by AMPA receptor antagonist NBQX, demonstrating the co-release of OT and glutamate (Knobloch et al., 2012, Fig. 4E2).

2. *In vivo*: In the revised manuscript we have completed the above *ex vivo* experiments with additional *in vivo* electrophysiology experiments in which we recorded from BL-excited and BL-inhibited spontaneous spiking neurons in the CeA corresponding, respectively, with the two types of *ex vivo* responses above (see **new supplementary Fig. 7**). We show that these BL effects are

significantly blocked by injection of the BBB permeable OT receptor antagonist and fully blocked by intraperitoneal administration of CNO. This shows that also *in vivo* the same types of neuronal responses can be found that are mediated by oxytocin. As mentioned above, it is difficult to test *in vivo* the involvement of glutamatergic transmission, because we cannot apply NBQX in the living animal without affecting other pathways.

In the revised version of the manuscript, we have added these new findings and we have referred to these previous findings in the discussion (*lines 219-222 in result section & lines 301-303 in discussion*)

6. I personally would like to see examples of ChR2 expression in OT PVN neurons as well as ChR2-expressing OTergic PVN projections in the CeA. The expression of inhibitory DREADD in the PVN should also be illustrated.

To answer the reviewer's request, we now show in *new supplementary Fig. 5* examples of immunostainings with the expression of inhibitory DREADD hM4Di and the blue light-sensitive ChR2 in PVN neurons and their overlap with OT expression. In this figure we also illustrate with whole-cell patch clamp recordings how CNO can inhibit the neuronal activation and how BL-activation of these ChR2 expressing OTergic PVN projections in the CeA can activate the local GABAergic circuitry as expected (see also above). We have also included in the revised version of the manuscript references on the functional expression of inhibitory DREADD (Eliava et al., Neuron, 2016) and the extensive, histological and functional characterization of the viral expression of ChR2 in OTergic PVN projections in the CeA and other brain regions (Knobloch et al., Neuron, 2012)

Reviewer #2 (Remarks to the Author):

Hegoburu et al. investigate the mechanism of a unique form of fear extinction that is achieved through social buffering (SBF). Using chemogenetic, optogenetic, and pharmacological tools, they provide evidence that OT released from the PVN axons in the CeA is required for SBF and fear extinction. The authors detect changes in firing of OT neurons in PVN and in three types of neurons in the CeA along the course of the SBF paradigm and conclude that the observed long-lasting changes mediate the SBF-driven fear extinction.

The finding of the role of OT in CeA in fear extinction and SBF are unique and add important mechanistic information in the field of social modulation of fear. They are strongly supported by experiments with highly specific chemogenetic, optogenetic, and pharmacological manipulations of OT signaling in the CeA. On the other hand, there are technical issues and questions about interpretation of the *in vivo* recording data. Without proper controls, the causal link between neuronal firing and fear extinction remains unproven (detailed in 3-6 below).

We thank the reviewer for the appreciation of our findings. To address the technical issues and questions about the interpretation of the *in vivo* data, we have conducted a new series of *in vivo* recordings. These *in vivo* recordings allowed us to identify 146 single unit activities in a total of 7 rats. By expressing a new combination of viruses in the CeA we could also test the responses of these units to local optogenetic activation of OTergic projections and their inhibition by CNO. These have given interesting new results, with which we have been able to better address the causal link between neuronal firing and fear extinction, as we hope the reviewer may appreciate in our answers to the questions below.

1. The CNO control in DREADD experiments was performed on mice without viral transduction. Potential non-specific effects of “viral transduction” x “CNO” interaction must be controlled by using AAV that expresses something else instead of DREADD molecules. Such control must be included in the main Figure 3.

To control for “potential non-specific effects of “viral transduction” x “CNO” interaction”, we have added experiments in which we used an AAV that expresses, instead of inhibitory DREADD, ChR2 and mCherry under the OT promoter in the PVN (characterized in our previous publications: Knobloch, Neuron 2012, Tang et al, Nat. Neurosci 2020, as well as in the present manuscript: Figs. 3b&4b and **new Supplementary Fig. 5b**) Intraperitoneal application of CNO in rats that expressed this viral construct did (as expected) NOT inhibit the social buffering of fear (**see new Supplementary Figure 4b**): Thus, the decrease in freezing of rats that were exposed to the CS1 in the presence of a companion was similar to animals that had received vehicle (Fig. 3a) or which had received CNO administration but no injection of viral construct (see Supplementary Fig. 4a). We have included these new findings, as requested by the reviewer in the **revised main Fig. 3a** (by an additional bar, in green). The complete findings are represented in **new supplementary Fig. 4b**. We have referred to these control experiments in **lines 144-146** in the revised version of the manuscript.

2. In experiments with ChR2, the blue light was at high power. There is no control for non-specific effects of “light+viral transduction”. Stimulation in rats expressing GFP instead of ChR2 could provide such control.

Similar to the above approach we have added a control experiment in which we used the viral construct that expresses hM4Di DREADD instead of Chr2 in the PVN under the OT promoter and exposed this to blue light "to control for potential non-specific effects of "light + viral transduction". The hM4Di construct for this experiment is one that we had already used and characterized in previous work for inhibition of PVN neurons by CNO (E.g. Eliava et al., *Neuron*, 2016) as well as in the present manuscript (Fig. 3a). In rats infected with this construct we applied blue light (instead of CNO) without the presence of the companion and this did, as expected, NOT induce any decrease in freezing behavior, similar to control rats that had NOT been injected with any viral construct and also received blue light (Fig. 3b). We show the complete findings of this experiment in **new supplementary Fig. 4c**. and refer to these in **lines 160-161** of the revised manuscript.

3. The *in vivo* recording of neuronal activity in the CeA and in PVN along the entire behavioral paradigm lacks an essential control, the animals that were not exposed to the companion. Without this control, it is impossible to attribute any changes in neuronal activity to social buffering.

To answer this question, we have conducted an additional series of *in vivo* electrophysiological recordings. We designed these experiments based on previous work published by colleagues in the field (e.g. Ciochi et al., *Nature* 2010 and Haubensak et al., *Nature* 2010) who studied neuronal activity in the CeA during auditory cued fear conditioning without social buffering of fear. Their findings had revealed two types of neurons in the CeA that exhibit acute, opposite responses to a single CS after fear conditioning which they called CeLon and CeLoff neurons. In our new findings, we have been able to identify a larger number of neurons with acute responses to both CS1 and CS2. We included the CS2 as control, as it was never played in the presence of the companion. Although we realize that this is not precisely the same experiment as the reviewer has asked for ("including neuronal activity in an animal that was NOT exposed at all to a companion"), we believe our experimental design shows, in fact, even better the contrast between the consequences for neuronal responses to CS1 and CS2 after, respectively, all or none exposure to the companion (since they occur in the same animal).

In short, for these experiments we injected an additional number of 7 rats with a combination of viral constructs in the CeA and PVN in order to express Chr2 selectively in those OTergic PVN neurons that project to the CeA. To this purpose we injected in the CeA a retrograde CAV2 virus expressing Cre, and in the PVN an AAV virus expressing a double floxed DIO mCherry-Chr2 under the OT promoter (pOT, see **new supplementary Fig. 5b**). After sufficient expression, we implanted optrodes in their CeA. We obtained a total of 146 recordings of single unit spiking activity of which 48 neurons showed distinctive combined responses to blue light and to the CS that could be divided into two groups:

Type 2 neurons (n=24) showed acute (transient) excitatory responses to the CS1 that increased after fear conditioning. These CS1 responses decreased during exposure to the companion ("SBF") and remained decreased the next day without the companion ("retention of SBF"). The development of these responses expand on the acute responses we had already found in previously recorded Type 2 neurons (Fig. 5). Responses to the CS2 were equally excitatory from the start and increased after fear learning but remained unchanged throughout all further sessions (as they were never evoked in the presence of the companion, Day 3 and 4, **new Figures 6, a3&a4, g, and Supplementary Fig. 8**). Responses to optogenetic activation of OTergic projections from the PVN were mostly inhibitory (**new Fig. 6f**). We refer to these as "**Type 2, Fear neurons**"

Type 1 neurons (n=24) were characterized by initially absent or slow, blunted inhibitory responses to the CS1 but, interestingly, these neurons started to develop acute, excitatory responses to the CS1 during exposure to the companion ("SBF") and these remained the next day without the companion ("retention of SBF", **new Figures 6, a1&a2, g and Supplementary Fig. 8**). Responses to

blue light were typically excitatory (*new Fig. 6f*). Responses of type 1 neurons to the CS2 (which was never paired with the presence of the companion) were absent or blunted and, contrary to their responses to the CS1, did not change across the different sessions. Responses to optogenetic activation of OTergic projections from the PVN were typically excitatory. Because of their increased acute responses during the buffering of fear and their activation by OTergic projections from the PVN (as the type 1 neurons in Fig. 5) we refer to these neurons as "**Type 1, Buffer neurons**".

The ensemble of these new results is now shown in three new figures that illustrate the changes in responses to CS1, CS2 and blue light (*new main Fig. 6 and supplementary Fig. 8*) and responses in combination with CNO and BBB-permeable OT receptor antagonist L-368 899 (Tocris, Bristol, UK, *new supplementary Fig. 7*). As the reviewer may appreciate, our new findings show that excitatory neuronal (and concomitant freezing) responses to CS2 (without exposure to the companion) do NOT change, whereas the excitatory neuronal (and freezing) response to the CS1 (with exposure to the companion) significantly decreased during SBF and remained decreased the following day.

Thus, through these experiments we are now able to more precisely attribute the changes in neuronal activity to social buffering, as they concentrate on the changes in CS-triggered activity instead of only on baseline responses that we previously assessed (old Fig. 5). We hope the reviewer appreciates our new findings.

4. It is necessary to know whether the companion presence alters OT neuronal activity in PVN of the animals that are not fear conditioned. Do fear conditioning and companion presence during fear recall have cumulative, additive, or other effects on OT neuronal firing during CS presentation?

We thank the reviewer for raising this interesting point. In previous work, we have exactly been able to show that the presence of a companion with a rat that is not fear-conditioned can significantly increase OT neuronal activity in the PVN. Increases could amount up to, on average, 150% (Tang et al., Grinevich, Nat Neuroscience 2020, Fig. 1d). Interestingly, fear learning by itself can also significantly increase baseline activity of OTergic PVN neurons as shown by Hasan et al (...Grinevich, Neuron 2019) and similar to what we have also found in PVN (and type 1 CeA) neurons (in Fig. 5b during "Fear Recall"). These increases were comparable to the increases found after introduction of the companion (Tang et al., Nat. Neurosci. 2020). It thus seems that OTergic neurons in the PVN are, on one hand, sensitive to external stimuli (triggered by the presence of a companion) and, on the other hand, also to internal stimuli (triggered by fear conditioning). The latter could represent a type of homeostatic reaction through which the effects of fear conditioning/fear recall can be internally buffered (as the reviewer also suggests below in the next point).

In light of these new findings, the second part of the reviewer's question represents a very interesting expansion on this first point ("Do fear conditioning and companion presence during fear recall have cumulative, additive, or other effects on OT neuronal firing during CS presentation?"). One may indeed wonder whether the very same OT neurons that are increasing activity after fear learning (reflecting the homeostatic, internal buffering of fear) are also the neurons that are sensitive to the external cue (the presence of the companion), i.e. whether individual neurons show cumulative effects in their responses. To answer this question, we have, at the single cell level, analyzed whether the same cells that increased baseline activity during the fear recall ("Fear Recall") session still increased activity further during the social buffering ("SBF") session. Alternatively, a different group of PVN neurons might have started to increase activity during SBF, on average still leading to a total increase of average spike rates during SBF

To illustrate our analyses, we have drawn, in the *revised version of Fig. 5a&b*, lines between individual dots to track spike rates of the same individual cell across different sessions. As can be seen, PVN (and type 1 CeA) neurons that increased activity during fear recall, showed additional increases in baseline activity during SBF i.e. there are "additive/cumulative" effects on OT neuronal firing. This suggests that the activated PVN neurons are sensitive both to internal as well as external signals, i.e. that the very same OTergic neurons that play a role in encoding the "internal buffering" of the fear response (maintaining homeostasis after fear conditioning) also are activated by the external signals that induce the social buffering of fear.

Besides these modifications in (revised) figure 5, we have added a text in the discussion of our revised manuscript to address this point (*lines 276-281*). We thank the reviewer for raising this most interesting question which has allowed us to deepen our initial message.

5. The activity of OT neurons and CeA neurons (except the Type 2 cells) on Fig. 5 appears independent on CS and invariant throughout each recording session. Can the activity differences among sessions represent homeostatic changes triggered by fear conditioning per se unrelated to fear recall and SBF? Is the increase on Day2 at SBF really caused by the companion animal? Or it happens gradually during the 3 h interval between "fear recall" and "SBF" and does not require the companion animal?

We thank the reviewer for making this observation. Indeed, in the recordings that we had been able to obtain for Fig. 5 we only had a few Type 2 neurons with responses to the CS and we had only had found type 1 neurons that did not show development of responses to the CS. At the same time the baseline activity of these neurons changed across the different sessions. It is possible that the changes in baseline activity after fear conditioning represent a homeostatic change to the fearful stimulus (as we discussed at point 4, above). The increased baseline activity in type 1 neurons on Day 2 after SBF, however, seems to be really caused by the introduction of the companion animal as it did not appear before the introduction of the companion. To illustrate this, we have now included in *new supplementary figure 5c* two examples of baseline recordings of type 1 buffer neurons before and during the introduction of the companion.

At the same time, in our new electrophysiological recordings (as described above under point 3 and below under point 7) we also found acute activity responses to the CS 1 and 2 that differentially changed for type 1 and type 2 cells across the behavioral sessions. Based on these new findings we can formulate the following responses to the questions of the reviewer: i) that the (baseline) activity differences across sessions may represent changes that are mainly mediated by activity in OTergic projections from the PVN. ii) that fear conditioning initially increases some OT release (as an initial response to the fear conditioning to keep homeostasis) leading to an increase in baseline activity in Type 1 Buffer neurons. iii) During SBF, as a result of the presence of the companion, OTergic PVN neurons are strongly activated and cause further increase baseline activation of Type 1 buffer neurons. Considering these neurons can make local GABAergic synapses in the CeA, this may lead to a concomitant inhibition of baseline activity of the Type 2 Fear neurons.

We have included further text on this most interesting point in the discussion of the revised version of the manuscript (*lines 276-278*).

6. What is the dynamic of neuronal activity following fear conditioning training without fear recall or SBF along the time course of the behavioral paradigm?

It is difficult to answer this question, because it is through their changes in neuronal activity after fear i.e. during fear recall and SBF that we are able to identify recruited neurons and classify them into Type 1, 2 and 3 categories in Figure 5.

However, with the new recordings, in which we were able to identify multiple acute neuronal responses to the CS1 and CS2 after fear conditioning on Day 1, we can answer this question with regard to the responses to the CS2 stimulus (which was not exposed to the fear recall+SBF along the time course of the behavioral paradigm on Day 2). In these recordings we found, both in type 1 (CeLOff, buffer) neurons and type 2 (CeLOn, fear) neurons, changes in acute responses to CS2 across the 3 day behavioral paradigm. Thus, on Day 3, type 2 neurons showed increased responses to the CS2, and type 1 neurons showed decreased (or no increased) responses to the CS2 (**new Fig. 6g**). These latter results conform the previous findings of our colleagues who conducted similar measurements after fear conditioning to only one CS stimulus (without fear recall on Day 2). They also found on Day 3 increased acute responses in CeLOn (our type 2) neurons and decreased (or no changed) acute responses in CeLOff (our type 1) neurons (Ciocchi et al., Nature 2010, Fig. 5k&l; Haubensak et al., Nature 2010, Fig. 2a&b).

7. Do the changes in CeA neuronal activity require OT?

In our new *in vivo* electrophysiological findings in which we show a substantial number of neurons that exhibit changes in acute responses to the CS, we have assessed whether these changes require OT. To examine this requirement, we have used a combinatorial approach of the two viral ChR2 and hM4Di constructs to label OTergic PVN neurons that project to the CeA (based on the constructs that we characterized in Knobloch et al., Neuron 2012, Eliava et al., Neuron 2016) with which we injected 7 rats for electrophysiological *in vivo* recordings. After implantation of optrodes targeting the CeA we have been able to obtain single unit recordings in a total of 146 neurons. With these we demonstrate consecutively:

1. BL excitation of ChR2 expressing OTergic CeA projections induced changes in CeA spiking patterns:

The blue light exposure in the CeA allowed us to distinguish between two types of changes in spontaneous local spiking activities: 1) neurons that increased their frequency 2) neurons that decreased frequencies (besides neurons that were not affected by blue light).

(see **new supplementary Fig. 7**)

2. Both CNO and OTR antagonist L-368 899 are effective in blocking both types of changes

By co-expressing hM4Di in the OTergic neurons projecting from the PVN to the CeA, we could show that the BL-evoked responses (inhibitory as well as excitatory) could be significantly reduced by BBB permeable OT receptor antagonist L-368.899 and completely blocked by intraperitoneal injections of CNO. This shows that CNO effectively blocks activity of OTergic neurons with projections in the CeA (**new supplementary Figs. 7a2,a3, b3&c3**).

3. Changes in acute responses after fear conditioning and after SBF

Within these 146 neurons, we classified neurons in two groups based on their development of acute responses to CS stimulations across different sessions (as described above in our answer to question 3). We found that Type 1 and Type 2 neurons demonstrated similar changes in acute responses to the CS as found by our colleagues in respectively CeLOff and CeLOn neurons of the central amygdala after fear conditioning (Ciocchi et al., and Haubensak et al., Nature 2010a,b)

4. SBF-induced changes in acute responses can also be blocked by CNO.

After exposure to SBF, we then administered CNO, to observe whether it could reverse the changes in spiking activity that accompanied the behavioral changes to CS1 and CS2. As can be seen in **new Figs. 6f&g**, the increase in acute responses to CS1 in type 1 neurons after SBF could

also be blocked by administration of CNO as could the decreased acute responses of type 2 neurons to the CS1.

Together, we think these findings show that "*the changes in CeA neuronal activity (notably the responses to CS1) require OT*".

At the same time, we realize that our new experimental results also raise additional questions about the precise timing of the involvement of OT. Thus, one can wonder whether the development of these changes in acute spiking responses (that parallel the behavioral responses) requires the activity of OTergic neurons during SBF. From the behavioral experiments (that were performed with or without OTA in parallel groups of animals, *see Fig. 2a*), this indeed seems to be the case.

To assess this in parallel groups of neurons for the *in vivo* recordings would require a substantially larger number of experiments (as we would have to compare how many neurons show changes in acute responses across the different sessions in animals all or not exposed to OTA). Indeed, it is possible that postsynaptic actions of OT in the CeA induces long-term changes in activity in combination with the exposure to the CS of the presynaptic terminal through a form of Hebbian synaptic plasticity. A larger in depth study would be necessary to find the precise requirements for when and under which conditions activity in projections from the OT PVN neurons is able to induce immediate and maintained changes in spiking patterns in the CeA.

Besides the additional *Figure 6*, and *supplementary figures 5-8*, we have described this role for OT in SBF now more extensively in our results and discussion section of the revised version of the manuscript. (*lines 323-333*)

8. Is there evidence that OT neurons were indeed silenced and less OT was released during DREADD suppression?

To show that OT neurons were indeed silenced during DREADD suppression, we have performed whole cell patch clamp recordings in *in vitro* slices of the PVN on OTergic neurons that expressed hM4Di. We show in *new supplementary Fig. 5a* that application of CNO leads to a decrease in activation by injected current, thus providing evidence that they were indeed silenced. These *in vitro* findings with CNO are in line with the ones we obtained with a previous construct that expresses hM4Di OT-PVN neurons (Eliava et al Neuron, 2016, Figure S7).

We furthermore show, *in vivo*, (see *new supplementary figure 7*) that activation of these OT neurons after blue light leads to changes in CeA neuronal activity (increases in activity in Type 1 buffer neurons and decreases in activity of Type 2 fear neurons) that can both be significantly blocked by CNO and by oxytocin receptor blocker. It thus appears that CNO indeed leads to a block of OT release from these neurons and that less OT was released during DREADD/CNO-induced suppression.

9. In Figure 2, for the SBF "Companion" group, the freezing level is twice higher than for the SBF "Companion" group in Figure 1. They are supposed to be comparable.

We agree with the reviewer that absolute freezing levels for SBF can differ somewhat between experiments. One reason for this may be different endogenous oxytocin levels between animals. This corresponds with different neuronal baseline activities of Type 1 neurons that already increase during fear recall, possibly reflecting internal buffering of fear through a homeostatic mechanism. We now comment on this in the discussion (*lines 271-272*). On the other hand, more detailed analyses across experiments showed that SBF reduces freezing across experiments (Student's t-test,

$t=7.148$, $df=44$, $p<0.0001$), demonstrating the effectiveness of SBF. Also, when taken all the SBF results into account for a complete comparison between experiments, we found globally no statistical difference between freezing during SBF ($F(2,13) = 0.21$, $p = 0.82$). We therefore conclude that social buffering was successful across experiments, resulting in comparable, although not exactly similar, reduction of fear. Indeed, although across different experiments the absolute freezing and SBF levels could vary, within experimental designs we always found significant effects of SBF on the freezing levels where expected.

Minor:

1. Context-independence of the SBF-mediated fear extinction is a unique and important property. It is mentioned in Discussion but the supporting data is not shown in the main figures.

We thank the reviewer for drawing attention to the context-independence of the SBF-mediated fear extinction. All data of the main figures follow the same protocol as described in Fig. 1a, with a change in context for each session: Thus, Fear conditioning on Day 1 was performed in standard fear conditioning box, Fear recall on Day2 was performed in a type of home cage, Social buffering on Day 2 (three hours later) was conducted in a specially designed double cage and Retention of SBF on Day 3 was conducted in a hexagonal cage. We have emphasized this change in context for all situations in the results and material and methods (*lines 426-433*) and discussed this more elaborately in the revised version of the manuscript (*lines 386-392*).

2. The analysis SBF retention with CS2 must be included in the main Fig. Given that fear conditioning depends on CS frequency (<https://www.ncbi.nlm.nih.gov/pmc/articles/PMC7190350/>), it is essential to show the extinction does not depend on CS frequency.

We thank the reviewer for drawing our attention to this interesting consideration and for pointing us to the publication from Hersman et al. eLife 2020. In their publication Hersman exposed mice to a 7.5 kHz CS followed by white noise (or vice versa, both 75 dB), as well as CFC to 3 and 12 kHz tones (as sensitivity for 12 kHz is 100 times lower). They showed that after training, the white noise induced a flight response and the CS a freezing response, regardless of the order of training and that flight responses to 12 kHz were also higher than to 3 kHz. Indeed mice seem to exhibit more sensitivity to higher tones (16 kHz vs. 7.5 kHz, Koay et al., Hearing research 2002).

In our experiments, we obtained similar results across all sessions when we applied social buffering to the 15 kHz tone instead of the 5 kHz tone (supplementary figure 1C). It is possible that this is species related, as rats have a larger spectrum of hearing compared to mice (5-40 kHz versus a more restrictive tuning around 16 kHz for mice (see Heffner and Heffner, 2007).

With regard to the electrophysiological recordings, it is true that single unit recordings in the lateral amygdala of rats have shown more units tuned to higher frequency responses (around 20 kHz), which may be related to their native fear responses to the 20-25 kHz spectrum (see Bordi et al., J. Neurosci 1992). We believe our 15 kHz CS is sufficiently far removed from this spectrum as we did not observe any freezing responses to 15 nor 5 kHz exposure during habituation (before fear conditioning). Indeed single unit responses ranging between 5 to 15 kHz seem rather equally distributed in the Lateral amygdala (see Bordi et al., J. Neurosci 1992). Similarly, our ephys recordings in the central amygdala showed no differences, which may be further reinforced because the CeA is likely to have even broader receptive fields as this area is farther removed from the sensory system (see also Bordi, et al., 1992).

Following the question of the reviewer, we have added an additional series of experiments in which we used CS2 instead of CS1 as the tone that was coupled with SBF. We have included these in main Fig. 1 (added in *revised Fig. 1c*) and we have included the references in a more -extended text on this issue in the discussion of the revised version of the manuscript (*lines 258-261*).

3. The AAV titer information is missing.

We thank the reviewer for drawing our attention to this omission. We have added a table in the supplementary material showing the viruses that we used, their origin and titer.

Reviewer #3 (Remarks to the Author):

In this manuscript, the authors delineate the function of the neuropeptide oxytocin (OT) in social buffering of fear in rats. The authors hypothesize that OT release from the PVN into the central amygdala—a fear processing center with dense OT innervation and receptor expression—likely underlies this social support behavior. Using a series of pharmacogenetic manipulations and *in vivo* electrophysiological recordings, the authors conclude that OT transmission in the central amygdala is necessary for the anxiolytic effects of social buffering and that the circuit, extending from paraventricular hypothalamus to central amygdala, is fundamentally different from previously characterized fear extinction pathways involving the medial prefrontal cortex and surrounding amygdalar nuclei. Multiple aspects of study design, analysis, and interpretation of the data diminish the enthusiasm for the strength of current conclusions.

1. Here, animals must be individually housed for all behavioral experiments, to match housing conditions for animals with implanted electrodes. There is extensive literature on the stress associated with single housing of animals and associated neuropeptidergic mechanisms (Zelikowsky et al., 2018, Lee et al., 2021, Matthews GA, et al., 2016). This might preferentially increase the fear-reduction aspects of social buffering of fear (SBF) and its retention, characterized in Figure 1, as well as present potential confounds for oxytocin modulation. For the conclusions of this study it is important to demonstrate that single housing of animals does not alter SBF.

We thank the reviewer for raising this issue and for pointing us to this most interesting literature on Tac2, serotonin and dopamine during isolation stress. Considering that isolation stress can directly affect the central amygdala through these pathways, and particularly also fear memory consolidation (e.g. through Tac2, Andero et al., Neuron 2014), it is most important to address this issue. In fact, as Tac2 neurons in the CeM are important for such consolidation, one might even consider that the effects, that we found of oxytocin on the retention of social buffering, are mediated through a potential inhibition of these neurons (as oxytocin activates GABAergic projections from the CeL onto the CeM as we have shown previously (Huber et al., Science 2005; Viviani et al., Science 2011)). We thank the reviewer for drawing our attention to this literature.

With regard to effects of single housing on SBF, we had, in fact, for the initial pilot studies that led to the current protocol, first used rats that had been group-housed either with familiar (cage mates) or non-familiar cage mates (companion from a different cage). In these studies, we had used a contextual fear conditioning protocol with which we had found significant effects of fear buffering by the companion rat, without much apparent difference whether the companion was familiar or non-familiar (see figure and results discussed below under point 2).

When we started to combine behavioral observations with *in vivo* electrophysiological recordings, we had to change this initial protocol in a number of aspects and this included a cued fear conditioning to two tones and a change from grouped housing into single housing 3-5 days before starting the behavioral protocol (imposed by the implantations of cannulae, optical fibers and optrodes to avoid injuries by cage mates). Nevertheless, as reported in the present work, this still resulted in significant effects of SBF.

Following the reviewer's question (to "demonstrate that in our protocol single housing does not alter SBF"), we have now also tested (in our 2-tone, multiple-day SBF protocol) whether SBF similarly occurs in group-housed animals. The results of these additional group-housed findings have been plotted in **new supplementary Fig. 2a**. They show efficient SBF and its retention in group-housed animals, like we found in single-housed animals. It thus seems that the housing conditions that we

employed do not affect SBF in our protocol. We would like to mention that we worked with rats, not mice, and that the single housing for our animals was only imposed for 3-5 days before the behavioral experiments (contrary to the 2 weeks or more for the above mentioned studies). Furthermore (contrary also to above studies in which "all cage conditions remained otherwise identical" Zelikowsky et al., Cell 2018), we kept the single-housed rats in an enriched environment in order to reduce single-house stress and to compensate for the isolation according to precise legal requirements of the veterinary office of the canton of Vaud. These conditions included wood shelter, straw for nesting and hollow wood toy cubes to play and nibble.

We have added this information in the Material and Methods of the revised manuscript (**lines 413-415**) and the additional results in **new supplementary Fig. 2a** and we refer to these in the revised version of the discussion (**lines 255-258**). We thank the reviewer for drawing our attention to this interesting literature.

2. Generally, stressor levels are not well matched in conditions across the study, including combinations of isolation housing, novel conspecific introduction, etc.

We thank the reviewer for pointing out this apparent lack of matching conditions, which we think is rather due to our failure to mention or emphasize in our descriptions (which we have now corrected in the revised manuscript). We did test the influence of a number of stressor levels in this study that might be considered as potentially influencing the outcome. Among these were:

a. Isolated housing – see our response above

b. Gender differences: We tested both combinations of male-male and female-female rats as companions for social buffering. These results are shown in **supplementary Fig. 2b**. We did not find any significant differences in either type of configuration.

c. Novel conspecific introduction: In our initial SBF discovery in contextual fear conditioning of group-housed animals, we had also tested whether the exposure to the familiar (from the same cage) versus non-familiar (from different cage) companion would change the level of SBF. This did not seem to be the case, and we have, from then on, grouped the results from both experiments together.

The above recordings were made from animals that had been contextually fear conditioned in a traditional fear conditioning cage with a floor consisting of metal bars through which an electric current (0.5 mA, seconds) had been passed. The next day animal were placed back in the same box either alone or with, in the adjacent compartment, either a familiar or unfamiliar companion (non-conditioned) rat.

We are aware of the extensive literature on this topic (e.g. by Kiyokawa and Mori Behav Brain Res. 2014, and others), showing, dependent on the study, the all or not importance of familiarity of the companion with the experimental rat for the efficacy of social buffering.

In the new protocol companion rats were always obtained from different and separate cages, and were unfamiliar to the experimental rat to which they were never exposed twice, they were never exposed to the US and never before exposed to the CS. Special attention was paid to make sure companions were age- and weight-matched to the experimental rat so as not to cause any social stress or aggressive behavior.

In order to clarify this point in the manuscript, we have, in the material and methods section, more extensively described our protocol to show how we matched stressor levels in conditions across the study, including combinations of isolation housing, cleaning of the boxes in between sessions, how the companion was selected (never exposed to the US and not before SBF to the CS, matched in age/weight/sex to the experimental rat, time of introduction), prior handling of the experimental & companion rats for familiarization to the experimenter and maximal number of animals simultaneously tested per day across the different sessions (*lines 442-464*).

3. In the main body of results, the authors do not further clarify or explain the finding that TGOT application/ChR2 stimulation produces only an acute reduction in fear responses and does not play a role in retention of SBF. The authors should further expand on this and discuss that the more long-term effects of SBF likely rely on a form of observational learning component, whereas a single injection of TGOT (or OT stimulation) likely produces only a transient anxiolytic effect that does not actually contribute to the extinguishing of fear.

We thank the reviewer for bringing up this important question.

It is clear from our current results that the final decrease in response to the CS cannot be mimicked and explained only by an externally applied (TGOT) or internally increased (ChR2) activation of OT signaling in the CeA. Indeed, we have not found any internal signaling pathway (yet) whose direct stimulation is sufficient to reproduce both SBF and retention of SBF in the absence of the companion. Of interest here are findings of our colleagues (Brill-Maoz & Maroun, *Psychoneuroendocrinology* 2016) that also implicated the necessity of OTergeric activation of the mPFC for the retention of SBF. Interestingly, recent findings have shown that the presence of the companion can be encoded at the level of the mPFC (see Kingsbury et al., *Cell* 2018) and the communication between mPFC and amygdala plays an important role in both encoding and extinction of fear (reviewed in Triana del Rio et al., *Psychopharmacology* 2019).

While OT may also play a role in the mPFC in SBF, still other factors should be involved because our initial optogenetic experiments targeting all OT neurons in the PVN showed that even release of OT in all target regions of OT projections can still not substitute for the presence of the companion-induced retention of SBF (Fig. 3b). We favor a model in which OT signaling is paired with sensory signals that constitute an observational learning component, as the reviewer suggests. This pairing could lead to associative Hebbian-like plasticity that may underlie SBF retention. Future research should explore these issues to better understand which signaling parts of SBF are encoded in the communication between the mPFC and the BLA, and which other signals underlie the observational learning component of the presence of the companion (olfactory, auditory, visual).

In the revised version of the manuscript, we have explored this question further by an additional series of *in vivo* electrophysiology experiments including more neurons with acute responses to the CS1 and CS2. We studied the changes of these responses across the different behavioral sessions and their sensitivity to blue light and CNO reflecting OT activity. Based on these responses and sensitivities we identified two types of neurons that we present in new figure 6. Although they do not finally resolve the question about the "observational learning component" they do provide

deeper insight into the mechanisms that underlie the changes in neuronal responses underlying retention of SBF.

Following the suggestion of the reviewer we have, in the revised version of the manuscript, expanded on this question in the discussion section including the possibility of an observational learning component (*lines 351-357*).

4. Does increased social interaction (%time nose/nose and %time nose/wall) positively correlate with reduction of fear response? If so, active attention to conspecific may predict SBF expression on Day 3. This measure may also account for some of the individual variability present in the data.

We thank the reviewer for this interesting suggestion. Our further analysis of social interaction behavior (%time nose/nose and %time nose/wall) did indeed show a quite large variability in social behavior between individual animals. However, no clear correlation could be found with reduced freezing behavior to the CS during SBF (*new Supplementary Fig. 3d, black dots*). On the other hand, when we applied the OT-R antagonist OTA (which increased freezing and decreased social interaction, Supplementary Fig. 3c) and plotted the obtained values together with the previous values (see *new Supplementary Fig. 3d, green dots*) a correlation between freezing and social interaction emerged. However, we think this is rather caused by the direct effects of OTA on freezing behavior in the CeA of the fear-conditioned (FC) rats: Because OTA increases freezing behavior during SBF in the FC rats during their exposure to the CS, they will have less remaining time to engage in social interaction. Indeed, when we directly studied effects of OTA purely on social interaction under non-fearful conditions (during the habituation period before fear recall), OTA did not affect social interaction nor motivation (Supplementary Fig. 3a). We have included these data in the revised version of the manuscript as an *additional supplementary Figs. 3b-d*.

To assess whether active attention might predict SBF retention on Day 3, we analysed whether SBF levels on Day 2 could predict retention of SBF on Day 3: However, we found no correlation between individual levels on Day 2 and on Day 3 (see figure below).

Thus, it does not seem that likely that active attention to conspecific on Day 2 can predict SBF expression on Day 3. Furthermore, SBF expression levels on Day 3 do not show such a huge variability in most cases. Interestingly though, we did find that application of OTA (Supplem. fig. 3C) not only

decreased social behavior of FC rats, but also, conversely, increased social behavior in the companion rats. As mentioned above, we have previously found that OTA blocks the SBF-induced decrease of freezing in the experimental rat. It appears that the companion rats respond with an increase in social behavior which may be a reaction to compensate for increased freezing caused by OTA in the demonstrator rat (Supplementary Fig. 3c, +OTA, green vs white bar. Taken together, the two rats might operate in an emotional contagion, which would explain the large variability in social behavior but the much less variability in freezing levels (black dots, Supplementary Fig. 3d). We have emphasized these analyses in the results section (*lines 96-99*) and the conclusion (*lines 264-270*).

5. The authors used two different auditory conditioned stimuli CS1(5kHz) and CS2 (15kHz), subsequently fear conditioned by pairing each CS with an electric foot-shock of 0.5 mA on Day1. On day 2 recall of fear memory was assessed on to each CS, but the authors only measured SBF during CS1 re-exposure. Why was CS2 was not assessed in SBF, while for the retention of SBF assessment both CS1 and CS2 were used?

The reason for not assessing CS2 in SBF was that we wanted to use this second tone as an extra internal control to show that, within the same animal, non-exposure to SBF would not affect freezing to this tone on Day 3. To further control that the decreased response to CS1 on Day 3 was really caused by SBF and not merely by the re-exposure of the animal to CS1 on Day 2 (i.e. by extinction), we did a second control experiment in which we swapped CS1 and CS2 throughout the behavioral protocol. Thus, we re-exposed a separate cohort of animals to CS2 in the presence of the companion 3 hours after the first fear recall (instead of re-exposing CS1 in the presence of the companion). We have added this experiment in *new Fig. 1c*. This shows that the second exposure to CS1 on Day 2 did not lead to any significant decrease in freezing as compared to animals that had not been exposed a second time to a CS on Day 2 (no significant differences with responses to CS2 on Day 3). In comparison, the exposure to CS1 + companion on Day 2 significantly decreased the freezing response (Fig. 1b, Day 3). All these effects are independent of whether we paired CS1 or CS2 with the companion, as now shown in *revised Fig. 1*.

In the revised version of the manuscript we have added further explanation in the text, and we have also added a statistical comparison (in Figs. 1b and 1c, on Day 3, *lines 88-91*), to emphasize that there was no difference between the CS1 responses (replayed without companion on Day 2) compared to CS2 responses (without replay on Day 2).

6. Since the modulatory effects of Oxt can reconfigure circuits and alter behavior on the orders of days, the authors might consider expanding the time course of the social buffering retention. The authors perform the retention task just 48 hours after the initial training, which is not necessarily 'long-lasting'. This data may also further draw helpful comparisons between fear extinction—where reappearance of fear responses can occur via renewal, spontaneous recovery, or reinstatement shortly after conditioning— and social buffering of fear that may operate in circuits separate from extinction, as the authors posit, and therefore be a more persistent type of fear inhibition.

We thank the reviewer for this interesting suggestion regarding the modulatory effects of oxytocin to reconfigure circuits on the order of days. To answer this question (and also the questions 16 -18 as well as questions from the other reviewers – see below), we have conducted an additional series of electrophysiological *in vivo* experiments in which we also expanded the time course of the social buffering retention to a total length of 96 hours. We designed these experiments based on previous work published by colleagues in the field (e.g. Ciochi et al., Nature 2010 and Haubensak et al., Nature 2010) who studied neuronal activity in the CeA during auditory cued fear conditioning

(but without social buffering of fear). Their findings had revealed two types of neurons in the CeA that exhibit acute, opposite responses to a single CS after fear conditioning: they called these "CeLon" and "CeLoff" neurons. In our new findings, we have been able to identify a larger number of neurons with acute responses to both CS1 and CS2 and we studied how endogenous OT release, triggered by the presence of the companion, affected these acute responses. We included the CS2 as control, as it was never played in the presence of the companion. For these experiments we injected an additional number of 7 rats with a combination of viral constructs in the CeA and PVN in order to express Chr2 selectively in those OTergic PVN neurons that project to the CeA and to express inhibitory DREADD to allow inhibition by CNO of these CeA-projecting OTergic PVN neurons. After sufficient expression, we implanted optrodes in their CeA. We obtained a total of 146 recordings of single unit spiking activity of which 48 neurons showed distinctive (combined) responses to blue light and to the CS that could be divided into two groups:

Type 2 neurons (n=24) showed acute (transient) excitatory responses to the CS1 that characteristically increased after fear conditioning. These CS1 responses decreased during exposure to the companion ("SBF") and remained decreased the next day without the companion ("retention of SBF"). The development of these responses expand on the acute responses we had already found in previously recorded Type 2 neurons (Fig. 5). Responses to the CS2 became equally increased after fear learning but (contrary to the SBF-paired CS1) remained unchanged throughout all further sessions (as expected, since they were never evoked in the presence of the companion, see Day 3 and 4, **new Figures 6, a3&a4, g, and Supplementary Fig. 8**). Responses to optogenetic activation of OTergic projections from the PVN were mostly inhibitory (**new Fig. 6f**). We refer to this group of neurons as "**Type 2, Fear neurons**".

Type 1 neurons (n=24) were characterized by initially absent or slow, blunted inhibitory responses to the CS1 but, interestingly, these neurons started to develop acute, excitatory responses to the CS1 during exposure to the companion ("SBF") and these excitatory responses remained the next day without the companion ("retention of SBF", **new Figures 6, a1&a2, g and Supplementary Fig. 8**). Responses to blue light (optogenetic activation of OTergic projections from the PVN) of type 1 neurons were typically excitatory (**new Fig. 6f**). Responses of type 1 neurons to the CS2 (which was never paired with the presence of the companion) were absent or blunted and, contrary to their responses to the CS1, did not change across the different sessions. Because of their increased acute responses during the buffering of fear and their activation by OTergic projections from the PVN (as the type 1 neurons in Fig. 5) we refer to these neurons as "**Type 1, Buffer neurons**".

Taken together, these experiments show how OT release appears to modify the responsiveness

The ensemble of these new results is now shown in three new figures that illustrate the changes in responses to CS1, CS2 and blue light (**new main Fig. 6 and supplementary Fig. 8**) and responses in combination with CNO and BBB-permeable OT receptor antagonist L-368 899 (Tocris, Bristol, UK, **new supplementary Fig. 7**). As the reviewer may appreciate, our new findings show that excitatory neuronal (and concomitant freezing) responses to CS2 do NOT change (without exposure to the companion), whereas the excitatory neuronal (and freezing) response to the CS1 (with exposure to the companion) significantly decreased during SBF and remained decreased the following day.

Thus, through these experiments we are now able to more precisely attribute the changes in neuronal activity to social buffering, as they concentrate on the changes in CS-triggered activity instead of only on baseline responses that we previously assessed (old Fig. 5). These findings thus further demonstrate how endogenous release of oxytocin, triggered by the presence of the companion, can indeed reconfigure the circuitry in the CeA that involves the type 1 and type 2 neurons (as suggested by the reviewer)

We also tried to assess the development of these electrophysiological measurements over a longer period of time. Unfortunately, over a period of many days, we progressively lost units, and it appeared difficult to keep stably tracking them all. As a result, the differences in responses to CS1 and CS2, although a tendency persisted, lost significance on Day 4 (see new figure 6g). A full comparison with mechanism of fear extinction remains indeed technically challenging with this experimental approach. On the other hand, we would like to point out that, in the behavioral experiments, re-exposure to the CS1 leads to an immediate reduction of freezing, right from the very first re-exposure to the CS (Supplementary Fig. 1, Day 2 SBF). Extinction, on the other hand, typically only develops over time, after repeated exposure to the CS. At the same time, we acknowledge that our study has not been aimed at discovering to what extent the underlying neuronal mechanisms are completely different from fear extinction. Indeed it is possible that, notably the maintained, reduction of freezing responses is relying on extinction mechanisms (as also mentioned above regarding the importance of OT signaling in the mPFC. In order to distinguish to what extent these two processes are fundamentally different, one would have to start to examine whether renewal, spontaneous recovery, or reinstatement shortly after SBF are still possible and subsequently study in more detail the involvement of projections to and from the mPFC that have been implied in extinction processes. These experiments will be of interest to conduct in a future project.

In the revised version of the text, we have changed the reference of "long-lasting" to "maintained" (reduction of freezing to CS1 on Day 3) or, according to context, "next-day maintained" or "SBF retention" as more accurate. In addition, we have added text to the discussion regarding the relation between SBF and extinction (*lines 379-405*).

7. Similarly, another potential internal control for the validation of the paradigm would be to have a cohort of animals re-exposed to both auditory cues during SBF induction, but only a companion is present during one auditory cue (while the other cue is replayed without companion). The authors could then compare the fear reduction caused by extinction (repeated exposure to stimulus without foot shock) and that caused by presence of an untrained conspecific to provide further mechanistic clarity into the two forms of fear reduction. This control would also help with the interpretation of fig s3 and fig 6, which lack a 'no companion' internal control.

We thank the reviewer for this suggestion. We had indeed considered to include this potential internal control in our experiments, but the eventual experimental design would have made the final protocol quite lengthy with additional changes that could lead to potential confounding effects. Notably, in order to expose animals to both auditory cues during SBF induction on Day 2, e.g. CS1 with companion and (for the same animal) CS2 without companion, would have implied an additional series of exposures on Day 2 (that is, first recall to both CS1 and CS2 followed three hours later by first an exposure to CS1 with companion, then remove and replace companion with fur animal before exposure to CS2).

Instead, we decided to make a comparison between two different cohorts of animals, one cohort with and the other cohort without companion + exposure to CS1 on Day 2 (3 hrs after fear recall). And we conducted a similar experiment in which we swapped for both cohorts CS2 for CS1, using instead CS2 for exposure on Day 2 with the companion (3 hours after fear recall). To facilitate the comparison between these two different series of experiments, we have now placed them both in revised Fig. 1 (resp. Figs. 1b and 1c). As the reviewer may appreciate, the only difference we could find on Day 3 was the response to the CS that had been paired with the companion. Responses to the different CSs (re-exposed on Day 2 without companion and to the CS that was not played on Day 2) were all similar. We have indicated this in the *new Fig. 1c* as requested by the reviewer in the question below.

8. Line 121, the authors mention that Day 3 CS1 + no companion group was “similar to CS2” in Fig. 1B. Please provide statistical evidence for this comparison.

We have run a Mann-Whitney test on the two CSs without SBF on Day 3 in Figs. 1b and c (the red against the blue solid bars in the retention of SBF).

Fig. 1b: Two-tailed Mann-Whitney, $U=13$, $p=0.4848$, $n=6$

Fig. 1c: Idem, $U=12$, $p=0.3939$, $n=6$

Thus, they are far from significantly different.

We have added this statistical analysis in the new version of the manuscript (*lines 89-91*)

9. Overall, the authors have devised a paradigm that might offer the opportunity to compare SBF-induced fear reduction and fear extinction but do not perform these types of analyses. Also, considering that TGOT/OT stimulation in CeA produces only a transient effect in fear reduction, it is unlikely that SBF is mediated through a pathway “distinct from mPFC-BLA pathway” as mentioned in the introduction and discussion. Cortical activity and BLA are likely needed for the prolonged fear reduction and therefore the authors cannot confirm that PVN-CeA pathway is sufficient for SBF, as mentioned in discussion (line 302), so all statements mentioning the ‘independence’ of PVN-CeA pathway for SBF or how SBF is ‘completely subcortical’ should be moderated. The study provides no piece of evidence that BLA is not involved in the used SBF behavioral paradigm.

We agree with the reviewer and we have therefore moderated all statements mentioning the “independence” of PVN-CeA pathway for SBF or how SBF is “completely subcortical”. For further clarity, we have added references to other studies that implied the mPFC in SBF mechanisms and we have added further text to the discussion (*lines 380-392*).

However, we did find that OTA blocks retention of SBF on Day 3 (Fig. 2b) which indicates that endogenous OT is necessary for retention of fear reduction on Day 3. Although other regions may also be involved for producing retention of SBF, our previous findings already suggested an important role for the PVN-CeA pathway in memory of fear encoding (Hasan et al., Neuron 2019). This interesting phenomenon was referred to as “memory engram in oxytocinergic system”, and might contribute to reinforcement of positive emotional states (Wahis et al., Nat. Neurosci. 2021). In summary, these findings open up an alternative mechanism besides the conventional repetitive adaptation-induced extinction process and reveal a neuropeptidergic mechanism of social induced fear decrease, whose role in classical fear extinction is still to be assessed.

10. In fig 5 or supplemental fig 6, authors should provide more data related to waveform/spike sorting and PCA analyses that led to derivation of only three neuron types (e.g.% variance plots, k-means clustering). Additionally, the authors mention in line 455 of methods that neurons were selected for recording based on responses to CS1/2, however, the only type 2 neurons appear to show a distinct, time-locked response to presentation of auditory cue. Authors should clarify this experimental criterion further and explain how they ‘manually’ defined clusters as opposed to alternative automated methods.

We thank the reviewer for raising this issue, which revealed to us a potential point of confusion. In fact, the derivation of three neuron types in Fig. 5 was NOT based on their waveforms/spike sorting, but on the development over their baseline firing frequencies across the four different behavioral sessions. To illustrate this process better, we have now added a clustering analysis of the different developments of baseline spiking patterns across the four behavioral sessions under Fig. 5c, on which we based the classification into three different neuronal types.

Furthermore, and also in answer to the reviewers questions 16-18, following our previous findings of occasionally acute responses to the CS in Type 2 neurons (Fig. 5), we have conducted a new and more extensive series of electrophysiological *in vivo* experiments in which we also studied effects of BL stimulation of OT release in the CeA. In this new series, we recorded activity of 146 single units from 7 animals in which we were able to identify a substantially larger number of time-locked responses in type 2 as well as type 1 CeA neurons (plotted in **new Fig. 6 and new supplementary Fig. 8**). And, through their different responses to blue light (activation, inhibition or no responses), we had an additional criterion to classify these neurons into different types. These allowed us to make more precise observations about the development of their responses to CS1 and CS2 across the different sessions and classify these neurons into "Type 1, Buffer" and "Type 2, Fear" neurons. (**new Fig6f**) In the new version of the manuscript, we have included these new findings and we have for the previous electrophysiological recordings added a panel (**new Fig. 5c**) to clarify how we defined these types based on baseline changes using cluster analysis (legend of Fig. 5c, **lines 625-633**). We have also added a **new Fig. 6 & supplementary Figs. 5-8**) that show the new series of single unit recordings of CeA type1 and type2 neurons that were sensitive to BL and respond differently to CS1/CS2 exposures.

11. Considering that the authors are able to track neural activity of units reliably over the course of days, it would be helpful that bar plots to the right in Fig 5, showed paired data in order to depict neural activity changes over the course of the behavioral paradigm for each neuron type (as opposed to grouping all the data together).

Following the suggestion of the reviewer, we have now indicated paired data (through connecting lines) in the bar plots to the right of **Figs. 5a&b**. We thank the reviewer for this suggestion, because it also allowed us to answer another question, namely to what extent the same individual neurons that change activity during fear recall also change (or not) activity during SBF. Indeed it answers the interesting question whether the coding for fear learning and social buffering of fear takes place in different neurons (but in the same ensemble) or in the same neurons. This was also a question of reviewer 2.

To illustrate our analyses, we have drawn, in the **revised version of Fig. 5a&b**, lines between individual dots to track spike rates of the same individual cell across different sessions. As can be seen, neurons that increased activity during fear recall, showed additional increases in baseline activity during SBF i.e. there are "additive/cumulative" effects on OT neuronal firing. This suggests that the activated PVN neurons are sensitive both to internal as well as external signals, i.e. that the very same OTergic neurons that play a role in encoding the "internal buffering" of the fear response (maintaining homeostasis after fear conditioning) also are activated by the external signals that induce the social buffering of fear.

Besides the modification of figure 5, we have added a text in the discussion of our revised manuscript to address this point (**lines 278-281**). We thank the reviewer for raising this most interesting question which has allowed us to deepen our initial message.

At the same time, in the new recordings that we obtained (as described in **New Fig 6**) we also found acute activity responses to the CS 1 and 2 that differentially changed for type 1 and type 2 cells across the behavioral sessions. Based on these new findings we can formulate the following responses to the questions of the reviewer: i) that the (baseline) activity differences across sessions may represent changes that are mainly mediated by activity in OTergic projections from the PVN. ii) that fear conditioning initially increases some OT release (as an initial response to the fear conditioning to keep homeostasis) leading to an increase in baseline activity in Type 1 Buffer neurons. iii) During SBF, as a result of the presence of the companion, OTergic PVN neurons are

strongly activated and cause further increase baseline activation of Type 1 buffer neurons. Considering these neurons can make local GABAergic synapses in the CeA, this may lead to a concomitant inhibition of baseline activity of the Type 2 Fear neurons.

We have included further text on this most interesting point in the discussion of the revised version of the manuscript (*lines 320-333*).

12. In addition, tracking same units over time should be shown for more than a selected example. How consistent is this? Of note – it is unusual to be able to stably track units for many days, so failure to do so is acceptable. However, we just do not have enough QC info on the current dataset.

To answer this question of the reviewer, we have added illustrations in *new Supplementary Fig. 6* that show for several examples of how we identified and traced the same units over time and how stable this tracking was.

To ensure maximum quality of the single unit recording, we implanted tetrodes 500-600um above the CeA and propelled these by a microdrive 100-200um per day until we reached the target coordinates (about 7 days before start recording). This approach will largely exclude rapid units loss at the initial implant site caused by the inflammation of acute implantation. In each section, we applied PCA analysis of units sorted in tetrodes mode (*new Supplementary Fig. 6, d1-d3*), tracing the autocorrelation score to check the sorting quality (e1-e3), and perform waveform tracking in every 5min time bin (*f1-f3*). In addition to waveform and spike pattern analysis, we also used the opto-tagging approach to increase the success rate of identifying the same unit, as OTR interneurons will respond robustly to OT axonal release evoked by blue light (See Tang et al. Star Protocol 2022) In *new Fig.6 b-e*, we show two examples of Fear and Buffer neurons in the CeA, of which waveforms had large amplitude voltage signal close to 1 mV and were consistent across full sections.

We agree with the reviewer, however, that single units tracing across days is indeed difficult. To illustrate this process, we have plotted the number of units lost over the different sessions of the behavioral protocol in *new Supplementary Fig 6h* and we have also included an example of failure of unit tracing in *New Supplementary Fig. 6f3*.

13. In line 199, authors state “Type 1 neuronal changes correlated closely with PVN neuronal activity”, but do not provide any concrete correlational analyses.

To address this question, we first performed a cluster analysis of action potential frequency changes across sessions in CeA neurons. It turned out that a division of cells into three clusters explained 90% of the variability in the observed spiking changes, thus confirming the identification of the three cell types. Second, we plotted the action potential frequency changes of the hypothalamic cells in these three clusters, and found that they could be allocated to the Type 1 cluster. This additional cluster analysis is now shown in *new Fig. 5c*.

To further provide Spike count correlation, we calculated Pearson’s correlation coefficient in Excel. We first generated the average firing rate of PVN and CeA types across four sections (Hub, FC, SBF, Retain) into rows of data, then use Formula =“CORREL(raw1, raw2)” to get the correlation coefficient factors between cell types as below:

PVN	TYPE1	TYPE2	TYPE3
PVN	0.9831013	-0.1349268	0.50062016
	TYPE1	-0.99601222	-0.24282305
		TYPE2	0.15530799
			TYPE3

We can see the PVN and CeA Type1 are highly correlated, while CeA Type1 and Type2 are highly negatively correlated. (see also Cohen et al. Nat.Neurosci. 2011)

14. In line 201-203, authors state that Type 1 neuron activity “derives directly from OT-sensitive CeA neurons”. This finding is not currently supported by any data provided. There is no way to exclude numerous indirect circuit effects in these experiments

We thank the reviewer for raising this point, we agree that our statement that:

199 Type 1 neuronal changes correlated closely, throughout all sessions, with PVN
200 OTergic neuronal activities described above (Fig. 5b, Hab: 4.7 ± 1.5 Hz, FR:
201 18.9 ± 4.8 Hz, SBF: 32.7 ± 6.4 Hz, retention of SBF: 17.6 ± 3.6 Hz). This suggests this
202 activity derives directly from OT-sensitive CeA neurons as previously identified in
203 CeA^{24,30}.

even if we brought it forward as a "suggestion", is not supported by showing a direct connection between these PVN and CeA activities. We think, however, that our new series of *in vivo* electrophysiological experiments (already mentioned above) provide data to better support this statement:

In these *in vivo* electrophysiology experiments we expressed Chr2 and hM4Di selectively in projections from the PVN to the CeA in an extra series of rats. In 7 rats we were able to identify a total of 146 single units among which we found 48 units with excitatory or inhibitory responses to blue light, that showed characteristic changes in acute responses to CS1 and CS2 throughout the behavioral protocol that allowed us to classify them in two types of neurons. We have focussed on these acute changes to describe how Type 1 neuron activity “derives directly from OT-sensitive CeA neurons” by combination of CNO and OT antagonist applications.

1. BL excitation of Chr2 expressing OTergic CeA projections induced changes in CeA spiking patterns:

The blue light exposure in the CeA allowed us to distinguish between two types of changes in spontaneous local spiking activities: 1) neurons that increased their frequency 2) neurons that decreased frequencies (besides neurons that were not affected by blue light).

(see **new supplementary Fig. 7**)

2. Both CNO and OTR antagonist L-368 899 are effective in blocking both types of changes

By co-expressing hM4Di in the OTergic neurons projecting from the PVN to the CeA, we could show that the BL-evoked responses (inhibitory as well as excitatory) could be significantly reduced by BBB permeable OT receptor antagonist L-368.899 and completely blocked by intraperitoneal injections of CNO. This shows that CNO effectively blocks activity of OTergic neurons with projections in the CeA (**new supplementary Fig. 7a2,a3, b3&c3**).

3. Changes in acute responses after fear conditioning and after SBF

Within these 146 neurons, we classified neurons in two groups based on their development of acute responses to CS stimulations across different sessions (as described above in our answer to question 6). We found that Type 1 and Type 2 neurons demonstrated similar changes in acute responses to the CS as found by our colleagues in respectively CeLoff and CeLon neurons of the central amygdala after fear conditioning (Ciocchi et al., and Haubensak et al., Nature 2010a,b)

4. SBF-induced changes in acute responses can also be blocked by CNO.

After exposure to SBF, we then administrated CNO, to observe whether it could reverse the changes in spiking activity that accompanied the behavioral changes to CS1 and CS2. As can be

seen in **new Figs. 6f&g**, the increase in acute responses to CS1 in type 1 neurons after SBF could also be blocked by administration of CNO as could the decreased acute responses of type 2 neurons to the CS1.

Together, we think these findings show more conclusively, that "*Type 1 neuron activity change (in this case: the acute response to CS1) derives directly from OT-sensitive CeA neurons*".

At the same time, we realize that our new experimental results also raise new questions. Thus, one can wonder whether a sustained activation of OTergic PVN neurons after SBF is necessary to maintain the changes in the CeA or whether their transient activation (during SBF) is sufficient for a sustained induction of changes. Indeed, it is possible that a one-time increase in release of OT in the CeA (triggered by SBF) can induce long-term changes in activity in combination with the exposure to the CS through Hebbian plasticity mechanisms. A larger, in depth, study would be necessary to find the precise requirements for when and under which conditions activity in projections from the OTergic PVN neurons is able to induce immediate and maintained changes in spiking patterns in the CeA.

In the revised version of the manuscript we have added **new figures 6 and new supplementary figures 5-8**. In the revised discussion we have inserted text to clarify how we think, based on these results, that the "observed changes in spiking patterns in the CeA resulted from increased activity in direct projections from spiking OT PVN neurons to the CeA" (**lines 324-333**).

15. It also suggests that PVN-CeA circuit is primarily glutamatergic (i.e. that PVN input is directly eliciting activity in CeA neurons) in which there is conflicting evidence in the field that OT neurons can release glutamate as opposed to a primary neuromodulatory role in the brain.

Our current *in vivo* as well as earlier *in vitro* published data suggest that the PVN-CeA circuit is primarily oxytocinergic with a remaining component of glutamate.

Thus, in our previous *in vitro* electrophysiological recordings, we have shown that blue light activation of OTergic PVN projections in the CeA was able to increase spontaneous spiking activity of CeA neurons located in the lateral part of the CeA and increase inhibitory postsynaptic currents in neurons located in the medial part of the CeA (presumably through GABAergic innervations originating from the latter). These changes in neuronal activity in the CeA could be significantly reduced by OT receptor antagonist perfusion with a small remaining component that was fully blocked by addition of the AMPA receptor blocker NBQX (Knobloch et al., Neuron 2012, Fig. 4E2).

These findings correspond with our new *in vivo* recordings in which we activated spiking activity in a group of CeA neurons after blue light activation of OTergic PVN projections in the CeA (**new Supplementary Figure 7a1**). Oral administration of the OT antagonist L-368 899 significantly reduces this BL-evoked activation, and CNO further fully blocked the BL-evoked activation. This suggests that indeed, there is a co-release of another substance upon BL activation, and our *in vitro* experiments have shown the co-release of glutamate. To test this further *in vivo*, we would need to apply an antagonist of glutamatergic transmission onto AMPA receptors similar to above-described experiments *in vitro*. *In vivo*, however, NBQX causes too many side effects on other regions of the brain to furnish any further interpretable results. Indeed, our present *in vivo* preparation is not ideal for in depth pharmacological studies of the co-release of transmitters.

In the revised version of the manuscript, we have added these findings that indicate the glutamatergic involvement, as well as our previous published results (**lines 301-303**)

16. While it is compelling that the authors found a neuron group that increased activity during the retention of SBF, it would be more significant if the increase in activity was not so generalizable and time-locked to the presentation of the CS. Is this a feature to be expected for maintained inhibition of CeLON neurons (as mentioned in lines 275-76) or should the activity (and therefore posited inhibition of lateral amygdalar neurons) be more time-locked to the conditioned stimulus?

We thank the reviewer for this interesting point. To answer this question (and the previous question 6 of the reviewer), we have conducted (as mentioned above) an additional series of experiments with more recordings of single units and in which we also tested in the CeA for responses to BL-activated neuronal projections from OTergic neurons PVN neurons (see also above, and **new Fig. 6**) throughout the whole original protocol of SBF.

Among these, we were able to identify more time-locked responses to the CS presentations. Interestingly, these responses showed a development across the different behavioral sessions that could be grouped into two different types: CeA neurons whose time-locked activations to the CS1 and CS2 became larger after fear conditioning, suggesting they encode the fear learning, similar to the CeLON neurons of Haubensak et al. (Nature 2010) and Ciochi et al. (Nature 2010)). During SBF, their CS1 response was significantly reduced and this decreased CS1 response remained decreased during the retention of SBF. CS2 responses (not paired with the SBF) remained increased throughout the sessions. These new results are now plotted in **new Fig. 6**. These neurons were often inhibited by blue light and the development of the baselines activity of these neurons across the different sessions corresponded most closely to the type 2 neurons in Fig. 5. We henceforth refer to these neurons as "Type 2, Fear neurons".

As mentioned above (see question 11), we also identified CS time-locked responses in a second group of neurons, which we refer to, based on the development of their baseline activities across the behavioral protocol, as Type 1, Buffer neurons. These neurons were typically excited by BL, but showed initially no or weak responses to the CS1 or CS2. Interestingly, though, only during and after SBF exposure, did these neurons start to respond to the CS1 (not the CS2), as if they derived now a safety signaling from the CS1 stimulus. In view of this feature, we henceforth refer to these neurons as "Type 1, Buffer neurons" and they may correspond with the CeLoff neurons of Haubensak and Ciochi (Nature, 2010). We interpret these findings as that a synaptic strengthening of auditory input from CS1 has been associated with the presence of the companion during the fear recall. As a result CeLoff neurons (instead of CeLON neurons) now start to respond to the CS1. It is possible that this plasticity also involves the lateral amygdala, but we believe that OT receptors in the CeL play a role in the formation of this memory.

We have included these new findings through **new Fig. 6, and new Supplementary Figs. 6, 7 & 8** in the revised version of the manuscript.

17. The behavioral analysis depicted in fig 4e also shows low subject number, but it appears in fig 5 that the authors have more animals/data to contribute to this dataset. Is there an explicit reason why the authors have a low n for fig 4e?

For figures 4 and 5 we recorded single unit in total number of three animals. The behavioral responses of these animals are shown in Fig. 4e. In Fig. 5, the individual data points represent single unit recordings, not number of animals. For **new Fig. 6**, representing a new series of electrophysiological experiments, we recorded from an additional 7 animals. We have now better indicated in the legends of these figures the number of animals, to prevent misunderstandings.

18. Molecular identification for functionally defined amygdala neurons modulated by OT would be very useful for the resulting overall model of OT effects on SBF.

We fully agree with the reviewer. As we are working with rats, we do not have the molecular tools to directly identify these neurons in the central amygdala by their expression of e.g. fluorescent markers (in contrast to mice, see Ciochi et al. Nature 2010; Haubensak et al., Nature 2010). However, by expressing ChR2 specifically in OTergic projections to the CeA with a viral approach, we have now indeed been able to functionally define amygdala neurons based on their differential responses to blue light (by axonal opto-tagging, Eliava et al., Neuron 2016, Knobloch et al., Neuron 2012) We hope the reviewer will appreciate our new series of electrophysiological *in vivo* experiments that are based on this tool. Taking everything together, we therefore speculate the type 1, buffer CeA neurons were PKC delta neurons that expression OTR (P. Shrestha et al., Nature 2020) and type 2, fear CeA neurons were SOM interneurons (Kai Yu et al., Bo LI, J. Neurosci. 2016)

Minor points:

19. Figure 2b: Day 2 (right) plot should have 'SBF' (plot under light purple color) instead of "TGOT" similar to Fig 2a.

For this particular experiment, we replaced the "presence of the companion" ("SBF") by the administration of TGOT in the CeA, so there was, in fact, no SBF occurring. TGOT leads to a decrease in freezing to similar in extent to SBF during its administration, but with the important difference that the next day, no memory effect occurs. It thus seems that TGOT cannot fully replace the effect of SBF. To illustrate this better we have, in the revised version of the manuscript, added in figure 2c "instead of SBF" besides the TGOT label.

20. Generally, confirmed expression patterns from DREADD and ChR2 experiments should be shown.

To answer this request we show now in **new supplementary Fig. 5** examples of immunostainings illustrating the expression of inhibitory DREADD hM4Di and the blue light-sensitive ChR2 in PVN neurons and their overlaps with OT expression. In this figure we also illustrate how CNO can inhibit this neuronal activation in whole-cell patch clamp recordings and how projections from these ChR2 expressing OTergic PVN projections in the CeA can evoke local responses as expected (see also above). We would like to add that we previously published an extensive, histological and functional characterization of the viral expression of both constructs in Eliava et al., Neuron 2016, respectively Knobloch et al., Neuron 2012.

21. Figure 4 d: The images are arranged in a confusing manner. Areas marking BLA and CeA are a bit off than normal. A full section image would be ideal.

We thank the reviewer for signaling this confusing manner. In the new version of the manuscript we have rearranged these images and we have added a more accurate marking of BLA and CeA (**revised figure 4d**).

22. The data points on all histograms and plots are too small

In the revised version of the manuscript, we have enlarged the data points on all histograms and plots.

23. In each Fig. CS1 (5kHz) and CS2(15 kHz) are designated as Experimental and Control respectively. It is mentioned that CS1 and CS2 are for internal comparisons, however, designating them as Experimental and Control is confusing.

To avoid this confusion, we have now removed the designation of "Experimental" and "Control" in the figures.

24. Include more human lit in the introduction, such as Bratec et al 2020

We thank the reviewer for this suggestion. We have added the reference of Mulej Bratec et al., 2020 and described their interesting finding in relation to our study in the introduction.

25. Figure 4c: the y axes of all the plots are not starting from zero

Following the suggestion of the reviewer, we have added the zero axis on the **revised Fig.4c**. In order to show the spreading of the individual data points, we have added a break in the axis instead.

26. Figure 4e: needs more data points (error bar of "Pre-Tone" in SBF is extreme). "Hab" should be replaced with "Habituation" to maintain the uniformity across the paper. Same is in Figure 5 a,b,c (histograms)

As mentioned above, we have increased our observations in a new series of *in vivo* electrophysiology experiments in 7 rats now shown in **new Fig. 6** and **new Supplementary Fig. 8**. These show also more behavioral responses adding to the observations of Fig.4e. We decided to keep them apart because of different implant sites. We have changes "Hab" into "Habituation" in both figures 4 and 5

27. Figure 6 seems out of order and should likely be combined with Fig. 2, as both Figures are related to how oxytocin may regulate the acute and/or long-term effects of SBF.

Following the suggestion of the reviewer we have combined both figures. In the revised version of the manuscript, Fig. 6 has now become **Fig. 2b**.

References:

Herry, C., Ciocchi, S., Senn, V. et al. Switching on and off fear by distinct neuronal circuits. *Nature* 454, 600–606 (2008). <https://doi.org/10.1038/nature07166>

Shrestha, P., Shan, Z., Mamcarz, M. et al. Amygdala inhibitory neurons as loci for translation in emotional memories. *Nature* 586, 407–411 (2020). <https://doi.org/10.1038/s41586-020-2793-8>

S. Duvarci, D. Popa, D. Pare, Central Amygdala Activity during Fear Conditioning. *J Neurosci.* 31, 289–294 (2011). DOI: <https://doi.org/10.1523/JNEUROSCI.4985-10.2011>

M. T. Hasan, F. Althammer, M. S. da Gouveia, S. Goyon, M. Eliava, A. Lefevre, D. Kerspern, J. Schimmer, A. Raftogianni, J. Wahis, H. S. Knobloch-Bollmann, Y. Tang, X. Liu, A. Jain, V. Chavant, Y. Goumon, J.-M. Weislogel, R. Hurlemann, S. C. Herpertz, C. Pitzer, P. Darbon, G. K. Dogbevia, I. Bertocchi, M. E. Larkum, R. Sprengel, H. Bading, A. Charlet, V. Grinevich, A Fear Memory Engram and Its Plasticity in the Hypothalamic Oxytocin System. *Neuron*. 103, 133-146.e8 (2019). <https://doi.org/10.1016/j.neuron.2019.04.029>

K. Mikami, Y. Kiyokawa, Y. Takeuchi, Y. Mori, Social buffering enhances extinction of conditioned fear responses in male rats. *Physiol Behav*. 163, 123–128 (2016). <https://doi.org/10.1016/j.physbeh.2016.05.001>

T. Grund, Y. Tang, D. Benusiglio, F. Althammer, S. Probst, L. Oppenländer, I. D. Neumann, V. Grinevich, Chemogenetic activation of oxytocin neurons: Temporal dynamics, hormonal release, and behavioral consequences. *Psychoneuroendocrinology*. 106, 77–84 (2019).doi: 10.1016/j.psyneuen.2019.03.019

Y. Tang, D. Benusiglio, A. Lefevre, L. Hilfiger, F. Althammer, A. Bludau, D. Hagiwara, A. Baudon, P. Darbon, J. Schimmer, M. K. Kirchner, R. K. Roy, S. Wang, M. Eliava, S. Wagner, M. Oberhuber, K. K. Conzelmann, M. Schwarz, J. E. Stern, G. Leng, I. D. Neumann, A. Charlet, V. Grinevich, Social touch promotes interfemale communication via activation of parvocellular oxytocin neurons. *Nat Neurosci*. 88, 127–13 (2020).doi: 10.1038/s41593-020-0674-y

M. L. Smith, N. Asada, R. C. Malenka, Anterior cingulate inputs to nucleus accumbens control the social transfer of pain and analgesia. *Science*. 371, 153–159 (2021).10.1126/science.abe3040

X. Zhang, J. Kim, S. Tonegawa, Amygdala Reward Neurons Form and Store Fear Extinction Memory. *Neuron*. 105, 1077-1093.e7 (2020).doi: 10.1016/j.neuron.2019.12.025

1. J. Wahis, D. Kerspern, F. Althammer, A. Baudon, S. Goyon, D. Hagiwara, A. Lefevre, B. Boury-Jamot, B. Bellanger, M. Abatis, M. S. da Gouveia, D. Benusiglio, M. Eliava, A. Rozov, I. Weinsanto, H. S. Knobloch-Bollmann, H. Wang, M. Pertin, P. Inquimbert, C. Pitzer, J. Siemens, Y. Goumon, B. Boutrel, P. Darbon, C. M. Lamy, J. E. Stern, I. Décosterd, J.-Y. Chatton, W. S. Young, R. Stoop, P. Poisbeau, V. Grinevich, A. Charlet, Oxytocin Acts on Astrocytes in the Central Amygdala to Promote a Positive Emotional State. **103**, 133–76 (2020).

K. Yu, P. G. da Silva, D. F. Albeanu, B. Li, Central Amygdala Somatostatin Neurons Gate Passive and Active Defensive Behaviors. *J Neurosci*. **36**, 6488–6496 (2016).

M. Eliava, M. Melchior, H. S. Knobloch-Bollmann, J. Wahis, M. da S. Gouveia, Y. Tang, A. C. Ciobanu, R. T. del Rio, L. C. Roth, F. Althammer, V. Chavant, Y. Goumon, T. Gruber, N. Petit-Demoulière, M. Busnelli, B. Chini, L. L. Tan, M. Mitre, R. C. Froemke, M. V. Chao, G. Giese, R. Sprengel, R. Kuner, P. Poisbeau, P. H. Seeburg, R. Stoop, A. Charlet, V. Grinevich, A New Population of Parvocellular Oxytocin Neurons Controlling Magnocellular Neuron Activity and Inflammatory Pain Processing. *Neuron*. **89**, 1291–1304 (2016).

D. Viviani, A. Charlet, E. van den Burg, C. Robinet, N. Hurni, M. Abatis, F. Magara, R. Stoop, Oxytocin selectively gates fear responses through distinct outputs from the central amygdala. *Science (New York, N.Y.)*. **333**, 104–107 (2011).

S. A. Josselyn, P. W. Frankland, Fear Extinction Requires Reward. *Cell*. **175**, 639–640 (2018).

M. R. Cohen, A. Kohn, Measuring and interpreting neuronal correlations. *Nat Neurosci*. **14**, 811–819 (2011). [10.1038/nn.2842](https://doi.org/10.1038/nn.2842)

REVIEWER COMMENTS

Reviewer #1 (Remarks to the Author):

The authors responded to the reviewers' comments exceptionally well. In my view, the manuscript is acceptable for publication in its present form.

Reviewer #2 (Remarks to the Author):

The authors addressed most of the concerns, but one crucial issue remains. In their reply to comment #3 authors argue that the contrast between responses to CS1 and CS2 in Fig. 6 can serve as a "no partner control" for the unit analysis. The counterargument is that animals received more exposure to CS1 than CS2. As a result, the levels of extinction of the CS1-US and CS2-US associations differed when units' responses to CS1 and CS2 were compared. Therefore, the conclusions about the effects of SBF on unit activity remain questionable.

In addition, in Fig. 1 and 2, many groups include only 5-6 subjects, and the freezing variance is high. The experiments appear underpowered.

Minor:

In all diagrams with behavioral data, please connect the dots representing the same animal along the time course.

Reviewer #4 (Remarks to the Author):

Hegoburu et al. did a great deal of work to respond to reviewer comments, which is applauded and appreciated. The findings are of importance to the field, and the experiments appear well performed thanks to the authors' inclusion of additional details in the methods and by the additional control experiments they have performed. Social buffering of fear is an interesting and important phenomena, and this work provides evidence that the oxytocin-positive neurons of the PVN play an important role in it

There are, however, still concerns that remain not fully addressed, as well as concerns raised by some of the new experiments the authors have performed. Of particular concern are interpretation of the hM4Di + blue light control experiment, the methods and rationale for categorizing CeA neurons in Figure 5 into 3 types, and the sparseness of the evidence underlying the claim that type 1 neurons from Figure 5 are

the same type of neuron as type 1 neurons from Figure 6 (and that type 2 neurons from Figure 5 are of the same type of neuron as type 2 neurons from Figure 6).

Excerpt from overall reviewer comments: Multiple aspects of study design, analysis, and interpretation of the data diminish the enthusiasm for the strength of current conclusions.

In a new control experiment, the authors inject hM4Di and shine blue light. They then claim (lines 160-161, figure S4C) “BL had no effect in control rats infected with a hM4Di expressing AAV”. This claim does not appear as certain as the authors’ statement.

First, the hM4Di figure says ‘ChR2’ on the figure itself, above the data, creating confusion.

Second, hM4Di injected mice appear to have the same level of fear following blue light stimulation as ChR2 injected mice (please compare Figure 3b with Figure S4C, both appear to freeze ~30% of the time). This suggests that the infection with the virus may be the key variable at play here, rather than ChR2. In Figure S4C, the authors do a single pairwise comparison between CS1 of hM4Di and the pretone and show there is a statistically significant increase in fear. Does CS1 of ChR2 in Figure 3b NOT show a statistically significant increase in fear?

There are two additional sets of experiments that could help provide additional evidence that the authors’ interpretation of the data is correct. (To be more accurate, there are two I thought of- if there is/are better experiment(s) the authors would like to do instead they should feel free.)

1. Redo figure S4c, with uninjected animals in a more direct comparison. For example, if the hM4Di infected animals have freezing rates around 30%, but the uninfected animals do as well, then this result is much less concerning. I understand that freezing rates can vary over time/ replications of experiments.
2. Perform blue light stimulation at 20Hz (as done in all the in vivo experiments of this paper) from ChR2, and separately, from hM4Di, infected mice and record excited currents (or potentials) from mCherry labeled neurons in the PVN.

Reviewer comment #2: Generally, stressor levels are not well matched in conditions across the study, including combinations of isolation housing, novel conspecific introduction, etc.

Thank you for responding so thoroughly to this concern! Please include a version of the reviewer figure you provided in the supplemental figures of this manuscript. Also, lines 80-82: Consider re-writing, or run statistics to show this claim is true. Could simply say ‘SBF elicited a highly statistically significant response to both CS2 and CS1’, or something similar.

Reviewer comment #4: Does increased social interaction (%time nose/nose and %time nose/wall) positively correlate with reduction of fear response? If so, active attention to conspecific may predict SBF expression on Day 3. This measure may also account for some of the individual variability present in the data.

The authors response is greatly appreciated. However, their explanation in the text (lines 266-267) that “after blocking with OTA a correlation seemed to emerge” should be revised. Was there a correlation found in the green points or not? If there was not, please don’t claim that one ‘seemed to emerge’. If a correlation existed but only after pooling the data between the green points and the black points, please say that a correlation existed only after pooling the data between the green points and the black points.

Reviewer comment #6 Since the modulatory effects of Oxt can reconfigure circuits and alter behavior on

the orders of days, the authors might consider expanding the time course of the social buffering retention. The authors perform the retention task just 48 hours after the initial training, which is not necessarily 'long-lasting'. This data may also further draw helpful comparisons between fear extinction—where reappearance of fear responses can occur via renewal, spontaneous recovery, or reinstatement shortly after conditioning— and social buffering of fear that may operate in circuits separate from extinction, as the authors posit, and therefore be a more persistent type of fear inhibition.

Also see reviewer comment #14): In line 201-203, authors state that Type 1 neuron activity “derives directly from OT-sensitive CeA neurons”. This finding is not currently supported by any data provided.

There is no way to exclude numerous indirect circuit effects in these experiments

The authors' claim (lines 208-209) 'after which it remains dependent on OT signaling in the CeA' is 1) puzzlingly placed, as it comes just before an explanation of experiments that strengthen their argument that OT signaling is involved, and 2) is still worded too strongly. This phrase asserts a causal relationship between a peptide and region specific dependency to a specific behavior. This has not been shown in these experiments, although the bulk of the experiments are in agreement with such a claim.

Similarly, on lines 146-148, it is not a given that this is OT signaling. Something like 'OT+ neuron signaling in the PVN' would be more precise.

Reviewer comment #8: Line 121, the authors mention that Day 3 CS1 + no companion group was “similar to CS2” in Fig. 1B. Please provide statistical evidence for this comparison.

The authors' response in lines 89-91 is confusing to me. “It was neither due to the supplementary CS2 (Fig. 1b) resp. CS1 (Fig. 1c) presentation (Day 3 CS1 resp. CS2 + "no companion" was similar to CS2 resp. CS1, Two-tailed Mann-Whitney $U=13$, $p=0.48$, $n=6$ resp. $U=12$, $p=0.39$, $n=6$).” As best I can tell, 'resp' means 'respectively'. I am uncertain what the authors mean by 'CS2 respectively CS1 presentation'.

Reviewer comment #9: Overall, the authors have devised a paradigm that might offer the opportunity to compare SBF-induced fear reduction and fear extinction but do not perform these types of analyses. Also, considering that TGOT/OT stimulation in CeA produces only a transient effect in fear reduction, it is unlikely that SBF is mediated through a pathway “distinct from mPFC-BLA pathway” as mentioned in the introduction and discussion. Cortical activity and BLA are likely needed for the prolonged fear reduction and therefore the authors cannot confirm that PVN-CeA pathway is sufficient for SBF, as mentioned in discussion (line 302), so all statements mentioning the 'independence' of PVN-CeA pathway for SBF or how SBF is 'completely subcortical' should be moderated. The study provides no piece of evidence that BLA is not involved in the used SBF behavioral paradigm.

Citations should be provided for claims in lines 396-400. Please soften claim in line 397 to allow for input from other brain regions in this process (e.g. 'the central elements' instead of 'central elements').

Reviewer comment #10: In fig 5 or supplemental fig 6, authors should provide more data related to waveform/spike sorting and PCA analyses that led to derivation of only three neuron types (e.g.% variance plots, kmeans clustering). Additionally, the authors mention in line 455 of methods that neurons were selected for recording based on responses to CS1/2, however, the only type 2 neurons appear to show a distinct, time-locked response to presentation of auditory cue. Authors should clarify this experimental criterion further and explain how they 'manually' defined clusters as opposed to alternative automated methods.

I have both conceptual and technical concerns regarding the authors clustering of cell types.

Conceptually, I am concerned that by eye, the CeA neurons with colors removed do not appear to sort into 3 clusters.

1. It is unclear how the CeA neurons in Figure 5C are in three clusters, instead of 1, 2, 4, or 5. The authors say 90% of variability variance is explained by 3 clusters... is this substantially better than if there are 2 clusters? Please provide a justification for sorting into 3 clusters.
2. As currently shown in Figure 5C, the clustering has been performed incorrectly/ differently than what is written in the text. The red line from top-most type 1 neuron to the right most type 1 neuron should go directly between the two lines. Instead, it deviates outwards to encompass a nearby PVN neuron. It is unclear why/how this PVN neuron is within the type 1 cluster. This would also suggest that the Figure 5C legend should be corrected: two PVN neurons do not align with the type 1 cluster.
3. (Minor point) 'Type' usually means something in electrophysiological categorization, which is, recorded electrophysiological patterns (waveforms, presence or absence of particular currents, etc). Calling these Type I, type II neurons may confuse many readers. It is not required of the authors, particularly as CeA neurons have generally been classified as Type A, Type B, etc in the literature, but I do suggest considering switching to a different term than 'type'.

Reviewer comment #13: In line 199, authors state "Type 1 neuronal changes correlated closely with PVN neuronal activity", but do not provide any concrete correlational analyses.

The correlation should be provided in the main text. In the absence of what the authors actually compared it is difficult to be certain, as I am uncertain what is meant by 'generated the average firing rate into rows of data' from the response to reviewers. Regardless, the provided correlation coefficient is remarkably high. I strongly urge the authors to be certain the correlation calculation has been performed correctly.

Reviewer comment #16: While it is compelling that the authors found a neuron group that increased activity during the retention of SBF, it would be more significant if the increase in activity was not so generalizable and time-locked to the presentation of the CS. Is this a feature to be expected for maintained inhibition of CeLON neurons (as mentioned in lines 275-76) or should the activity (and therefore posited inhibition of lateral amygdalar neurons) be more time-locked to the conditioned stimulus?

1. Regarding data for Figure 6, the authors should confirm that there is no off-target ChR2 or hM4Di expression in the CeA. It would be very nice to see in a supplemental figure, as off-target expression would greatly alter the interpretation of the results shown here.
2. Regarding figure 6, in the main text, the description of the separation scheme comes before the split neuron examples. Using 'fear' and 'buffer' labels in Fig 6a and b is confusing when those terms have not yet been defined. The figure should be modified to follow the flow of the text (or vice-versa, but I think the authors have the right idea in how they set up the main text).
3. The descriptions of Figure 6g and 6h need additional detail either in the main text or figure legend or both. What does CS2+ mean? How is 'BL+' or 'BL-' defined? What does 'BLnon' mean? What does it mean for a CeA neuron to be neither BL+, nor BL-, nor BLnon, as many of them apparently are? (A similar concern holds for S6G.)
4. If I understand correctly, the authors claim that type 1 neurons in Figure 5 are the same sorts of neurons in Figure 6 (and that type 2 neurons are the same sorts of neurons in the two figures). If my

understanding is incorrect, the authors should clarify the relationship between the type 1s of Figure 5 and the type 1s of Figure 6, etc. If my understanding is correct, the authors should provide more supporting evidence that this is the case. For example, type 1 neurons in Figure 5 show an increase in firing rate upon fear recall on day 2. Do type 1 neurons in Figure 6 do this? (This is just one example, other confirmatory similarities between these populations would be useful as well.)

Minor comments

Reviewer point #19 (minor): Figure 2b: Day 2 (right) plot should have 'SBF' (plot under light purple color) instead of "TGOT" similar to Fig 2a. For this particular experiment, we replaced the "presence of the companion" ("SBF") by the administration of TGOT in the CeA, so there was, in fact, no SBF occurring. TGOT leads to a decrease in freezing to similar in extent to SBF during its administration, but with the important difference that the next day, no memory effect occurs. It thus seems that TGOT cannot fully replace the effect of SBF. To illustrate this better we have, in the revised version of the manuscript, added in figure 2c "instead of SBF") besides the TGOT label.

Lines 159: Maybe say 'in place of SBF'? and maybe say 'three weeks after virus injection, in place of SBF, OTergic cells were...'. This may be too complicated of a change to be worth making, as it is in figures as well, so it is fully up to the authors whether they think it will be useful.

Additional miscellaneous minor comments

Figure 2B: color of some infusion sites is different for no obvious reason, as are some data points in 2B CS1, retention of SBF

Line 139: should be 'hM4Di'

Figure 5A is missing y axis label

Line 256: I am uncertain what 'confir' means.

Suggestions

In the title, maybe change 'oxytocin triggering' to 'oxytocin-triggered'

Line 19: 'intraventrically' to 'intraventricularly'

Line 20: 'facilitate fear extinction to context in the presence'... maybe 'facilitate contextual fear extinction' would be better?

Line 32: maybe 'studied very little' would be better?

Line 36: neither of these commas are necessary

Line 86: maybe 'equally' instead of 'equal'

Lines 98-99: maybe something like "and the amount of fear reduction compared across animals and sessions did not significantly correlate with the level of social interaction"

Line 112: 'OTA also prevented'

Line 147: OT signaling 'from' the PVN instead of 'in'?

Line 303: 'conform our previous in vitro findings' sounds somewhat awkward to me. Maybe, 'in agreement with'? or 'in concordance with' or 'which conforms with' would work better?

Reviewer #2 (Remarks to the Author):

The authors addressed most of the concerns, but one crucial issue remains. In their reply to comment #3 authors argue that the contrast between responses to CS1 and CS2 in Fig. 6 can serve as a "no partner control" for the unit analysis. The counterargument is that animals received more exposure to CS1 than CS2. As a result, the levels of extinction of the CS1-US and CS2-US associations differed when units' responses to CS1 and CS2 were compared. Therefore, the conclusions about the effects of SBF on unit activity remain questionable.

We are glad to hear that the reviewer is happy with most of our responses.

We also thank the reviewer for pointing out the remaining, crucial issue in figure 6, namely that the extra exposure to the CS1 (as compared to the CS2) on Day 2 could cause a decreased unit response on Day 3 not as a result of SBF + CS1 but as a result of the extra CS exposure only. It should be noted here that we found that on Day 3 the responses to CS1 of the fear neurons had completely disappeared, whereas the CS2 responses were fully maintained (Fig. 6g "Day 3"). Although we cannot exclude that the (four) extra exposures to CS1 on day 2 could have caused this complete decrease independently of (without) SBF, we do not think that this is the case for the following reasons:

1. First of all we found that, when we exposed our animals on Day 2 four extra times to the CS1 without social buffering, we did not find on Day 3 any significant decreases in freezing levels to CS1 (Fig. 1b, red solid bars). This result was very robust, we found the same absence of extinction of freezing levels when we switched CS1 with CS2, i.e. in animals exposed four times extra to CS2 without social buffering (as compared to CS1, Fig. 1c, solid blue bars). Thus, in our experiments there is no extinction of freezing as a result of the extra exposure to the CS on day 2 in the absence of social buffering. On the other hand, when we did expose the animals on day 2 to the CS in the presence of social buffering, we found a highly significant decrease in freezing that was maintained on Day 3, regardless of whether the social buffering had been combined with CS1 (fig. 1b, striped red bars, Retention of SBF $p < 0.01$) or with CS2 (fig. 1c, striped blue bars, Retention of SBF $p < 0.001$).

2. Second, we found that the activity of the neurons ("in which the levels of extinction of the CS1-US and CS2-US associations differed when units' responses to CS1 and CS2 were compared") correlated in all aspects in our study with the behavioral freezing levels that we observed to CS1 and CS2. These neurons, which we refer to as "Fear neurons" showed increases in activity to CS1 and CS2 that were highly correlated with the increases in freezing to both CS1 and CS2 (Fig. 6i, right panel). The other way round, following exposure to CS1 after social buffering, their decreases in activity correlated with the decreases in freezing levels (see Fig. 6i, right panel, responses in purple cloud). As a result, the regression lines were highly significant (CS1: $r > 0.75$ and $p < 0.0001$, CS2: $r = 0.80$, $p < 0.0003$, Figure 6i).

A similar correlation has been obtained by colleagues of ours (Whittle et al., Nat. Commun. 2021), who studied the effects of fear learning and extinction on the activity of these CeA fear neurons (which they refer to as "CeLon neurons"). Similar to our "Fear neurons", they identified CeLon neurons in the CeA by their increased firing to the CS after fear conditioning. They also found that the selective decreases in CS1 responses of these single-unit responses in the CeA were always accompanied by the extinction of freezing levels to CS1. And the other way around, single-unit responses to CS2 (not presented on Day 2) were maintained on Day 3 and accompanied by maintained freezing levels to CS2. In fact, our colleagues found, throughout all their experiments ("extinction with CS1 only", "extinction with CS1 and no extinction with CS2", "fear renewal to CS1" and "impaired extinction in another strain", Figs. 1-4, Whittle et al., 2021) a strictly preserved one-to-one relation between single-unit responses to CS1/2 and freezing responses to CS1/2, leading them to conclude that "CS-related responses in this CeLon subpopulation track extinction-induced changes in the expression of fear responses to the CS".

Thus, both in theirs and in our study, changes in single unit responses from this CeA subpopulation track changes in the expression of fear responses to the CS. It seems therefore quite unlikely that in our experiments single-unit responses to CS1 would decrease without corresponding extinction of fear responses to CS1.

3. To confirm this further, we have conducted an extra series of experiments in which we have directly measured spiking activity in CeA fear-recruited neurons after the extra exposure to the CS only (without SBF). We identified these neurons by their increased activity of responses to the CS after fear conditioning (CS+US exposure) and we exposed these units, in this case without social buffering, to the 4 extra CSs (which in our previous behavioral experiments only led to decreased freezing when accompanied by SBF, see Figure 1b and 1c). In these neurons, we also found no significant decreases in responses to the CS (on Day 3, nor one day later, on Day 4). Indeed the CS responses remained at the same level, in spite of the extra exposure to the 4 extra CSs without SBF (see new supplementary figure 9).

Taken together, we believe these findings conclusively and unquestionably show that the decreases in single unit activity are not caused by the extra exposure to the CS alone, but rather, that they are caused by the combination of exposure of SBF in conjunction with the exposure to the CS.

With regard to possible changes in CS responses as a result of pure extinction (without SBF, which was not really our study aim), we would like to refer the reviewer further to this recent work by Whittle et al. (2021). In this study they assessed in great detail the decrease of single unit activity in the CeA as a result of repeated exposure to the CS alone (Nature Communications, Whittle et al., 2021). In their study CeLon neurons (identified by their increased firing to the CS after fear conditioning) DID show decreased responses to CS1 on Day 3, but only after 16 extra consecutive exposures to CS1 on Day 2. The responses to CS2 (which was not extra presented on Day 2) were not changed (See Figure 2b, Whittle et al., 2021). In comparison, in our experiments, we applied only 4 additional consecutive CS1 tones on Day 2 (as compared to no exposure to CS2). Taken together, from these experiments we conclude that the 4 additional CS1 tones to which our animals were exposed, are just not sufficient by themselves to cause decreases in freezing nor to cause decreases in fear-recruited single unit responses.

In the revised manuscript we have included these extra experimental data in new supplementary figure 9 and we have included an extra paragraph in the results and in the discussion about our findings and those of our colleagues.

Results, Line 243: *"The decreased response to CS1 required the presence of the companion, because in the absence of SBF on Day 2 (but in the presence of the extra CS1 exposure) it did not appear on Day 3: Extra exposure to the CS without concomitant SBF on Day 2 did not reduce freezing levels or single unit responses on Day 3 (Fig. 1b&c, Supplementary Fig. 9). In fact, the single unit responses to CS1 and CS2 were highly correlated with the changes in freezing responses to CS1 and CS2 (Figs. 6g & h and Supplementary Fig. 8b,c) throughout these sessions as also reported for CeA CeLon neurons during classical fear extinction³³"*

Discussion: We have added in the paragraph on "SBF effects are different from fear extinction":

Line 453:

"Furthermore, Whittle et al. recently reported that the increased responses to a CS by CeLon neurons after fear learning requires 16 additional consecutive CS presentations for extinction to occur³⁵. It is therefore unlikely that the four extra CS1 presentations during SBF have led to extinction (see also Fig. 2b&c and Supplementary Fig. 9)."

In addition, in Fig. 1 and 2, many groups include only 5-6 subjects, and the freezing variance is high. The experiments appear underpowered.

We thank the reviewer for raising this further concern, but we are a bit surprised that it arises after our first revision. In particular because we increased in our first revision the total number of subjects as compared to the original manuscript. In answer to this concern of the reviewer, we would like to bring forward that:

1. Although the freezing variance may sometimes seem high (e.g. in Fig. 1b), this is also because our data across days and animals were not normalized. In our statistical comparisons, the differences between days always came out as highly significant, even if the variability between animals was high. Thus, with

n=6 subjects we found a highly significant immediate effect of SBF ($p<0.001$) that was maintained on Day 3 ($p<0.01$).

2. Figure 1c originally contained 6 subjects, but for the revised version was increased to 9 subjects. Furthermore it shows that the same immediate and maintained effects of SBF as found in figure 1b, re-appear when we used a CS with a different frequency (CS2, 15 kHz), in both cases with a $p<0.001$.
3. These above, basic findings were reproduced in all control animals of subsequent experiments: In Fig. 2a (control n=5 for blocking effects of OTA on Day 2), Fig. 2b (control n=6 for blocking effects of OTA on Day 3) and Fig. 3a (control n= 5 for blocking effects of CNO). The effects were also found in group-housed animals (n=11, Supplem. Fig. 2a) and in female rats (n=6, Supplem. Fig. 2b).
4. For the experiments in Fig. 2, the variance in freezing was much less and again results were statistically significant.

In the revised version of the manuscript, we now refer to the above findings in the discussion to address this concern of the reviewer:

Line 320:

" Although we did find robust SBF in all our experiments (Figs. 1-3, Supplem. Figs. 1-4) independent of CS frequency, sex, grouped housing between experiments or familiarity with between rats), absolute freezing levels during "Fear Recall" or "SBF" could vary between animals. This might be related to individual differences in endogenous OT signaling: Thus, we found similar increased baseline activity of OTergic PVN neurons and a population of CeA neurons already during "Fear Recall", suggesting that endogenous OT can internally buffer fear through a homeostatic mechanism (Fig. 5a&b)."

Minor:

In all diagrams with behavioral data, please connect the dots representing the same animal along the time course.

As requested by the reviewer, in the revised version of the figures of behavioral data we have connected the dots representing the same animals along the time course. We believe this may also address the previous concern of the reviewer regarding the variance in (non-normalized) freezing" levels between different animals.

We are not completely convinced though that this representation will make the data more accessible. For that reason we have also submitted the version of these figures without lines. We leave it with the reviewer to decide.

We would like to thank the reviewer for the comments and suggestions to improve our manuscript.

Reviewer #4 (Remarks to the Author):

Hegoburu et al. did a great deal of work to respond to reviewer comments, which is applauded and appreciated. The findings are of importance to the field, and the experiments appear well performed thanks to the authors' inclusion of additional details in the methods and by the additional control experiments they have performed. Social buffering of fear is an interesting and important phenomena, and this work provides evidence that the oxytocin-positive neurons of the PVN play an important role in it. We would like to thank the reviewer very much for the extra work taken up to look at the previous comments of previous reviewer 3. We also thank the reviewer for the enthusiastic feedback as well as for the profound analysis of our findings which identified some core remaining issues: we are grateful for raising these points as we believe they have helped us to improve the quality of our manuscript.

There are, however, still concerns that remain not fully addressed, as well as concerns raised by some of the new experiments the authors have performed. Of particular concern are interpretation of the hM4Di + blue light control experiment, the methods and rationale for categorizing CeA neurons in Figure 5 into 3 types, and the sparseness of the evidence underlying the claim that type 1 neurons from Figure 5 are the same type of neuron as type 1 neurons from Figure 6 (and that type 2 neurons from Figure 5 are of the same type of neuron as type 2 neurons from Figure 6).

1) Excerpt from overall reviewer comments: Multiple aspects of study design, analysis, and interpretation of the data diminish the enthusiasm for the strength of current conclusions.

In a new control experiment, the authors inject hM4Di and shine blue light. They then claim (lines 160-161, figure S4C) "BL had no effect in control rats infected with a hM4Di expressing AAV". This claim does not appear as certain as the authors' statement.

First, the hM4Di figure says 'ChR2' on the figure itself, above the data, creating confusion.

Second, hM4Di injected mice appear to have the same level of fear following blue light stimulation as ChR2 injected mice (please compare Figure 3b with Figure S4C, both appear to freeze ~30% of the time). This suggests that the infection with the virus may be the key variable at play here, rather than ChR2. In Figure S4C, the authors do a single pairwise comparison between CS1 of hM4Di and the pretone and show there is a statistically significant increase in fear. Does CS1 of ChR2 in Figure 3b NOT show a statistically significant increase in fear?

We thank the reviewer for raising these points.

First of all, we corrected figure S4C, removing the indication of ChR2, which indeed was wrong. We replaced it with "hM4Di". We thank the reviewer for noticing this.

Second, to answer the reviewer question "Does CS1 of ChR2 in Figure 3b NOT show a statistically significant increase in fear?" we tested in Figure 3b whether the (low) freezing levels in the combined CS1+BL exposure in the animals in which the PVN was infected with ChR2 hM4Di (the red striped bar during BL exposure on day 2) showed any significance with the (low freezing levels) during the pre-tone (before CS1).

This difference was not statistically significant (2-way ANOVA, $F(3,20)= 9.245$, $p=0.000485$, Tukey posthoc test $P=0.66$): "ChR2 + No companion + BL + CS1 was not significantly different from the pretone freezing in fig. 3b"

Third, we compared Figure 3b with Figure S4C and found indeed that these findings were not significantly different. At the same time, we realized that the difference between CS1 on Day 2 during fear recall and during blue light exposure was significant. These results were in fact rather puzzling until we recalled our precise laboratory book notes, and realized that these rats had, by mistake, received both ChR2 and hM4Di injections (which explains the significantly low freezing levels during BL exposure) We have therefore redone these experiments with a new set of animals, now with only hM4Di injections in the PVN and subsequent blue light exposure (see below). These new findings are shown in a completely new Supplementary Fig. 4c. As expected, we found in these animals (that had only been injected in the PVN with the hM4Di virus, instead of ChR2), much higher freezing levels after giving BL+ CS1. These new, much higher levels of freezing have now also been added into revised figure 3b (represented by the orange bar). As expected, these were not statistically significantly different from the (high) freezing levels in animals in which the PVN was not expressing a virus (Fig. 3b, Day 2, ChR2 orange versus red solid bars, Mann Whitney test, $U=10$, $P = 0.21$). This has been added to the figure legend of figure 3.

There are two additional sets of experiments that could help provide additional evidence that the authors' interpretation of the data is correct. (To be more accurate, there are two I thought of- if thereis/are beter experiment(s) the authors would like to do instead they should feel free.)

1. Redo figure S4c, with uninjected animals in a more direct comparison. For example, if the hM4Di infected animals have freezing rates around 30%, but the uninjected animals do as well, then this result is much less concerning. I understand that freezing rates can vary over time/ replications of experiments.

We thank the reviewer for this suggestion. For reasons as explained above, we have repeated our original experiment with a new series of animals in which only hM4Di was injected in the PVN and BL was exposed on the PVN. These new findings now showed a convincing much higher level of freezing responses to the CS1 in the presence of hM4Di +BL (and not statistically significant from the CS1 responses without virus+ BL (Fig. 3b, Mann Whitney test, $U=10$, $P = 0.21$) or without BL/without social buffering (Supplementary Fig. 4c, n.s." two tailed student t-test, $p=0.8985$.)

These new findings have now been included in a new figure S4c and by an additional bar in Fig. 3b. They clearly show that *"the infection with the virus is not the key variable at play here"*. We thank the reviewer for noticing these inconsistent findings, and we are glad that we have been able to correct these and detect our mistake, and thereby improve our manuscript.

2. Perform blue light stimulation at 20Hz (as done in all the in vivo experiments of this paper) from ChR2, and separately, from hM4Di, infected mice and record excited currents (or potentials) from mCherry labeled neurons in the PVN.

We thank the reviewer for this suggestion. In previous work, we have recorded currents as well as potentials from ChR2 infected and non-infected neurons. (Knobloch et al., Neuron 2012) during blue light stimulations at 0, 10, 20 and 30 Hz. Spiking frequencies increased up to 20 Hz linearly according to stimulation frequencies (Fig. 4 and Supplementary Fig. S4, Knobloch et al., 2012). Responses of neurons only occurred in mCherry labeled (ChR2-infected) neurons and neurons without mCherry never responded to any stimulation with Blue light. We have included in the revised manuscript result section a statement referring to these previous findings.

Line 162: *" Also, we had previously found that ChR2 is efficiently activated by blue light (BL) up to frequencies of 30 Hz, hence stimulating neurotransmission in PVN-CeA projections (Knobloch et al., Neuron*

2012), which we further confirmed independently in the present study (Supplementary Fig. 5b)."

2) Reviewer comment #2: Generally, stressor levels are not well matched in conditions across the study, including combinations of isolation housing, novel conspecific introduction, etc.

Thank you for responding so thoroughly to this concern! Please include a version of the reviewer figure you provided in the supplemental figures of this manuscript. Also, lines 80-82: Consider re-writing, or run statistics to show this claim is true. Could simply say 'SBF elicited a highly statistically significant response to both CS2 and CS1', or something similar.

Following the request of the reviewer, we have included this reviewer figure in the revised version of the manuscript as a supplementary figure 2c, which we refer to in the results section as:

Line 101; ".....were independent of sex (female rats showed similar levels, supplementary Fig. 2b) and familiarity with the companion (cage mates induced similar levels, see supplementary Fig. 2c). "

Furthermore, following the suggestion of the reviewer we have rewritten lines 80-82 as follows:

Line 83: "*SBF effects were highly significant and frequency independent: they occurred regardless of whether we co-exposed CS1 or CS2 with the companion (Fig. 1b versus Fig. 1c)*"

3) Reviewer comment #4: Does increased social interaction (%time nose/nose and %time nose/wall) positively correlate with a reduction of fear response? If so, active attention to conspecific may predict SBF expression on Day 3. This measure may also account for some of the individual variability present in the data.

The authors response is greatly appreciated. However, their explanation in the text (lines 266-267) that "after blocking with OTA a correlation seemed to emerge" should be revised. Was there a correlation found in the green points or not? If there was not, please don't claim that one 'seemed to emerge'. If a correlation existed but only after pooling the data between the green points and the black points, please say that a correlation existed only after pooling the data between the green points and the black points.

Thank you for raising this point of precision. We have found a clear correlation between the green points (when we blocked with OTA) without pooling green and black points. We have now changed in the revised version of the manuscript figure 3d to reflect this correlation (green line) and we have revised in the results part the text:

Line 314: "*On the other hand, after blocking with OTA, we found a correlation between freezing and social behavior (Supplementary Fig. 3d, green dots), but we think this is indirectly caused by the increase of freezing caused by OTA (forcibly decreasing social interaction), as OTA applied during the habituation period (Supplementary Fig. 3a) did not directly affect social interaction, nor social motivation.*"

4) Reviewer comment #6 Since the modulatory effects of Oxt can reconfigure circuits and alter behavior on the orders of days, the authors might consider expanding the time course of the social buffering retention. The authors perform the retention task just 48 hours after the initial training, which is not necessarily 'long-lasting'. This data may also further draw helpful comparisons between fear extinction—where reappearance of fear responses can occur via renewal, spontaneous recovery, or reinstatement shortly after conditioning— and social buffering of fear that may operate in circuits separate from extinction, as the authors posit, and therefore be a more persistent type of fear inhibition.

Also see reviewer comment #14): In line 201-203, authors state that Type 1 neuron activity "derives directly from OT-sensitive CeA neurons". This finding is not currently supported by any data provided.

There is no way to exclude numerous indirect circuit effects in these experiments

We agree with the reviewer that this claim is hard to make based on these experiments alone. The opto-tagging strategy in Fig. 4 and 5 may indeed result in the indirect activation of CeA units via other inputs, for example through other brain areas that are activated by OT projections from the PVN and that subsequently activate the CeA. We had conducted a more direct labeling strategy in subsequent experiments shown in revised Fig. 6: With retrograde CAV-cre and pOT-DIO system, we were able to specifically tag the PVN OT neurons projecting to the CeA. After this, we carried out more refined opto-tagging experiments in CeA only: With BL activation (20Hz, 0.5second) in OT fibers in CeA we defined units through their responses to the BL and CS (n=48, N=7), and, based on their responses, we separated them into Fear and Buffer neurons. We believe these experiments provide more solid *in vivo* evidence of the functional link between PVN OXT and CeA neurons. With this approach, we found that the oxytocin receptor antagonist OTA was able to significantly block the responses of these neurons to blue light (Supplementary Fig. S7) further demonstrating the importance of OT signaling for affecting activity in these CeA neurons following the activation of the PVN neurons.

We have rewritten this part through a new paragraph starting with the sentence:

Line 205: *"The strong similarity between these responses in OTergic PVN and this group of CeA neurons suggests that there is a direct connection between the two. Indeed, when we virally expressed Chr2 and inhibitory DREADD hM4Di specifically in OT-ergic PVN neurons that project to the CeA and implanted optrodes one week after virus injection (Supplem. Fig. 7a1), we found that blue light (BL) stimulation directly excited a subpopulation of CeA neurons, and that this response could be inhibited by OTA and completely blocked by CNO (Supplem. Fig. 7a2 and b).*

The authors' claim (lines 208-209) 'after which it remains dependent on OT signaling in the CeA' is 1) puzzlingly placed, as it comes just before an explanation of experiments that strengthen their argument that OT signaling is involved, and 2) is still worded too strongly. This phrase asserts a causal relationship between a peptide and region specific dependency to a specific behavior. This has not been shown in these experiments, although the bulk of the experiments are in agreement with such a claim.

We thank the reviewer for pointing out our "too strong wording". We have changed this in the revised version of the manuscript. We have restructured this part, starting with this question, and we have tried to better explain the link with the subsequent *in vivo* experiments in figure 6, with the following revised text:

Line 225 However, the baseline activity changes we observed do not provide causal proof of this concept. Also, baseline activity does not encode freezing behavior, as the reduction of freezing during and following SBF is specific to the CS that is buffered. As SBF and SBF retention appeared both to depend on OT signaling (Figs 2a,b), one could imagine that the retention of SBF reflects long-term plasticity changes in response to the CS that require both the instantaneous (on Day 2) and continued (on Day 3) activity of OTergic PVN projections^{18,24}. We therefore assessed neuronal responses in the CeA to CS1 and CS2, by viral opto- and chemotagging specifically of PVN OT neurons to CeA neurons pathway, as validated above (Supplem. Fig. S7).

Similarly, on lines 146-148, it is not a given that this is OT signaling. Something like 'OT+ neuron signaling in the PVN' would be more precise.

We agree with the reviewer that we cannot claim, (as we blocked the activity of OT-ergic neurons), "*that OT signaling in the PVN is required*". We have changed this in the revised version of the manuscript into:

Line 152: *"that activity of OT neurons in the PVN is required on Day 2"*.

5) Reviewer comment #8: Line 121, the authors mention that Day 3 CS1 + no companion group was "similar to CS2" in Fig. 1B. Please provide statistical evidence for this comparison.

The authors' response in lines 89-91 is confusing to me. "It was neither due to the supplementary CS2 (Fig. 1b) resp. CS1 (Fig. 1c) presentation (Day 3 CS1 resp. CS2 + "no companion" was similar to CS2 resp. CS1, Two-tailed Mann-Whitney $U=13$, $p=0.48$, $n=6$ resp. $U=12$, $p=0.39$, $n=6$)." As best I can tell, 'resp' means 'respectively'. I am uncertain what the authors mean by 'CS2 respectively CS1 presentation'.

To clarify this point, (and to simultaneously also respond to reviewer 2, comment 1) we changed the text as follows:

Line 92: *"Furthermore, it was not caused by extinction, because only exposure to the CS 3 hours after fear recall (CS1 in Fig. 1b or CS2 in Fig. 1c) in the absence of the companion did not lead to a significant decrease to CS1 respectively CS2 on day 3 (CS1 and CS2 responses on Day 3 were not significantly different (Two-tailed Mann-Whitney $U=13$, $p=0.48$, $n=6$ (Fig. 1b) respectively $U=12$, $p=0.39$, $n=6$ (Fig. 1c)."*

6) Reviewer comment #9: Overall, the authors have devised a paradigm that might offer the opportunity to compare SBF induced fear reduction and fear extinction but do not perform these types of analyses. Also, considering that TGOT/OT stimulation in CeA produces only a transient effect in fear reduction, it is unlikely that SBF is mediated through a pathway "distinct from mPFC-BLA pathway" as mentioned in the introduction and discussion. Cortical activity and BLA are likely needed for the prolonged fear reduction and therefore the authors cannot confirm that PVN-CeA pathway is sufficient for SBF, as mentioned in discussion (line 302), so all statements mentioning the 'independence' of PVN-CeA pathway for SBF or how SBF is 'completely subcortical' should be moderated. The study provides no piece of evidence that BLA is not involved in the used SBF behavioral paradigm.

Citations should be provided for claims in lines 396-400. Please soften claim in line 397 to allow for input from other brain regions in this process (e.g. 'the central elements' instead of 'central elements').

We thank the reviewer for this suggestion. We have included further references and softened the claim in line 397, now starting on line 467 of the revised manuscript: *"the central elements for successful SBF"*

7) Reviewer comment #10: In fig 5 or supplemental fig 6, authors should provide more data related to waveform/spike sorting and PCA analyses that led to derivation of only three neuron types (e.g.% variance plots, kmeans clustering). Additionally, the authors mention in line 455 of methods that neurons were selected for recording based on responses to CS1/2, however, the only type 2 neurons appear to show a distinct, time-locked response to presentation of auditory cue. Authors should clarify this experimental criterion further and explain how they 'manually' defined clusters as opposed to alternative automated methods.

I have both conceptual and technical concerns regarding the authors clustering of cell types. Conceptually, I am concerned that by eye, the CeA neurons with colors removed do not appear to sort into 3 clusters (see below).

1. It is unclear how the CeA neurons in Figure 5C are in three clusters, instead of 1, 2, 4, or 5. The authors say 90% of variability variance is explained by 3 clusters... is this substantially better than if there are 2 clusters? Please provide a justification for sorting into 3 clusters.
2. As currently shown in Figure 5C, the clustering has been performed incorrectly/ differently than

what is written in the text. The red line from top-most type 1 neuron to the right most type 1 neuron should go directly between the two lines. Instead, it deviates outwards to encompass a nearby PVN neuron. It is unclear why/how this PVN neuron is within the type 1 cluster. This would also suggest that the Figure 5C legend should be corrected: two PVN neurons do not align with the type 1 cluster.

3. (Minor point) 'Type' usually means something in electrophysiological categorization, which is, recorded electrophysiological patterns (waveforms, presence or absence of particular currents, etc). Calling these **Type I, type II neurons may confuse many readers**. It is not required of the authors, particularly as CeA neurons have generally been classified as **Type A, Type B, etc in the literature**, but I do suggest considering switching to a different term than 'type'.

We thank the reviewer for pointing out these potential issues of confusion in figure 5. To answer these points, as well as further point 4 below (regarding the relation between different types of neurons in figures 5 and 6), and the original point #13 of reviewer 3 (regarding the correlational analysis between Type 1 and PVN neurons), we have made the following changes:

1. We have dropped the term "type" and now refer directly to these neurons as "fear" or "buffer" neurons, based on our observations in figures 5, 6, and Supplementary Figure 7 together.
2. To avoid the confusion on how many clusters might be found in the CeA neurons, and the search for precise criteria to arrive at a precise distinction, we have discarded the cluster analysis and restructured the buildup of the text between figures 5 (PVN and CeA baseline activities during Fear Recall and SBF), Supplementary figure 7 (effects on baseline activity of blue light stimulation of OTergic PVN projections to the CeA) and figure 6 (combined baseline responses to BL and acute CS responses of CeA neurons during Fear Recall and SBF). We have used the ensemble of these criteria to identify and characterize the "fear neurons" and the "buffer neurons" (see also below).

8) Reviewer comment #13: In line 199, the authors state "Type 1 neuronal changes correlated closely with PVN neuronal activity", but do not provide any concrete correlational analyses.

The correlation should be provided in the main text. In the absence of what the authors actually compared it is difficult to be certain, as I am uncertain what is meant by 'generated the average firing rate into raws of data' from the response to reviewers. Regardless, the provided correlation coefficient is remarkably high. I strongly urge the authors to be certain the correlation calculation has been performed correctly.

See also our answer above. We identified, among others, two groups of neurons in the CeA based on their changes in baseline activity during SBF: those that increased activity during SBF, (similarly to PVN neurons) and those whose baseline activity decreased during SBF after first increase after fear conditioning. These were qualitative classifications without any precise quantitative precision. As it turned out, they gave us a first indication for the differentiation into "fear" and "buffer" neurons that we identified and characterized further in figures 6 and Supplementary Fig. 7.

9) Reviewer comment #16: While it is compelling that the authors found a neuron group that increased activity during the retention of SBF, it would be more significant if the increase in activity was not so generalizable and time-locked to the presentation of the CS. Is this a feature to be expected for maintained inhibition of CeLON neurons (as mentioned in lines 275-76) or should the activity (and therefore posited inhibition of lateral amygdalar neurons) be more time-locked to the conditioned stimulus?

1. Regarding data for Figure 6, the authors should confirm that there is no off-target ChR2 or hM4Di expression in the CeA. It would be very nice to see in a supplemental figure, as off-target expression would greatly alter the interpretation of the results shown here.

In this study, we used the oxytocin promoter viral system that we developed together with Prof. Valery Grinevich (Knobloch et al., 2012) and which proved in the past decade to have a 91% specificity and a

penetration of 89% for labeling oxytocinergic neurons (Knobloch et al. Neuron 2012, Oettl et al. Neuron 2016), labeling subtypes of Magnocellular and Parvocellular oxytocin neurons (Eliava et al. Neuron 2016, Tang et al. Nat. Neurosci. 2020), projection-specific oxytocin neurons (Hasan et al. Neuron 2019, Iwasaki et al. Nat. Commun. 2023), as well as the efficiency of oxytocin neuron-specific DREADD tools (Grund et al. Psychoneuroendocrinology 2019, Tang et al. Nat Neurosci. 2020).

To confirm no off-target expression of both the oxytocin promoter with the CAV-Cre system of Chr2 and hM4Di, we have added magnified staining to show both Gi (Supplem. Fig 5a) and Chr2 (Supplem. Fig 5b) colocalized with OT antibody in the PVN, and OT fibers surrounded with OTR expression in the CeA. To further clarify the virus system: in Fig. 4 and Fig. 5, the virus is injected into the PVN (using the OT promoter AAV) not the CeA, whereas in Fig. 6 and Fig. 7, we used a combinatorial approach in which we injected viruses in both the PVN (using the OT promoter with Cre-dependent AAV) and CeA (using the CAV-Cre), hence we can focus on the specific oxytocinergic pathway from PVN to CeA, thereby excluding indirect pathways caused by activating all oxytocinergic neurons.

2. Regarding figure 6, in the main text, the description of the separation scheme comes before the split neuron examples. Using 'fear' and 'buffer' labels in Fig 6a and b is confusing when those terms haven't yet been defined. The figure should be modified to follow the flow of the text (or vice-versa, but I think the authors **have the right idea in how they set up the main text**).

In line with the earlier suggestions and comments regarding the separation of cells, we felt it necessary to name two cell populations after combining the evidence from figures 5, 6 and Supplementary Figure 7. This has led to the consequence that the naming of the two cell types (buffer and fear neurons) comes relatively late in the revised text. We think that the labeling in figure 6 helps to better appreciate the opposite effects of SBF and the roles of the fear and buffer neurons.

3. The descriptions of Figure 6g and 6h need additional detail either in the main text or figure legend or both. What does CS2+ mean? How is 'BL+' or 'BL-' defined? What does 'BLnon' mean? What does it mean for a CeA neuron to be neither BL+, nor BL-, nor BLnon, as many of them apparently are? (A similar concern holds for S6G.)

We have redesigned this Venn diagram with additional detail in the figure legend

4. If I understand correctly, the authors claim that type 1 neurons in Figure 5 are the same sorts of neurons in Figure 6 (and that type 2 neurons are the same sorts of neurons in the two figures). If my understanding is incorrect, the authors should clarify the relationship between the type 1s of Figure 5 and the type 1s of Figure 6, etc. If my understanding is correct, the authors should provide more supporting evidence that this is the case. For example, type 1 neurons in Figure 5 show an increase in firing rate upon fear recall on day 2. Do type 1 neurons in Figure 6 do this? (This is just one example, other confirmatory similarities between these populations would be useful as well.)

As also mentioned above, our recordings of changes in baseline (as represented in figure 5) only gave us a first indication of the changes in activity and with it the local circuitry that might be underlying the changes during SBF. Only after more targeted labeling with Chr2 of the projections from the PVN to the CeA were we able to better identify neurons that were responding to endogenous activation of OTergic projections from the PVN to the CeA. We can draw the following parallels between the two groups of neurons we found in Figs. 5, 6 and supplementary Fig. 7:

1. Following Chr2 expression specifically in those OTergic PVN neurons that projected to the CeA, we could identify neurons in the CeA whose baseline activity increased after BL stimulation of these projections in the CeA (Supplementary figure 7). This effect was largely blocked by the oxytocin antagonist OTA.

Considering that PVN activation as shown in figure 5 should lead to similar OT release in the CeA, this suggested we were activating the same groups of CeA neurons as identified in figure 5a&b.

2. Similarly, BL allowed us to identify a group of neurons in the CeA whose baseline activity was strongly inhibited in an OT signaling dependent manner. Again, considering the inversely correlated decrease in CeA

baseline activity BL during heightened OTergic PVN baseline activity (Fig. 5a&b) this suggested that we are inhibiting the same group of neurons in the CeA that showed decreased activity during SBF.

3. A further analysis in Fig. 6 of the neurons in the CeA with acute responses to the CS1 that potentiated after FC and disappeared during SBF, revealed that most of them were inhibited by BL stimulation. Similarly, in figure 5 we found this correlation between neurons with this kind of development of responses to CS1 during FC and SBF and their baseline decrease during SBF. It is clear that the separation is not complete and we cannot but point out that some of these CS1 potentiated CeA neurons were also somewhat excited by BL. This is why we have opted to categorize cells until after the description of Fig. 6, as indicated above.

In the main text of the revised manuscript we have now indicated this supporting evidence for the relationship between the two groups of CeA neurons in figure 5 and figure 6.

Minor comments

Reviewer point #19 (minor): Figure 2b: Day 2 (right) plot should have 'SBF' (plot under light purple color) instead of "TGOT" similar to Fig 2a. For this particular experiment, we replaced the "presence of the companion" ("SBF") by the administration of TGOT in the CeA, so there was, in fact, no SBF occurring. TGOT leads to a decrease in freezing to similar in extent to SBF during its administration, but with the important difference that the next day, no memory effect occurs. It thus seems that TGOT cannot fully replace the effect of SBF.

To illustrate this better we have, in the revised version of the manuscript, added in figure 2c "*instead of SBF*") above the figure in the label: "TGOT or vehicle in CeA (instead of SBF)".

Lines 159: Maybe say 'in place of SBF'? and maybe say 'three weeks after virus injection, in place of SBF, OTergic cells were...'. This may be too complicated of a change to be worth making, as it is in figures as well, so it is fully up to the authors whether they think it will be useful.

We have changed the text as the reviewer suggested:

Line 165: "*Three weeks after virus injection, in place of SBF, OTergic cells were stimulated with a blue laser light (BL, 473 nm) during the recall of CS1 (Fig. 3b).*"

Additional miscellaneous minor comments

Figure 2B: color of some infusion sites is different for no obvious reason, as are some data points in 2BCS1.

Thank you for noticing this, we have changed the colors to make them uniform

Line 139: should be 'hM4Di' Figure 5A is missing y axis label.

hM4Di, has been corrected (Line 145), and in Fig. 5A we have added the missing y axis label

Line 256: I am uncertain what 'confir' means.
Has been changed into "compare"(Line 304)

We thank you for your careful observations.

Suggestions

In the title, maybe change 'oxytocin triggering' to 'oxytocin-triggered' Line 19: 'intraventrically' to

'intraventricularly'

Line 20: 'facilitate fear extinction to context in the presence'... maybe 'facilitate contextual fearextinction' would be better?

Line 32: maybe 'studied very little' would be better?

Line 36: neither of these commas are necessary

Line 86: maybe 'equally' instead of 'equal'

Lines 98-99: maybe something like "and the amount of fear reduction compared across animals and sessions did not significantly correlate with the level of social interaction" Line 112: 'OTA also prevented'

Line 147: OT signaling 'from' the PVN instead of 'in'? *We changed this phrase following the earlier suggestion by the reviewer into: "these findings indicate that activity of OT neurons in the PVN is required on Day 2 both for the acute and maintained effects of SBF" (Line 152)*

Line 303: 'conform our previous in vitro findings' sounds somewhat awkward to me. Maybe, 'in agreement with'? or 'in concordance with' or 'which conforms with' would work better? *Changed into "in agreement with, Line 353)*

We have introduced all suggestions in the revised text.

We thank the reviewer for these valuable suggestions, as well as for the in depth reading and insightful comments on our manuscript. These have helped us to substantially improve our manuscript.

REVIEWER COMMENTS

Reviewer #2 (Remarks to the Author):

The authors convincingly addressed the unequal exposure to CS1 and CS2 in Fig. 6. However, the statistical comparisons in several experiments remain underpowered. For example, in Fig. 2a, there are only 5 and 6 subjects for the vehicle and OTA groups, respectively, making it difficult to interpret the near-significant p-value of 0.059 on Day 3 SBF retention. To remedy that problem, authors need to define the statistical power of the analysis and then compute the required number of subjects for each experiment based on the variance and the effect size they aim to detect.

Reviewer #4 (Remarks to the Author):

I appreciate the authors careful consideration of and responses to the reviewer comments. I'm particularly glad they were able to solve the mystery of the non-control control data! I have some additional comments (listed below) but all are incredibly minor (very small text changes or formatting questions) and none will prevent acceptance regardless of how the authors respond. I think the work is insightful and significant, and its publication will help the field greatly.

Minor comments:

1. Line 465: In my previous comments, I incorrectly said that adding 'the' here would soften the statement, when in fact it strengthens it. I would recommend (but don't require) that you remove the 'the' that I asked you to add, as I think it is probably more scientifically accurate. I'm sorry for giving incorrect guidance the first time around on this point!
2. Line 92: "Furthermore, it was not caused by extinction, because only exposure to the CS 3 hours after fear recall (CS1 in Fig. 1b or CS2 in Fig. 1c) in the absence of the companion did not lead to a significant decrease to CS1 respectively CS2 on day 3 (CS1 and CS2 responses on Day 3 were not significantly different (Two-tailed Mann-Whitney $U=13$, $p=0.48$, $n=6$ (Fig. 1b) respectively $U=12$, $p=0.39$, $n=9$ (Fig. 1c)."

I am still confused by the authors' wording here. I think the following edited text may get their point across more clearly.

"Furthermore, it was not caused by extinction, because only exposure to the CS 3 hours after fear recall (CS1 in Fig. 1b or CS2 in Fig. 1c) in the absence of the companion did not lead to a significant decrease to CS1 with respect to CS2 on day 3 (CS1 and CS2 responses on Day 3 were not significantly different (Two-tailed Mann-Whitney $U=13$, $p=0.48$, $n=6$ (Fig. 1b) versus $U=12$, $p=0.39$, $n=9$ (Fig. 1c))."

Trivial comments:

1. S4C, purple panel: the dots are not centered over the bar in the bar chart.
2. The lower left of the confocal images in Figure S5 (all of which look very nice!) is missing a scalebar

legend. Also, it should be μm , not um .

3. Figure 5d: There should be a space between 'retention' and 'of' in 'Retention of SBF'. Maybe there is, but it looks very compressed.

4. Line 228: could read better as 'both appeared'

5. Line 258: there should be a space between 'exhibit' and 'only' and there might be an extra space in this sentence as well.

6. Table S1: 'supraoptic nucleus' is more typical than 'supra optic nucleus'

7. Authors use R in Fig. S3D and r in Fig. S6I.

8. Remember to un-underline your supplemental text before submitting it! (Speaking from embarrassing personal experience, sometimes people forget to do this.)

Answers to Reviewer 2

The authors convincingly addressed the unequal exposure to CS1 and CS2 in Fig. 6. We are glad the reviewer appreciates our answer to this previous question.

However, the statistical comparisons in several experiments remain underpowered. For example, in Fig. 2a, there are only 5 and 6 subjects for the vehicle and OTA groups, respectively, making it difficult to interpret the near-significant p-value of 0.059 on Day 3 SBF retention. To remedy that problem, authors need to define the statistical power of the analysis and then compute the required number of subjects for each experiment based on the variance and the effect size they aim to detect.

Following the suggestion of the reviewer we present below the statistical power analysis that we had performed on the initial series of experiments to determine the required number of subjects for our subsequent studies. These were based on the first results that we had obtained for the "social buffering of fear" (SBF) on Day 2 and the retention of social buffering of fear (Retention of SBF) on Day 3. The precise values for these experiments (rounded off to 2 decimals) are indicated in the table below.

Experiment	Group size used	Effect size based on initial experiments	Power post-hoc	Computed required n subjects (to obtain power > 0.8)
		(rounded off to 2 decimals)		
5 kHz tone				
SBF	6 vs 6	3.85	1	3
Retention of SBF		1.64	0.84	6
15 kHz tone				
SBF	9 vs 9	3.88	1	3
Retention of SBF		2.26	1	4

Although the high power that we obtained from these initial experiments would, in principle, allow us to utilize a relatively small number of animals for subsequent experiments, we did not want to risk an underpowering. Furthermore, since these further experiments were aimed to reveal underlying neurobiological mechanisms they required implanting animals with cannulae (in figure 2) or other surgeries. To allow for some loss of implantations or other possible complications (but without using more animals than necessary) we decided to start out these experiments with a minimal of 8 animals per group. For practical purposes (it was impossible to test 16 animals in the same week), we would split these experiments up in two consecutive sessions of 4 animals per group (4 controls, 4 experimental subjects). To indicate our procedure we have now added in the Material and Methods section the following sentence: "*Power analysis (calculated with G*power 3.1.9.4., Franz Faul, Univ. Kiel, Germany) - was based on the effect size (calculated as Cohen's d effect sizes) of our first observations (in figures 1b and 1c) and with this we designed subsequent experiments with large enough sample sizes to acquire the power above 0.8 to test our hypotheses of interest.*"

With regard to the near-significant p-value of 0.059 on Day 3 SBF retention in figure 2a, we would like to apologize because it concerns a mistake in the figure. In fact, this text comes from a copy and

paste procedure from another figure in Prism Graphpad (unrelated to this manuscript): It occurred because Prism Graphpad does not allow connecting the dots with lines directly in bar graphs. To show the connections between the dots (in answer to the reviewer's previous request), we had to resort to creating two separate graphs (one with dots and another with lines) and plot these two graphs on top of each other.

The real p-value of this difference is in fact $p=0.011$. As the reviewer may have noticed, in the other version of figure 2a (without the connected dots) as well as in all previous versions of the manuscript, this difference is and has, consequently, always been indicated with one asterisk *. We have corrected figure 2a now in the revised version of the manuscript and we thank the reviewer for noticing this inconsistency.

Taken together, with this required number of animals per group we came to a statistical significance of $p<0.05$ or better for all groups that we analysed in the subsequent experiments. We nevertheless thank the reviewer for the remedy to perform a post-hoc power analysis. The results of this power analysis for our different, separately tested hypotheses can be found in the table below. It shows that we had sufficient statistical power of analysis and the required number of subjects for each experiment. Since this result could be mainly useful in a follow up analysis (or in case we had not had significant results in our experiments), we have not included it in the manuscript and only present it here as a complement to fully answer the reviewer's question.

Test	Group size used	Effect size	Power posthoc
		(rounded to two decimals)	
2a – SBF (OTA on Day 2)	5 vs 7	2.90	1.00
2a – SBF retention (OTA on Day 2)	5 vs 7	2.20	0.97
2b – SBF retention (OTA on Day 3)	5 vs 6	1.63	0.80
2c – SBF (TGOT on Day 2)	5 vs 7	2.03	0.94
3a – SBF (CNO on Day 2)	5 vs 6	4.61	1.00
3a – SBF retention (CNO on Day 2)	5 vs 6	2.20	0.96
3b – SBF (ChR2 + BL on Day 2)	6 vs 6	2.24	0.97
S2a – SBF (group housed)	8 vs 11	2.21	1.00
S2a – SBF retention (group housed)	8 vs 11	1.42	0.90

We hope this sufficiently answers the questions of the reviewer.

Answers to Reviewer 4

I appreciate the authors careful consideration of and responses to the reviewer comments. I'm particularly glad they were able to solve the mystery of the non-control control data! I have some additional comments (listed below) but all are incredibly minor (very small text changes or formatting questions) and none will prevent acceptance regardless of how the authors respond. I think the work is insightful and significant, and its publication will help the field greatly.

We thank the reviewer very much for this great, enthusiastic feedback.

Minor comments:

1. Line 465: In my previous comments, I incorrectly said that adding 'the' here would soften the statement, when in fact it strengthens it. I would recommend (but don't require) that you remove the 'the' that I asked you to add, as I think it is probably more scientifically accurate. I'm sorry for giving incorrect guidance the first time around on this point!
2. Line 92: "Furthermore, it was not caused by extinction, because only exposure to the CS 3 hours after fear recall (CS1 in Fig. 1b or CS2 in Fig. 1c) in the absence of the companion did not lead to a significant decrease to CS1 respectively CS2 on day 3 (CS1 and CS2 responses on Day 3 were not significantly different (Two-tailed Mann-Whitney $U=13$, $p=0.48$, $n=6$ (Fig. 1b) respectively $U=12$, $p=0.39$, $n=9$ (Fig. 1c))."

I am still confused by the authors' wording here. I think the following edited text may get their point across more clearly.

"Furthermore, it was not caused by extinction, because only exposure to the CS 3 hours after fear recall (CS1 in Fig. 1b or CS2 in Fig. 1c) in the absence of the companion did not lead to a significant decrease to CS1 with respect to CS2 on day 3 (CS1 and CS2 responses on Day 3 were not significantly different (Two-tailed Mann-Whitney $U=13$, $p=0.48$, $n=6$ (Fig. 1b) versus $U=12$, $p=0.39$, $n=9$ (Fig. 1c))."

Trivial comments:

1. S4C, purple panel: the dots are not centered over the bar in the bar chart.
2. The lower left of the confocal images in Figure S5 (all of which look very nice!) is missing a scalebar legend. Also, it should be μm , not um.
3. Figure 5d: There should be a space between 'retention' and 'of' in 'Retention of SBF'. Maybe there is, but it looks very compressed.
4. Line 228: could read better as 'both appeared'
5. Line 258: there should be a space between 'exhibit' and 'only' and there might be an extra space in this sentence as well.
6. Table S1: 'supraoptic nucleus' is more typical than 'supra optic nucleus'
7. Authors use R in Fig. S3D and r in Fig. S6I.
8. Remember to un-underline your supplemental text before submitting it! (Speaking from embarrassing personal experience, sometimes people forget to do this.)

We thank the reviewer for these precisions and final corrections of our manuscript. We have corrected the text and figures according to the reviewer's suggestions and we appreciate the caring for making sure all is correct.